# Myosin forces remodel F-actin for mechanosensitive protein recognition

Ayala G. Carl[1,2,6], Matthew J. Reynolds[1,6], Xiaoyu Sun[1,6], Pinar S. Gurel[1], Donovan Y. Z. Phua[1], Keith Hamilton[1], Lin Mei[1,2], John W. Watters[1], Yasuharu Takagi[3], Alex J. Noble[4,5], James R. Sellers[3] & Gregory M. Alushin[1✉]

Cells interface mechanically with their surroundings through cytoskeleton-linked adhesions[1,2], which enable them to sense physical cues that instruct development and drive diseases such as cancer[3–5]. Contractile forces generated by myosin motor proteins[6,7] mediate these mechanical signal transduction processes through unknown protein structural mechanisms. Here we show that force generated by myosin elicits structural changes in actin filaments (F-actin) that modulate binding by the mechanosensitive adhesion protein α-catenin[8]. Using correlative cryo-fluorescence microscopy and cryo-electron tomography, we identify F-actin featuring sinusoidal regions of nanoscale oscillating curvature at cytoskeleton–adhesion interfaces enriched in zyxin, a marker of actin–myosin-generated traction forces[9]. We introduce a reconstitution system for visualizing F-actin in the presence of myosin forces using cryo-electron microscopy, which reveals morphologically similar F-actin supercoils. In simulations, compressive forces that mimic myosin activity produce supercoils, which can be generated by ensembles of asynchronous motors regardless of their directionality. Three-dimensional reconstruction of supercoils uncovers extensive asymmetric remodelling of the helical lattice of F-actin. This is recognized by α-catenin, which binds cooperatively along individual strands, preferentially engaging interfaces that feature extended inter-subunit distances while simultaneously suppressing rotational deviations to regularize the lattice. In sum, we find that myosin forces can deform F-actin, generating a conformational landscape that is detected and reciprocally modulated by a mechanosensitive protein, providing a direct structural glimpse at active force transduction through the cytoskeleton.

Myosin motor proteins hydrolyse ATP to apply contractile forces to cytoskeletal F-actin[6,7] linked to plasma membrane-spanning adhesion complexes[1,2], enabling cells to interface mechanically with their tissue microenvironments. Actin–myosin contractility is critical for cell migration[10,11], cell division[12] and multicellular tissue dynamics[3], fundamental processes required for proper morphogenesis and tissue homeostasis that are frequently dysregulated in developmental diseases and cancer[4,13]. Beyond shaping cells and tissues, contractile forces facilitate the transduction of mechanical cues into biochemical signalling pathways that govern cell behaviour[14–16]. How these piconewton-scale forces initiate downstream mechanical signal transduction remains unclear at the protein structural level.

Beyond serving as a force-generating apparatus, the cytoskeleton mediates force transduction through mechanically regulated binding interactions between F-actin and actin-binding proteins (ABPs)[17–20]. F-actin engagement of canonical ABPs can be modulated by force, including regulators of F-actin assembly and disassembly, crosslinkers and cell adhesion proteins[19,20], with the critical cell–cell adhesion protein α-catenin specifically displaying enhanced binding to F-actin in the presence of active myosin force generation[8]. Collectively, these factors contribute to force-sensitive dynamics of cytoskeletal networks underlying cell migration[21,22] and adhesion[23]. Additionally, several proteins containing LIN11, ISL1, and MEC3 (also known as LIM) domains solely bind F-actin in the presence of force[24,25], thereby localizing to the cytoskeleton in the presence of active contractility[26–28]. This force-activated F-actin binding facilitates repair of mechanical damage to the cytoskeleton by zyxin[29,30] and governs mechanosensitive nuclear localization of the transcriptional regulator FHL2 (refs. 16,24). How these proteins detect the presence of force on F-actin is unknown.

F-actin features intrinsic structural polymorphism[31–33] and displays specific conformations when bound by ABPs such as the F-actin disassembly factor cofilin[34–36], suggesting that force could modulate F-actin structure to regulate ABP binding[17]. Indirect reporters of F-actin subunit structural dynamics[32], as well as X-ray diffraction studies of contracting muscle fibres[37–39], have suggested that F-actin rearrangements occur in the presence of myosin activity. Consistently,

[1]Laboratory of Structural Biophysics and Mechanobiology, The Rockefeller University, New York, NY, USA. [2]Tri-Institutional Program in Chemical Biology, The Rockefeller University, New York, NY, USA. [3]Laboratory of Molecular Physiology, Cell and Developmental Biology Center, National Heart, Lung, and Blood Institute, National Institutes of Health, Bethesda, MD, USA. [4]Simons Electron Microscopy Center, New York Structural Biology Center, New York, NY, USA. [5]Present address: CryoArcana, New York, NY, USA. [6]These authors contributed equally: Ayala G. Carl, Matthew J. Reynolds, Xiaoyu Sun. ✉e-mail: galushin@rockefeller.edu

chemical modifications of F-actin[40–42] and actin mutations[43] that are permissive for myosin binding and ATPase activation but refractory to force production have been reported, implying conformational reciprocity between the filament and the mechanochemical cycle of myosin. Cryo-electron microscopy (cryo-EM) structural studies of F-actin–myosin complexes under saturation binding conditions have furthermore visualized subtle alterations in actin subunit conformation accompanying the mechanochemical cycle of the motor[44–47]. However, it is unclear whether and how local changes at the actin–myosin interface propagate along the filament to modulate additional ABP-binding sites. We recently reported that filament bending substantially alters F-actin structure, remodelling inter-subunit interfaces that are engaged by ABPs[48]. Whether active force generation by myosin also substantively remodels F-actin and how ABPs discriminate mechanically excited F-actin structural states through binding contacts remain to be determined.

## Sinusoidal F-actin regions in cells

To investigate whether myosin contractility can modulate F-actin structure in cells, we pursued cryo-electron tomography (cryo-ET) studies. The subcellular localization pattern of zyxin is sufficient to infer traction forces exerted through cell–extracellular matrix (focal) adhesions[9], consistent with the force-activated F-actin binding of the protein. We therefore utilized zyxin as a marker of adhesions featuring high mechanical load. We examined PtK2 cells expressing zyxin–mNeonGreen and the F-actin label F-tractin–mScarlet. After extensive optimization (Extended Data Fig. 1a and Methods), these cells were thin enough to image adhesion–cytoskeleton interfaces without cryo-focused ion beam milling. To increase contractility, we treated cells with Rho Activator II (Methods), a RhoA-activating toxin.

Using correlative cryo-fluorescence microscopy, we targeted zyxin-enriched adhesions at the cell periphery (Fig. 1a, Extended Data Fig. 1b,c and Supplementary Video 1). Semantic segmentation of tomograms showed an abundance of actin filaments in both co-linear bundles (Extended Data Fig. 1b), as previously observed in cryo-ET of focal adhesions marked by paxillin[49], as well as more disorganized networks (Fig. 1b, left and Extended Data Fig. 1c). Many filaments in both network types exhibited substantial curvature (Fig. 1b, left and Extended Data Fig. 1b,c). Of note, some filaments featured sinusoidal regions of sharply oscillating curvature spanning 300–400 nm (corresponding to around 100–150 subunits) that were continuous with canonical straight F-actin (Fig. 1b, right and Extended Data Fig. 1b,c). Quantification of sinusoidal filament region prevalence (Extended Data Fig. 1d–f) showed that these regions were significantly more abundant in zyxin-high subcellular areas (Fig. 1c), which featured overall amounts of F-actin that were indistinguishable from those in zyxin-low areas (Extended Data Fig. 1e). As the morphology of sinusoidal filament regions is distinct from the uniplanar curvature of bent filaments found at sites of propulsive actin polymerization against membranes[50–52], we hypothesized that they could be specifically evoked by myosin forces.

## Myosin forces evoke supercoiled F-actin

To examine whether myosin forces directly modulate F-actin conformation, we reconstituted myosin activity on holey carbon cryo-EM substrates (Methods), adapting our previous approach for analysing force-activated F-actin binding with fluorescence microscopy[8,24]. In this 'dual motor' assay, a mixture of plus (barbed) end-directed myosin V and minus (pointed) end-directed myosin VI are immobilized on a substrate. They engage in a 'tug-of-war' in the presence of ATP, applying mechanical stress to surface-adjacent F-actin (Fig. 1d). In the presence of myosin V or myosin VI individually, fluorescence microscopy confirmed ATP-dependent gliding of F-actin across cryo-EM substrates (Supplementary Video 2), whereas the dual motor condition results

in desultory motions and breakage events (Supplementary Video 3), consistent with filaments coming under mechanical load.

We next plunge-froze specimens to arrest motor dynamics and imaged them by cryo-EM. As the motors are anchored to the carbon film, imaging in holes facilitates visualizing force-dependent rearrangements that are distal from local allosteric effects at motor binding sites. Consistent with our cellular cryo-ET data, we observed sinusoidal F-actin regions that spanned hundreds of nanometres in the presence of active force generation, continuous with canonical straight F-actin regions within the same filament (Fig. 1e). In vitro sinusoidal regions are morphologically similar to those in cells, with a median wavelength of 159 nm (Extended Data Fig. 2a). Quantification showed significant increases in filament curvature and curvature oscillation amplitude in the presence of ATP (Extended Data Fig. 2b), consistent with motor activity enhancing sinusoidal region formation. In addition to sinusoidal regions in filaments spanning holes (Extended Data Fig. 2c), a configuration in which they can bear load, we also observe them in non-load-bearing configurations where they project into holes from the ends of filaments that contact the carbon film along a single edge (Extended Data Fig. 2d). We interpret these to represent remnants of mechanical severing events, as we observed with fluorescence microscopy (Supplementary Video 3). This suggests that sinusoidal F-actin regions have the capacity to persist after filament breakages, when force has dissipated.

As the dual motor system is anticipated to produce a complex distribution of forces, we next assessed the role of motor directionality. We modified our system by polymerizing F-actin from biotinylated seeds (Methods), producing filaments with biotinylated regions at their pointed ends. In the presence of surface-anchored streptavidin, immobilized filaments were then exposed to either myosin VI (pointed-end-directed force) or myosin V (barbed-end-directed force) individually. Fluorescence microscopy of the pointed-end-directed force condition showed straight filaments which began gliding after mechanical ruptures, whereas the barbed-end-directed force condition produced micron-scale buckling, consistent with the anticipated force distribution in each preparation (Supplementary Video 4). Unexpectedly, cryo-EM images revealed the formation of sinusoidal regions in both conditions (Fig. 1f). Sinusoidal regions in the pointed-end-directed force condition featured a significantly longer median wavelength (206 nm) than those in the barbed-end-directed force condition (166 nm), but they were otherwise morphologically similar (Fig. 1g). To assess the impact of filament annealing events in the pointed-end force condition, which could produce filament segments with biotinylated regions at their barbed ends, we implemented a reconstitution procedure refractory to this phenomenon (Extended Data Fig. 2e,f and Methods). We nevertheless observed sinusoidal filament regions (Extended Data Fig. 2g), which were significantly enhanced in the presence of ATP (Extended Data Fig. 2h). These observations suggest that sinusoidal region formation is an intrinsic response of F-actin to myosin forces, which is insensitive to motor directionality.

To probe the ultrastructure of sinusoidal regions, we pursued cryo-ET studies of both the barbed-end-directed and pointed-end-directed force conditions. Although visual inspection suggested that sinusoidal regions exhibited three-dimensional (3D) character (Fig. 2a and Supplementary Video 5), degraded resolution in the $z$-dimension due to incomplete tilt angle coverage (the 'missing wedge' artefact) precluded detailed analysis. To overcome this limitation, we applied a tomogram denoising approach[53], enabling direct filament tracing for quantification by principal component analysis (PCA) decomposition (Fig. 2b and Methods). Alignment of filament trace projections revealed an approximately quarter wavelength phase offset between the oscillatory components of each filament region, despite substantial morphological variability (Fig. 2c,d), consistent with sinusoidal filament regions adopting a supercoil morphology that corkscrews around the filament axis.

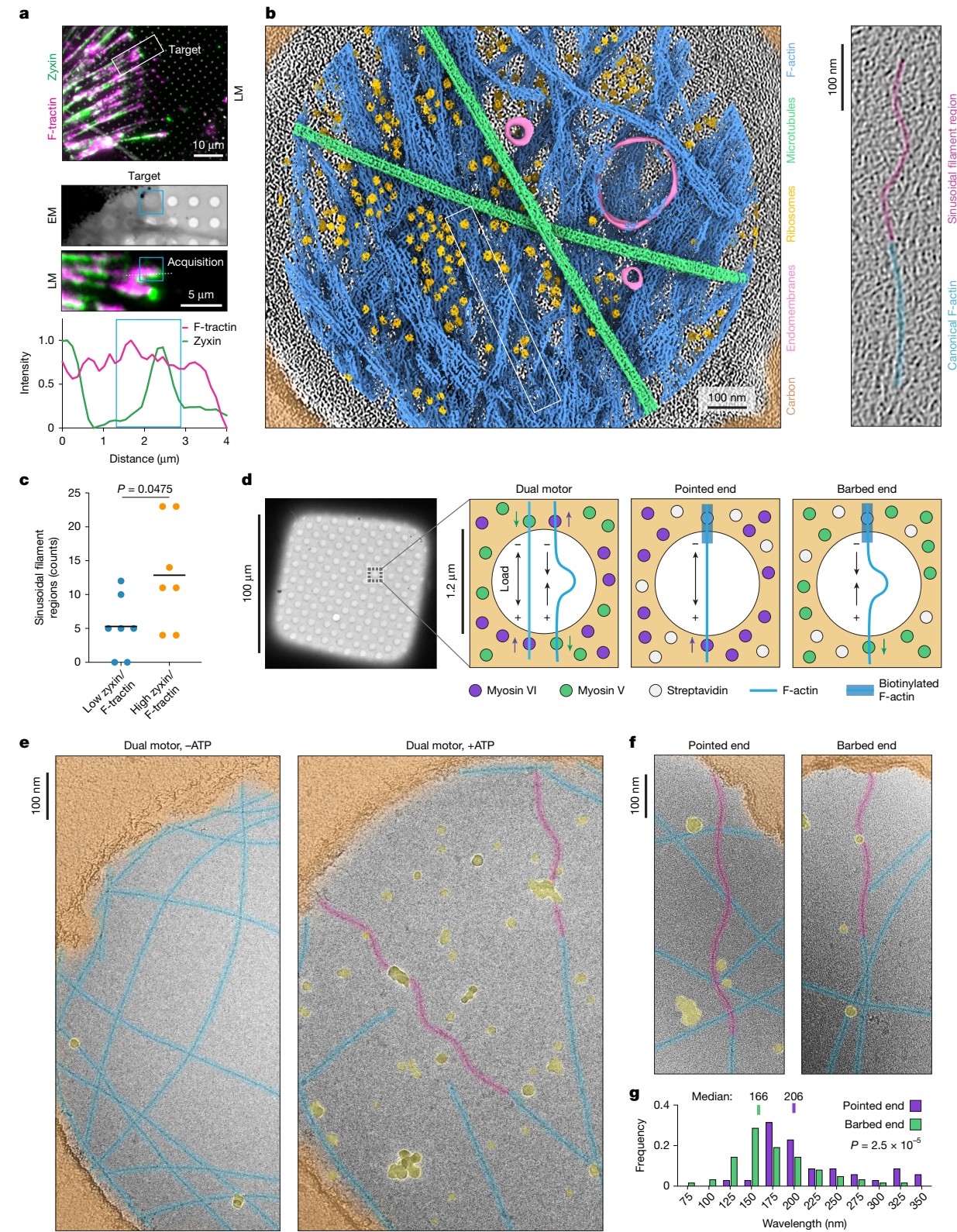

**Fig. 1 | Myosin forces evoke sinusoidal regions in F-actin. a**, Top, low-magnification cryo-light microscopy (LM) image of a PtK2 cell, highlighting targeted adhesion. Middle, medium-magnification correlation between cryo-EM and cryo-LM, highlighting the site of tomogram acquisition. Bottom, fluorescence intensity scan along the dashed line. Blue box indicates acquisition area, which is enriched in zyxin. **b**, Left, segmented tomogram. Right, false-coloured 5.1-nm-thick projection of boxed area, highlighting the F-actin sinusoidal region. **c**, Quantification of sinusoidal filament regions in low zyxin/F-tractin (*n* = 7) versus high zyxin/F-tractin (*n* = 7) subcellular areas.

Bars represent means; conditions were compared with an unpaired two-tailed *t*-test. **d**, Schematic of the myosin force reconstitution assay. **e**, False-coloured cryo-EM images of dual motor reconstitution in the presence and absence of ATP. Sinusoidal regions, magenta; canonical F-actin, blue; carbon film, orange; ice contamination, yellow. **f**, Cryo-EM images of sinusoidal regions formed in single-motor conditions, false-coloured as in **e**. **g**, Quantification of sinusoidal region wavelengths in the pointed-end-directed (*n* = 35) and barbed-end-directed (*n* = 63) conditions, from two independent experiments. Distributions were compared with an unpaired two-tailed Mann–Whitney test.

Supercoils featured elliptical rather than circular cross-sections (as would be anticipated for ideal supercoils) with major and minor axis lengths of $20.1 \pm 5.6$ nm and $8.6 \pm 4.6$ nm (mean ± s.d.), respectively, in the pointed-end-directed force condition versus $14.0 \pm 6.8$ nm and $6.2 \pm 3.1$ nm in the barbed-end-directed force condition (Fig. 2c,d). The major axes were randomly oriented relative to the plane of the ice film (Extended Data Fig. 2i), suggesting that this asymmetry results from the helical architecture of F-actin rather than flattening imposed by surface tension. We also occasionally observed sites that featured protruding densities suggestive of dislocated actin subunits (Extended Data Fig. 2j), supporting an association between supercoil formation and mechanical damage to F-actin. Collectively, these data show that myosin forces can directly generate F-actin regions that feature an asymmetric supercoil morphology.

## Compressive forces produce supercoils

Myosin force generation is complex, featuring ATP hydrolysis-coupled F-actin binding and unbinding during the mechanochemical cycle of the motor[6,7]. To assess whether these features of myosin force generation could contribute to directionality-insensitive F-actin supercoil formation in our cryo-EM experiments, we conducted coarse-grained molecular dynamics simulations in which actin subunits were abstracted as spherical bodies (Fig. 2e–h; Extended Data Figs. 3 and 4, Supplementary Video 6 and Methods). We first performed control simulations of thermal fluctuations in the absence of external force for filaments with zero, one or two ends fixed to mimic different levels of tethering or engagement by motors (Fig. 2e and Extended Data Fig. 4a). We observed minimal supercoiling in all of these conditions, evidenced by the low peak-to-peak amplitude of curvature oscillations (Fig. 2f). Correspondingly, using a peak-to-peak amplitude cut-off of 16 nm (approximately twice the diameter of an actin filament), less than 10% of simulation frames featured supercoiling (Extended Data Fig. 3e).

We then performed simulations in which the filament (which lacks polarity in our coarse-grained model) was fixed at one end, then either tension or compression was applied on the other end. The application of constant compression nearly instantaneously resulted in supercoil formation, whereas constant tension conversely straightened the filament, suppressing thermal fluctuations (Fig. 2e and Supplementary Video 6). In compression simulations, the amplitude of curvature oscillations decayed with distance from the site of force application, an effect that was attributable to viscous dissipation by the thermal bath. This is consistent with our experimental observation that sinusoidal filament regions are localized at the edges of holes (where motors are tethered to the carbon), transitioning into canonical straight F-actin towards hole centres (Fig. 1e,f).

To calibrate the duration of force application, we implemented a maximum bond length cut-off based on structural analysis (Methods), beyond which bonds rupture. Compression promoted supercoil formation prior to filament rupture, whereas tension resulted in rupture without supercoiling (Extended Data Fig. 4b and Supplementary Video 6). Supercoiling persisted after filament rupture in the compression condition, consistent with our inference that mechanical severing is likely to produce sinusoidal regions on filaments attached to the carbon at one end in our experiments (Extended Data Fig. 2d). Restricting the duration of force application to the period prior to filament rupture, we quantitatively analysed the relationship between force and supercoil formation and properties. Tension effectively suppressed supercoil formation at all force magnitudes, whereas increasing the magnitude of compression increased the median amplitude of oscillations and supercoil prevalence (Fig. 2g and Extended Data Fig. 3e). Median supercoil pitch (the equivalent of wavelength in our two-dimensional (2D) analysis of sinusoidal regions) remained essentially constant at around 275 nm across the conditions that we examined, deviating from our experimental observations (Extended Data Fig. 3f). We speculate that

this discrepancy is due to the rupture bond length cut-off we used, a parameter that would require future experimental studies to optimize.

Although compression is anticipated to dominate in the barbed-end force condition in our experiments, intuition suggests that tension should predominate in the pointed-end condition (Fig. 1d). As constant tension did not produce supercoils in our simulations, we next probed how motor dynamics could contribute to their formation. To mimic the dynamics of motor mechanochemical cycles, we examined the dynamics of the filament after force was released in each condition. Supercoils persisted after the release of compression (Extended Data Fig. 4c), which were morphologically similar to those produced during the application of constant compression (Fig. 2g,h and Extended Data Fig. 3e,f). Conversely, upon tension release, the filament relaxed and displayed dynamics statistically indistinguishable from thermal fluctuations (Fig. 2f,h and Extended Data Fig. 3e,f). As our experiments feature ensembles of uncoordinated motors that may simultaneously operate on individual filaments, we next performed simulations that mimicked the activity of five stochastically firing motors. Compression with random firing once again produced supercoiling, whereas tension continued to resemble thermal fluctuations (Extended Data Fig. 4d).

As tension did not produce supercoiling in any of the simulation conditions that we examined (Extended Data Fig. 4f), we posit that, counterintuitively, compression is also responsible for the formation of sinusoidal filament regions in the pointed-end force condition. Ensembles of stochastically firing myosin VI motors could also impose local compression when a subset of motors effectively operate as tethers during the strong F-actin-binding phase of their mechanochemical cycles (Extended Data Fig. 4g). If a tethering molecule is localized towards the barbed-end of a filament relative to a force-generating molecule, compression will be imposed on the intervening segment, a phenomenon that has previously been suggested based on interferometric fluorescence microscopy studies of myosin VI gliding filament assays[54]. In our experiments, this could conceivably occur on either biotin-anchored or unanchored (seed-detached) filaments, which are also present in our preparations (Extended Data Fig. 2f).

Additional simulations (Extended Data Fig. 4e; Supplementary Video 6 and Supplementary Discussion) show that torque[55,56] does not contribute to supercoiling. Varying the mechanical properties of F-actin and helical parameters (notably rise) affect supercoil architecture (Extended Data Fig. 3g,h), whereas the ellipticity of supercoils appears to arise through their temporal evolution from uniplanar buckles (Extended Data Figs. 3i and 4h and Supplementary Video 6). Collectively, these analyses suggest that axial compression generated by stochastically operating motor ensembles can produce F-actin supercoils, regardless of motor directionality, without the requirement for torque.

## Structure of supercoil F-actin

We next sought to directly visualize the structure of myosin force-evoked F-actin supercoils using single-particle cryo-EM. We focused on the dual motor condition, as we found that it generated the greatest abundance of supercoils, collecting datasets in the absence and presence of ATP (Extended Data Fig. 5 and Extended Data Table 1). We adapted a neural network-based approach that we previously developed for analysis of F-actin bending[48] to estimate the signed curvature of filament segments (Extended Data Fig. 5a,b and Methods), facilitating specific detection of sinusoidal filament regions. The overall distribution of absolute curvatures in both datasets deviated markedly from F-actin undergoing thermally driven bending fluctuations[48,57], with an increased proportion of low-curvature segments, probably owing to myosins tethering F-actin to the carbon film (Extended Data Fig. 5c). We nevertheless observed a long tail of highly curved segments, which was significantly increased in the presence of ATP (Extended Data Fig. 5c).

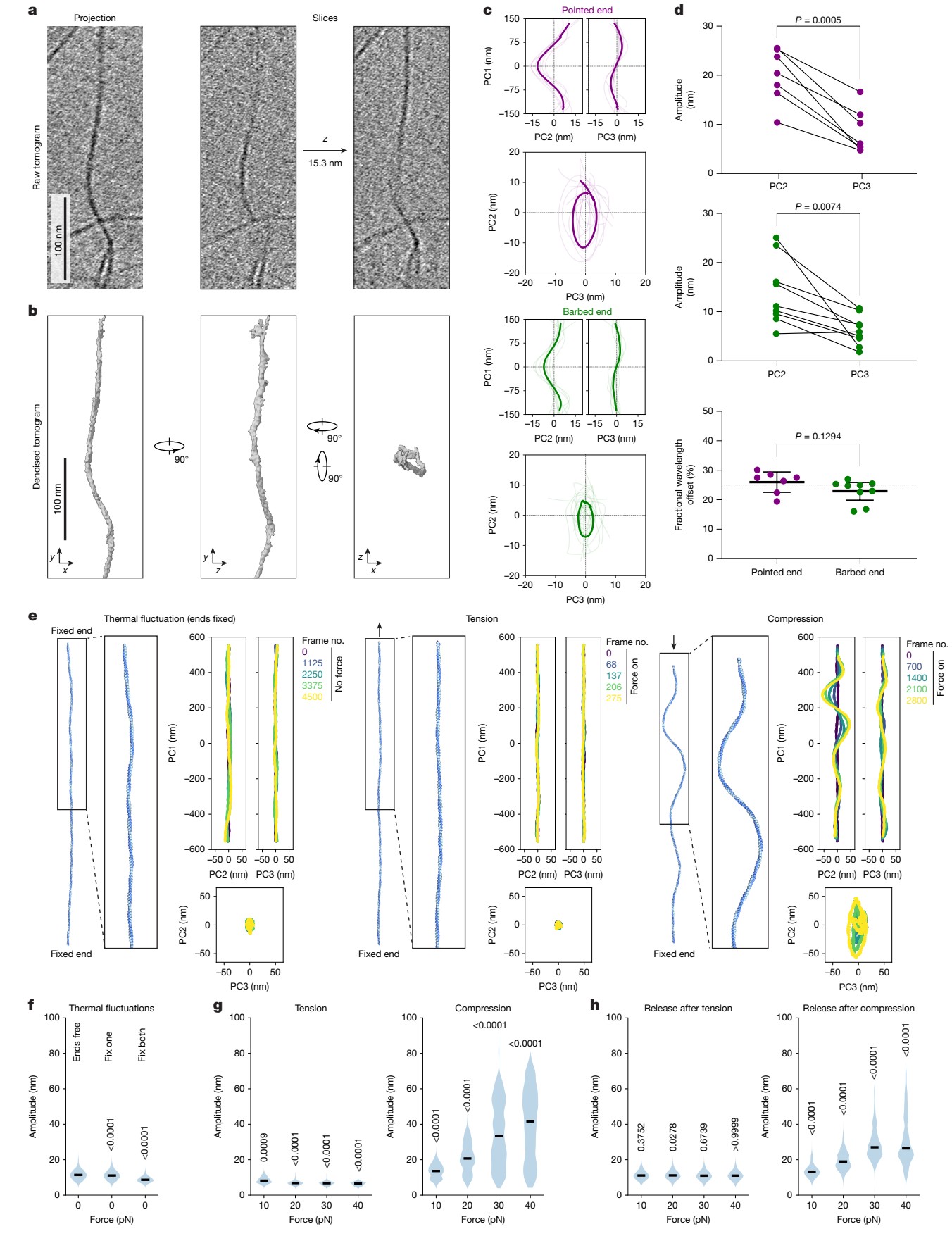

**Fig. 2** | See next page for caption.

**Fig. 2 | Compression produces F-actin supercoils. a**, Projection (left) and serial slices (right) of an F-actin sinusoidal region tomogram (pointed-end-directed force condition). **b**, Orthogonal views of denoised density from the tomogram in **a**, highlighting supercoiling oscillations in $xy$ and $xz$ dimensions. **c**, Aligned projection views along principal components (PC1, PC2 and PC3) of filament traces with supercoil character in pointed-end-directed (top; $n = 7$) and barbed-end-directed (bottom, $n = 9$) force conditions. Transparent lines represent individual filament traces and solid lines represent averages of aligned traces. **d**, Quantification of data from **c**. Analysis of filament trace amplitudes in PC1–PC2 versus PC1–PC3 planes from pointed-end (top) and barbed-end (middle) force conditions, compared by paired two-tailed $t$-test. Bottom: fractional wavelength offset between trace projections in PC1–PC2 versus PC1–PC3 planes. The centre line represents mean and error bars represent 95% confidence interval. The dashed horizontal line indicates an offset of 25%. Conditions were compared by unpaired two-tailed $t$-test. **e**, Representative simulation snapshots and PCA of key frames in indicated conditions (thermal fluctuations with both ends fixed and constant tension–compression). **f–h**, Quantification of 3D peak-to-peak amplitude of evenly sampled simulation frames for thermal fluctuations (**f**; $n = 25,000$), constant tension and compression (**g**; $585 \leq n \leq 13,350$), and release after tension and compression ($23,310 \leq n \leq 24,420$). Horizontal lines represent medians; $P$ values are indicated for statistical comparisons (Kruskal–Wallis test followed by post hoc Dunn's multiple comparisons test) versus thermal fluctuations with both ends free (**f**; $P < 0.0001$ for both cases), thermal fluctuations with both ends fixed (**g**; $P = 0.0009$ for 10 pN tensile force and $P < 0.0001$ for all other tensile forces tested; $P < 0.0001$ for all compressive forces tested), or thermal fluctuations with one end fixed (**h**; 10 pN: $P = 0.3752$; 20 pN, $P = 0.0278$; 30 pN, $P = 0.6739$; 40 pN, $P > 0.9999$; $P < 0.0001$ for release after all the compressive forces tested). Five simulation runs per condition.

---

We focused on filament regions featuring contiguous stretches of sharp positive and negative curvature (Methods). After extensive classification (Extended Data Fig. 5d), a subset of 13,146 segments produced a consensus 9.5 Å resolution asymmetric reconstruction spanning 25 protomers (Fig. 3a and Extended Data Figs. 5e and 6). Stitching multiple copies of this reconstruction (Fig. 3a) yielded a volume that morphologically resembled supercoils observed in tomograms (Fig. 2a,b and Supplementary Video 7). This stitched reconstruction features a circular, rather than elliptical, cross-section, deviating from our tomography and simulation results. We infer this to be the consequence of averaging heterogeneous filament segments, each with potentially uniquely varying curvature. Consistently, cryoSPARC three-dimensional variability analysis (3DVA) supports extensive conformational variability (Extended Data Fig. 5e). This heterogeneity, along with the low particle number in the final reconstruction (on average 1 per 2.4 micrographs), limited the achievable resolution.

To probe changes in the helical lattice formed by F-actin, we rigid-body docked actin subunits into the map and analysed their positioning along a deformed filament axis determined by numerical fitting (Fig. 3b and Methods). We also performed this analysis on two control reconstructions determined from the no-ATP dataset (Extended Data Fig. 6a,g), featuring low curvature and helical parameters similar to those of canonical F-actin (Extended Data Fig. 6g). Conversely, in the supercoil F-actin reconstruction, we observe extensive architectural remodelling (Fig. 3b). Helical twist exhibits a similar wavelike pattern to bent F-actin[48] (Extended Data Fig. 6h), where one strand is over-twisted and the other under-twisted while maintaining the canonical F-actin twist as their instantaneous average. This supports twist–bend coupling as a general response of the F-actin lattice to curvature[58]. As observed with uniplanar bending[48], the twist deviations of supercoil F-actin exceed those imposed by cofilin binding[34–36]. However, supercoil F-actin also features distinct and extensive modulation of helical rise (Fig. 3b), which is only modestly affected by bending[48] (Extended Data Fig. 6h). The strands exhibit alternating over-extension and under-extension, once again occurring in a wavelike pattern around an instantaneous average matching that of canonical F-actin (Fig. 3b). Consistent with our simulations showing coupling between F-actin's helical rise and supercoil architecture (Extended Data Fig. 3g), these data support rise remodelling as a specific feature of supercoil F-actin formation.

We next examined structural deformations of individual actin protomers (Fig. 3c and Extended Data Fig. 7). Owing to the modest map resolution, we performed molecular dynamics flexible fitting (MDFF) in ISOLDE with extensive restraints that restricted changes to repositioning of the four subdomains of actin (Methods). Whereas fitting the control reconstructions uncovered negligible transitions (Fig. 3c and Extended Data Fig. 7), supercoil F-actin featured substantial subunit deformations characterized by inward compaction of subdomains 1 and 4 into the nucleotide cleft. This contrasts with bent F-actin, which features subunit shearing outwards from the filament axis[48].

Additionally, the absolute curvature distribution of segments which met our oscillating curvature selection criteria (Extended Data Fig. 5c) deviates markedly from bending fluctuations[48]. These data collectively suggest that myosin force-evoked supercoil F-actin is structurally distinct from bent F-actin.

## α-Catenin modulates supercoil F-actin

As myosin forces regulate F-actin engagement by force-sensitive ABPs[8,24,25,59], we postulated that these proteins may discriminate supercoil F-actin through binding contacts. To test this hypothesis, we focused on the isolated actin-binding domain (ABD) of αE-catenin, which displays enhanced F-actin binding in the presence of myosin forces[8]. We prepared dual motor cryo-EM specimens with 1 μM α-catenin and 0.6 μM F-actin (Methods), a sub-saturating regime of α-catenin in which force-sensitive binding is observed. After multiple rounds of classification (Extended Data Fig. 8), we obtained a 10.4 Å resolution consensus reconstruction (Extended Data Fig. 9 and Extended Data Table 1). This map features asymmetric decoration, with α-catenin preferentially binding along one strand, then switching to the other at the crossover where the strands exchange sides along the filament. This binding mode is distinct from structures obtained under saturation binding conditions in the absence of force, where complete symmetric decoration was observed[8,60].

To probe the links between F-actin conformational remodelling and force-dependent α-catenin binding, we performed 3DVA (Extended Data Fig. 8, Supplementary Video 8 and Methods). Inspection of the first variability component revealed correlations between α-catenin binding and F-actin rearrangements (Fig. 4a). To quantify α-catenin binding, we rigid-body docked the high-resolution structure of an α-catenin ABD–F-actin interface (Protein Data Bank (PDB): 6UPV[8]), composed of two actin protomers and one ABD, at each binding position in each frame of the 3DVA trajectory. We then measured the integrated intensity of the map within a mask calculated from the ABD coordinates, a proxy for α-catenin occupancy. Plotting filament curvature (Methods) versus average α-catenin intensity in each frame shows two high-curvature regions in the trajectory (Fig. 4b), only one of which also displays high average α-catenin intensity. Examining the helical parameters of these two 3DVA trajectory regions (Fig. 4c and Extended Data Fig. 10a) shows that although the low α-catenin intensity frames are similar to unbound supercoil F-actin, the high α-catenin intensity frames display a distinct pattern, featuring even larger rise deviations that persist further along each strand without oscillations, coupled to suppression of twist deviations.

To examine how F-actin remodelling is linked to α-catenin binding, we selected particles that contributed to frames with high α-catenin intensity. Refinement produced a 12.3 Å resolution reconstruction (Extended Data Figs. 9 and 10b and Extended Data Table 1) featuring a similar α-catenin binding pattern (Extended Data Fig. 10c) and

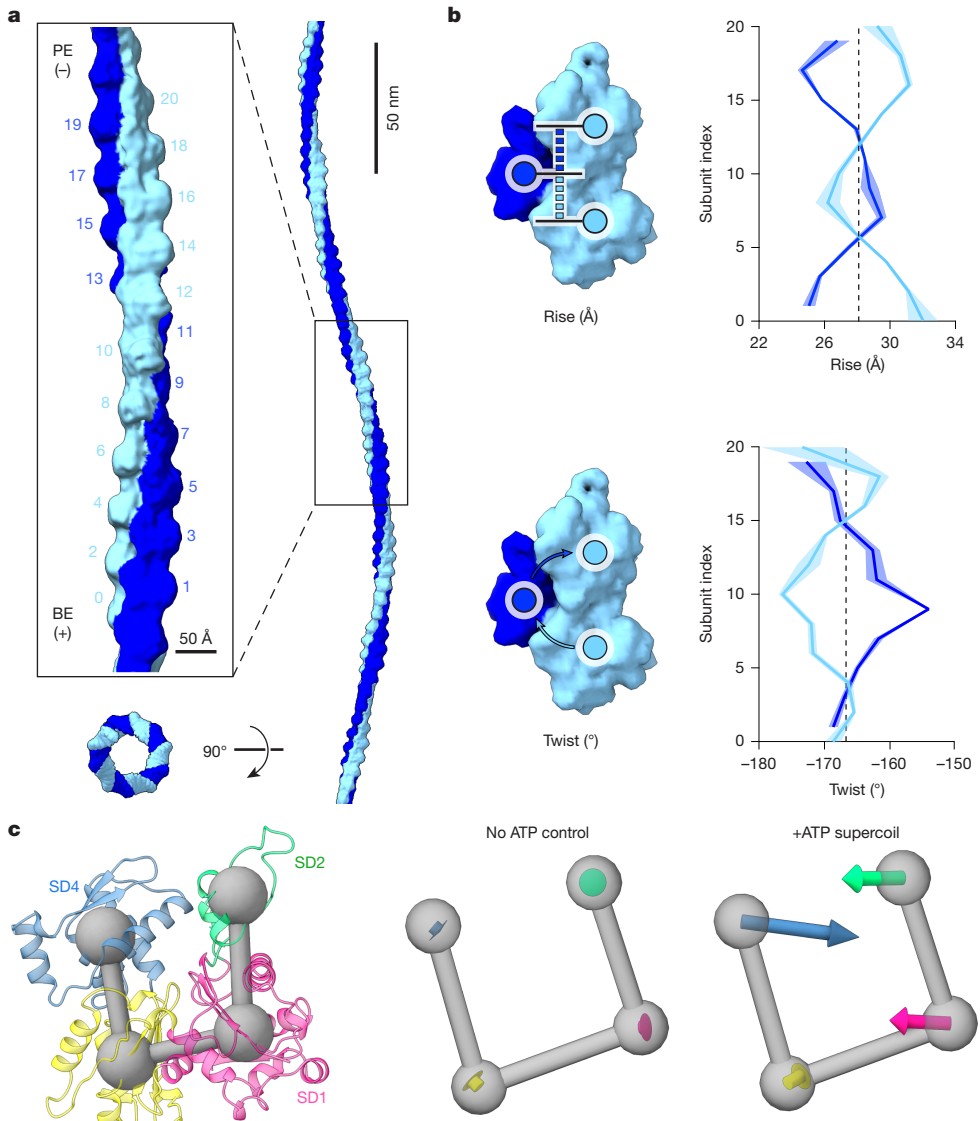

**Fig. 3 | Supercoil F-actin features unique architectural remodelling and subunit deformations. a**, Left, cryo-EM density map of supercoil F-actin reconstructed from the dual motor condition. Strands are coloured in alternating shades of blue, and subunit indexes are indicated. BE, barbed-end; PE, pointed-end. Right, stitch of five copies of the map, recapitulating the morphology of sinusoidal filament regions. **b**, Diagrams (left) and plots (right) of instantaneous helical parameters. Strands are coloured as in **a**. Shaded regions represent the 95% confidence interval from three independent analyses. Vertical dashed lines indicate parameters of canonical F-actin[48]. **c**, Left, ribbon diagram of an actin subunit, with subdomains coloured in varying shades. Subdomain centroids are indicated by connected grey spheres. Right, averaged subdomain displacements (scaled 15× for visualization) after MDFF analysis relative to a canonical F-actin subunit (PDB: 8D13) for the indicated conditions. SD, subdomain.

helical parameters (Extended Data Fig. 10d). Despite the low map resolution, density connecting longitudinally adjacent ABDs was visible (Fig. 4d), which docking analysis assigns as the ordered portion of the C-terminal extension of α-catenin (residues 865–871), a region previously implicated in mediating inter-ABD contacts that is necessary for force-activated actin binding[8]. We then examined the relationship between α-catenin intensity and the instantaneous rise at each protomer index, revealing preferential α-catenin engagement at positions featuring extended rise versus canonical F-actin (Fig. 4e).

To assess whether preferential α-catenin binding is associated with actin subunit remodelling, we performed MDFF and examined actin subdomain rearrangements (Fig. 4f and Extended Data Fig. 10e). Although the low map resolution warrants caution in interpretation, all actin subunits feature apparent repositioning of subdomain 2, a flexible segment that mediates inter-subunit contacts, regardless of α-catenin intensity (Fig. 4f and Extended Data Fig. 10e). This suggests that substoichiometric

α-catenin binding to mechanically excited F-actin also modulates actin conformation at unbound positions, consistent with the established role of subdomain 2 in F-actin structural plasticity[33,44–48]. However, high-occupancy positions feature a characteristic actin rearrangement adjacent to α-catenin ABD-binding contacts, where subdomain 2 is pulled away from the filament core (Extended Data Fig. 10f), a conformation that is distinct from both canonical F-actin and unbound supercoil F-actin (Fig. 4f and Extended Data Fig. 10e). This subdomain 2 rearrangement is coupled to reduced compaction of subdomains 1 and 4, which is likely to relieve steric strain. Together, these data show that α-catenin asymmetrically binds F-actin in the presence of myosin forces, preferentially engaging positions with extended rise while suppressing twist deviations. Cooperative inter-α-catenin ABD contacts are likely to stabilize these lattice transitions, as are binding contacts that license actin subdomain 2 rearrangements, relieving force-induced steric strain on subunits.

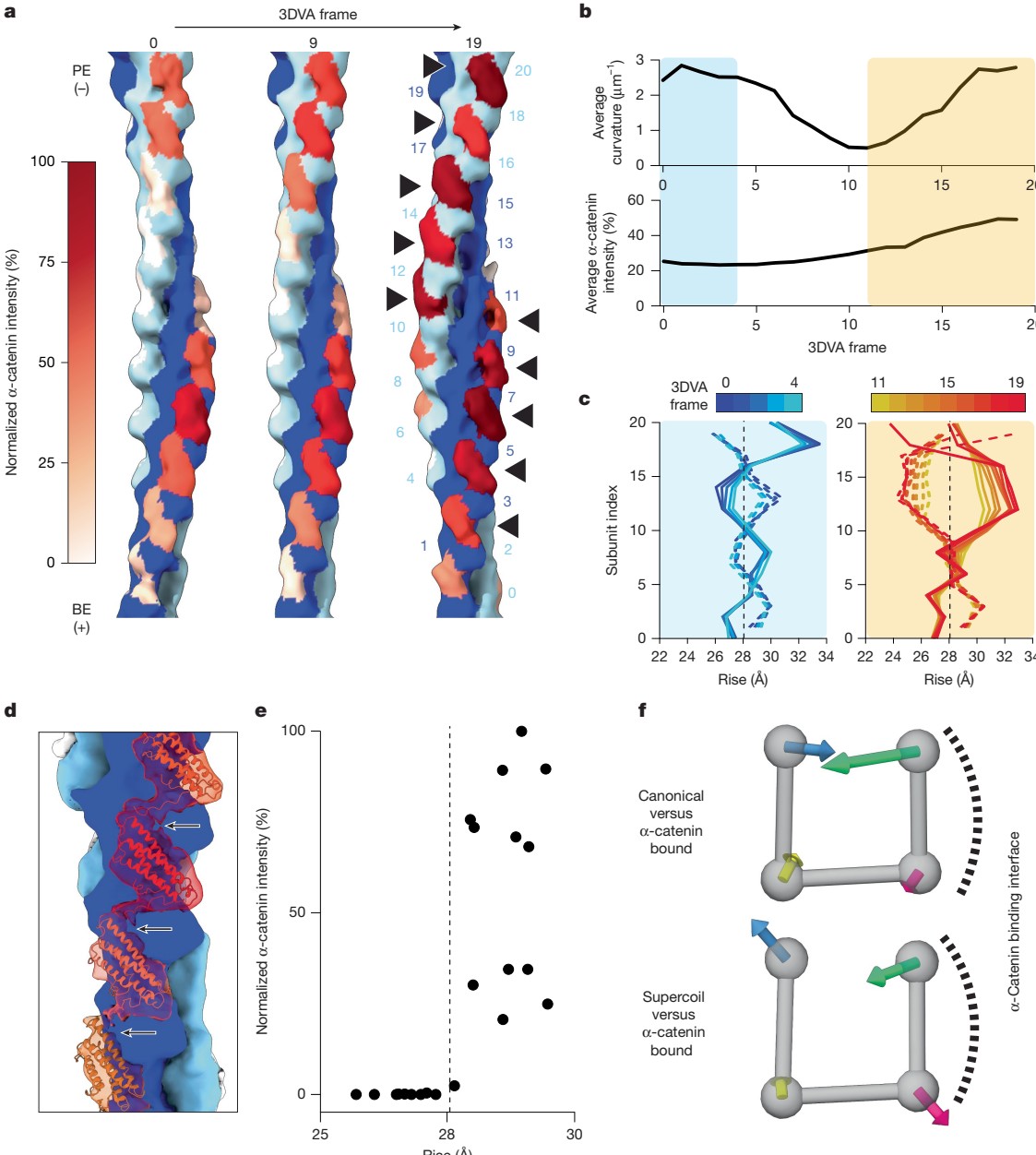

**Fig. 4 | α-Catenin detects and reciprocally modulates force-evoked structural changes in F-actin. a**, Frames from 3DVA of the α-catenin ABD–F-actin complex reconstructed in the dual motor force condition, highlighting alternating binding along F-actin strands (arrowheads) and varying intensity of α-catenin density across the trajectory. Subunit indexes are indicated. **b**, Quantification of average filament curvature (top) and average α-catenin intensity (bottom) across the 3DVA trajectory. Boxes indicate segments with high filament curvature and either low (blue) or high (orange) α-catenin intensity. **c**, Plots of instantaneous helical rise from trajectory regions indicated in **b**. Vertical dashed lines indicate canonical F-actin rise. **d**, Docking analysis (PDB: 6UPV) of consensus reconstruction highlights inter-ABD contacts mediated by the C-terminal extension of α-catenin (arrows). **e**, Quantification of α-catenin intensity versus instantaneous rise of the consensus reconstruction (Extended Data Fig. 10b). Vertical dashed line indicates canonical F-actin rise. **f**, Averaged subdomain displacements (scaled 15× for visualization) after MDFF analysis of the consensus map versus canonical F-actin (top; PDB: 8D13) and supercoil F-actin (bottom; MDFF model from Fig. 3c).

## Discussion

We find that myosin activity can generate supercoiled F-actin regions in vitro, which are also present at load-bearing cytoskeleton-adhesion interfaces in cells. Whereas micrometre-scale F-actin supercoils have previously been observed in modified gliding assays[61], to our knowledge the nanoscale rearrangements we report here have not previously been described, although we are aware of one report of oscillatory nanoscale curvature in F-actin under supraphysiological conditions of saturation myosin binding[62]. In this kinetic trapping cryo-EM study, photo-uncaging of ATP transiently induced filament curvature prior to myosin dissociation, probably through direct allosteric effects at actin–myosin interfaces as motors stochastically bound nucleotide. Here we show that myosin forces can produce supercoiling in filament regions that are distal from motor binding sites, generating a F-actin conformational mark for force-sensitive ABPs (Fig. 5). Our simulations suggest that cycles of myosin applying compressive force and unbinding can elicit this characteristic rearrangement due to the helical architecture of F-actin, regardless of motor directionality. Further work will be required to analyse the precise force regimes

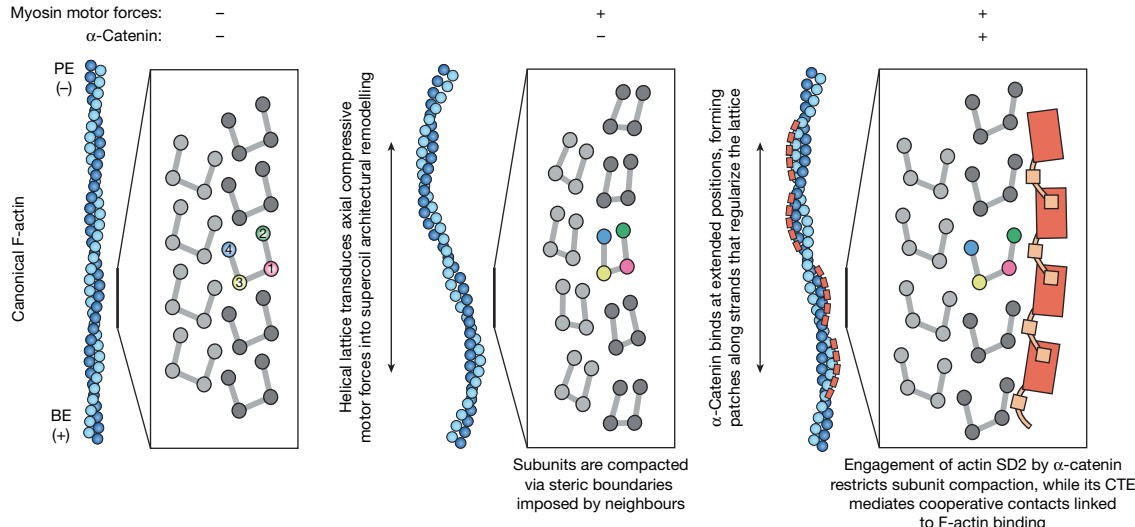

**Fig. 5 | Conceptual model of myosin force transduction through mechanosensitive F-actin binding.** Schematic showing generation of destabilized supercoil F-actin by myosin forces, which is detected and stabilized through cooperative binding contacts by α-catenin. CTE, C-terminal extension.

and timescales associated with generating supercoil F-actin, as well as whether its formation is reversible or invariably leads to filament rupture. Additionally, within cells, actin filaments are embedded in higher-order networks, which may restrict supercoiling. Myosin forces could nevertheless produce internal F-actin rearrangements that can be discriminated by ABPs, as we have described here, and visualizing mechanically active cytoskeletal networks in protein structural detail (which are suitable for cryo-ET[53]) is an important priority.

Although eukaryotic actins are highly conserved in sequence and structure, divergent prokaryotic actin homologues with diverse filament architectures have been reported[63,64], including a stable supercoil formed by *Bacillus thuringiensis* parM[65]. In eukaryotes, the latent potential of the actin fold for forming supercoiled assemblies may have been harnessed to transduce myosin forces into downstream biochemical processes through differential ABP engagement, concomitant with the unique appearance of cytoskeletal motors in this lineage[66]. Selection for the capacity to undergo such mechanically evoked transitions could provide one explanation for the extreme sequence conservation of eukaryotic actins, consistent with previous speculation[17].

Myosin force-evoked supercoil F-actin features specific lattice remodelling and actin protomer deformations that are distinct from those elicited by bending forces[48], which could facilitate the discrimination of these two force regimes by ABPs (Fig. 5). We find that the α-catenin ABD preferentially engages positions featuring extended rise, a hallmark of supercoil F-actin. This asymmetric binding further modulates the mechanically excited conformational landscape of F-actin, accentuating rise changes while regularizing twist. As supercoils appear to be associated with mechanical damage (Extended Data Figs. 2j and 4b), we speculate that α-catenin binding stabilizes F-actin in the presence of myosin forces, which could mediate contractility-enhanced adhesion–cytoskeleton coupling. While this work was under review, we found that cooperative α-catenin clusters along curved F-actin are stabilized by higher-order adhesion complexes[67], consistent with this concept.

Mechanistically, the F-actin-binding contacts of α-catenin mediate repositioning of actin subdomain 2. In addition to α-catenin providing stabilizing binding energy, this rearrangement can be explained by bound α-catenin modifying the space available for actin subunits to occupy as they are deformed by lattice rearrangements, consistent with a steric boundaries framework for mechanical regulation of F-actin[48]. Furthermore, inter-ABD cooperative binding interactions mediated by the C-terminal extension of α-catenin appear to support its accumulation along individual strands, rationalizing the requirement

of this region for force-activated actin binding[8]. Other force-sensitive ABPs feature tandem LIM domains with precise interdomain linker lengths[24,25], suggesting that they may also recognize an F-actin structural signature spanning multiple subunits. We speculate that multi-actin subunit engagement mechanisms confer sensitivity to the ångström-scale rearrangements of actin promoters and inter-subunit interfaces. These subtle transitions accumulate into substantial lattice architectural changes that can be more readily discriminated through binding contacts at the multi-subunit scale. Further structural characterization of force-activated F-actin complexes will be necessary to probe the generality of this concept, which should be accessible through the approaches that we introduce here. More broadly, direct visualization of these active mechanical signalling assemblies will advance our mechanistic understanding of force transduction and its malfunction in disease.

## Limitations of the study

Cryo-fluorescence data used to target zyxin-enriched subcellular areas for cryo-ET data acquisition are diffraction limited and thus do not feature sufficient resolution to assess whether zyxin is bound to specific actin filaments in tomograms. Future in vitro and cellular work, including subtomogram averaging studies, will be required to determine how LIM domain proteins such as zyxin detect force along F-actin and the influence of other ABPs including tropomyosin, which is bound to the majority of cellular F-actin[68]. Although we provide evidence that our low-resolution cryo-EM reconstructions are sufficient to resolve F-actin subunit repositioning and actin subdomain remodelling (Figs. 3b,c and 4f and Extended Data Figs. 7 and 10e), they do not contain atomistic details. Further studies will be required to visualize detailed structural rearrangements in supercoiled F-actin and their recognition by α-catenin. Additionally, our reconstruction of the α-catenin–F-actin complex in the presence of myosin forces does not delineate the sequence of events that mediates force-activated binding. This will require an approach for dynamically visualizing nanoscale F-actin subunit repositioning while α-catenin engages in the presence of force, which may reveal the sensitivity of α-catenin to additional force regimes and structural rearrangements.

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

## Methods

Unless otherwise noted, chemicals and buffer components were purchased from Sigma-Aldrich.

### Cell culture

PtK2 cells (ATCC, CCL-56) were cultured in MEM (Gibco) supplemented with 10% fetal bovine serum (FBS), 1 mM sodium pyruvate, 1× non-essential amino acids (Gibco), and 1× antibiotic–antimycotic (Gibco) at 37 °C in 5% $CO_2$. All cell lines were obtained from ATCC, which validates lines by STR profiling. We did not further validate them after purchase. Cells were monitored for mycoplasma contamination with the ATCC universal mycoplasma detection kit (a PCR-based assay) and tested negative. No commonly misidentified cell lines were used in this study.

### Cell line generation

For generation of a doxycycline-inducible F-tractin–mScarlet and zyxin–LDO–mNeongreen PtK2 cell line, HEK293T (ATCC, CRL-3216) cells were co-transfected with lentiviral plasmids and packing plasmids (PMD2.G and PsPAX2) using Lipofectamine 2000 (Thermo Fisher Scientific). Six hours post-transfection, the culture medium was refreshed. Forty-eight hours post-transfection, the culture medium containing lentivirus was collected and filtered. Lentivirus infection with polybrene (5 µg ml⁻¹) followed immediately after collection. Three types of lentivirus were added to PtK2 cells: F-tractin–mScarlet, zyxin–LDO–mNeongreen, and TetOn. Twenty-four hours post-infection, cells that stably integrated the lentiviral plasmids were selected with puromycin and blasticidin for one week.

### Cellular cryo-ET grid preparation

C-flat 1.2/1.3 holey carbon Au 200-mesh grids (Electron Microscopy Sciences) were sputter coated with 22 nm of carbon and baked in a dry oven at 60 °C overnight. Grids were plasma cleaned and coated with 10 µg ml⁻¹ fibronectin (Sigma, FC010) in DPBS for 1–3 h at 37 °C in a tissue culture incubator. Fibronectin-coated grids were washed with DPBS and stored in fresh PtK2 culture medium in a 35 mm tissue culture dish for immediate use.

Doxycycline-inducible F-tractin–mScarlet and zyxin–LDO–mNeongreen PtK2 cells were grown to 90% confluency before passaging. Cells were thoroughly trypsinized and filtered through a 0.45-µm filter to remove clumps. Thirty thousand cells were added to each 35 mm tissue culture dish containing the fibronectin-coated grids. Twenty-four hours later, the medium was replaced with fresh PtK2 medium containing 100 ng ml⁻¹ of doxycycline to induce expression of F-tractin–mScarlet and zyxin–LDO–mNeongreen. The next day, grids were prepared for plunge freezing. Grids for the majority of tomograms (presented in Fig. 1b and Extended Data Fig. 1c,d) were treated with 1 µg ml⁻¹ Rho Activator II (Cytoskeleton, CN03) for 3.5 h, while the grid for the tomogram presented in Extended Data Fig. 1b (tomogram 11 in Extended Data Fig. 1d) was untreated. The grids were then washed with DPBS and loaded onto a Leica EM-GP plunge freezer operating at 25 °C. Three microlitres of DPBS was added to the front of the grid, followed by blotting from the back using a Whatman no. 5 filter paper for 7 s, then flash frozen in liquid ethane.

### Cryo-fluorescence imaging

Grids were mounted on a Leica Cryo-CLEM microscope. Epifluorescence microscopy was performed using a 50× 0.9 NA air objective at −180 °C. z-Stacks of images were captured at a depth of 16 bit. Image acquisition was performed using LAS X software (Leica). Images were post-processed with THUNDER (Leica) to denoise, then maximum intensity projected.

### Cellular cryo-ET data collection

Data were collected on a spherical-aberration (Cs) corrected Titan Krios TEM (Thermo Fisher Scientific) operating at 300 kV and equipped with a K3 direct electron detector (Gatan) and a BioQuantum energy filter (Gatan) using a slit width of 20 eV. Micrographs were collected at a nominal magnification of 15,000×, which corresponds to a calibrated pixel size of 5.05 Å (2.525 Å in super-resolution mode). The tilt-series were collected using the FastTomo acquisition scheme[69] in SerialEM[70] using a target defocus of −5 µm underfocus. Tilt angles ranged from −54° to +54° in 3° increments, grouped in a dose-symmetric manner[71]. Each tilt in the tilt-series had a total acquisition time of 2.4 s and total dose of 2.82 e⁻ Å⁻², fractionated across 16 frames (0.18 e⁻ Å⁻² per frame).

### Cellular cryo-ET data processing

Individual tilts were motion-corrected and binned 2× (to a pixel size of 5.05 Å) using MotionCor2 (ref. 72), and contrast transfer function (CTF) parameters were estimated using CTFFIND4 (ref. 73). Tilt-series alignment was carried out using AreTomo2 (ref. 74), followed by reconstruction using IMOD[75] with back-projection at a binning of 2 (voxel size of 10.1 Å). The reconstructed tomograms were denoised using cryo-CARE[76], then the missing wedge was predicted and restored with IsoNet[77]. The tomograms were then segmented using a U-Net convolutional neural network with the Dragonfly software[78]. For the tomogram displayed in Extended Data Fig. 1b, the plasma membrane was segmented using MemBrain v2 (ref. 79). All membrane segmentations were smoothed using mean curvature motion[80]. Segmentations were visualized using ChimeraX[81].

### Cellular cryo-ET quantification

Cellular tomogram quantification was performed in a blinded fashion. D.Y.Z.P. manually assigned tomograms into low zyxin/F-actin and high zyxin/F-actin bins based on the absence or presence of local peaks of zyxin signal intensity relative to F-actin intensity in line scans of cryo-fluorescence micrographs (Extended Data Fig. 1d). Tomograms were then randomized before quantification by K.H. and G.M.A. Sinusoidal filament regions were manually annotated using 3DMOD based on the appearance of continuous stretches of oscillating curvature, followed by unblinding for analysis. With the exception of tomograms 2 and 10, both annotators consistently found similar numbers of sinusoidal filament regions across tomograms (Extended Data Fig. 1f). However, relatively few overlapping sinusoidal filament regions were selected. Therefore, after removing overlaps, the results for both annotators were combined for final quantification (presented in Fig. 1c). The total volume occupied by F-actin in each tomogram was estimated by Dragonfly segmentation as described above.

### Actin purification

Actin was prepared from chicken skeletal muscle as previously described[82]. All steps were performed at 4 °C unless otherwise indicated. In brief, 1 g of acetone powder was resuspended in 20 ml of G-Ca buffer (G buffer: 2 mM Tris-HCl pH 8, 0.5 mM dithiothreitol (DTT), 0.2 mM ATP, and 0.01% $NaN_3$, supplemented with 0.1 mM $CaCl_2$), then mixed by inversion for 30 min. The mixture was ultracentrifuged in a Beckman Ti70 rotor at 42,500 rpm (79,766 g) for 30 min. The supernatant, containing G-actin monomers, was collected, and 50 mM KCl and 2 mM $MgCl_2$ were added to induce F-actin polymerization for 1 h. 0.8 M KCl was then added and the mixture incubated for 30 min to dissociate contaminants from F-actin, followed by ultracentrifugation in a Ti70 rotor at 42,500 rpm (79,766 g) for 3 h. The pellet containing F-actin was collected and resuspended in 2 ml G-Ca buffer, then incubated overnight to depolymerize filaments. The mixture was then Dounce homogenized for 10-15 passes, sequentially sheared with 26 G and 30 G needles, then dialysed against G-Ca buffer overnight in Spectra/Por 1 dialysis tubing (6-8 kDa MWCO). The solution was then collected, sheared again with a 30 G needle, then dialysed against G-Ca buffer for an additional day. The solution, containing dissociated G-actin monomers, was collected and ultracentrifuged in a Beckman Ti90 rotor at 70,000 rpm (187,354 g) for 3 h. The upper two-thirds of the

supernatant was then loaded on to a Cytiva HiLoad 16/600 column for size-exclusion chromatography. Purified G-actin was maintained in G-Ca buffer at 4 °C.

## Protein expression and purification

Plasmids were propagated in NEB 5-alpha competent *Escherichia coli* cells and purified with either Qiaprep spin miniprep kits (Qiagen, for bacterial expression) or the PureYield plasmid maxiprep system (Promega, for transfection). Full-length human calmodulin was expressed in Rosetta2(DE3) *E. coli* cells and purified at 4 °C according to a published protocol[83]. The purified protein was collected in Storage Buffer (20 mM Tris-HCl pH 8.0, 100 mM NaCl, 2 mM β-mercaptoethanol, and 5% v/v glycerol), flash frozen in liquid nitrogen, and stored at −80 °C prior to use. For experiments presented in Fig. 1, Extended Data Fig. 2a–d, and Supplementary Videos 2–4, constructs for murine myosin Va (amino acids 1–1090) and human myosin VI (amino acids 1–1021) featuring C-terminal GFP and Flag tags were expressed and purified from Sf9 insect cells using the baculovirus system as previously described[84]. In the myosin VI construct, a GCN4 leucine zipper dimerization domain replaced the smooth muscle coiled-coil region originally used by Nishikawa et al.[85]. For all other experiments, equivalent constructs for human myosin Va (amino acids 1–1090) and myosin VI (amino acids 1–1021) were expressed from a modified pCAG vector featuring a C-terminal GFP and Flag tag in FreeStyle 293-F cells (ThermoFisher) and purified as described[24].

In brief, cells were cultured in FreeStyle expression medium (ThermoFisher) at 37 °C on an orbital shaker in the presence of 8% CO$_2$. Cells were transfected when the culture reached a density of $1.8 \times 10^6$ cells per ml. Per 400 ml of culture, 1.2 ml of 1 mg ml$^{-1}$ PEI MAX (PolySciences) was pre-mixed with 400 μg of plasmid in 15 ml of FreeStyle expression medium and incubated for 20 min at room temperature prior to transfection. Myosin Va and myosin VI were co-transfected with a modified pCAG vector containing untagged, full-length human calmodulin (amino acids 1–149) at a mass ratio of 1:6. Cells were cultured for an additional 60 h, then collected and snap frozen in liquid nitrogen. Cell pellets were stored at −80 °C.

For myosin purification, all steps were performed at 4 °C. Cells were resuspended in myosin lysis buffer (50 mM Tris-HCl pH 8.0, 150 mM NaCl, 2 mM MgCl$_2$, 0.2% 3-((3-cholamidopropyl) dimethylammonio) −1-propanesulfonate (CHAPS), 2 mM ATP, 1 mM phenylmethylsulfonyl fluoride (PMSF), 1 μg ml$^{-1}$ aprotinin, leupeptin and pepstatin) and incubated with rocking for 40 min. The lysate was clarified by centrifugation in a Beckman JA-25.50 rotor at 20,000$g$ for 30 min. The supernatant was collected and incubated with anti-Flag M2 affinity beads (Sigma-Aldrich) on a rocker for 1.5 h. The beads were collected and washed three times with myosin purification buffer (50 mM Tris-HCl pH 8.0, 150 mM NaCl, 2 mM MgCl$_2$ and 2 mM ATP), then eluted with myosin purification buffer supplemented with 100 μg ml$^{-1}$ Flag peptide (Sigma-Aldrich). The eluted protein was buffer exchanged into myosin storage buffer (10 mM Tris-HCl pH 8.0, 100 mM NaCl, 2 mM MgCl$_2$ and 3 mM DTT) using an Amicon Ultra-4 concentrator (50 kDa MWCO), then snap frozen in liquid nitrogen. Purified myosins were stored at −80 °C prior to use.

The C-terminal ABD of human α-catenin (amino acids 664–906) was expressed from a pET vector encoding an N-terminal 6 × His tag, strep tag, and TEV protease cleavage site as previously described[8]. Transformed Rosetta2(DE3) *E. coli* were cultured in LB medium at 37 °C to an optical density of 0.8–1.0, then induced with 0.7 mM isopropyl β-D-1-thiogalactopyranoside (IPTG) and cultured at 16 °C for 16 h. Cells were then collected and flash frozen in liquid nitrogen. Cell pellets were stored at −80 °C.

For purification, all steps were performed at 4 °C. Cells were resuspended in lysis buffer (50 mM Tris-HCl pH 8.0, 150 mM NaCl, 2 mM β-mercaptoethanol, 20 mM imidazole) and disrupted in an Avestin Emulsiflex C5 homogenizer. The lysate was clarified at 15,000$g$ in a Beckman JA-25.50 rotor for 30 min, then the supernatant was collected and incubated with Ni-NTA resin (Qiagen) with rocking for 1 h. The resin was collected, washed with 5 bed volumes of lysis buffer, then eluted with lysis buffer (50 mM Tris-HCl pH 8.0, 150 mM NaCl, 2 mM β-mercaptoethanol, 300 mM imidazole). Purified His-tagged TEV protease (prepared according to a published protocol[86]) was then added to reach a 0.05 mg ml$^{-1}$ working concentration, and the solution was dialysed against dialysis buffer (20 mM Tris-HCl pH 8.0, 300 mM NaCl, 2 mM β-mercaptoethanol) for 16 h. The sample was then collected and re-applied to Ni-NTA resin to remove TEV protease. The flowthrough was collected, then sequentially purified by anion exchange chromatography using a HiTrapQ HP column (Cytiva), followed by size-exclusion chromatography on a Superdex 200 Increase column (Cytiva) in gel filtration buffer (20 mM Tris-HCl pH 8.0, 100 mM NaCl, 2 mM β-mercaptoethanol). The protein was snap frozen in liquid nitrogen and stored at −80 °C until use.

## Force reconstitution on cryo-EM grids

All steps were performed at room temperature unless otherwise noted. F-actin was prepared from G-actin monomers in G-Mg (G buffer supplemented with 0.1 mM MgCl$_2$) plus KMEI buffer (50 mM KCl, 1 mM MgCl$_2$, 1 mM ethylene glycol-bis(β-aminoethyl ether)-*N,N,N′,N′*-tetraacetic acid (EGTA), 10 mM imidazole pH 7.0, 1 mM DTT) as previously described[45]. CF-1.2/1.3-3Au 200-mesh gold C-flat holey carbon cryo-TEM grids (EMS, CF213-50-Au) were used for initial cryo-EM imaging and tomography studies, while CF-1.2/1.3-3 Au 300-mesh grids (EMS, CF313-50-Au) were used for single-particle studies.

For the dual motor condition, the following solutions were prepared at 4 °C in motility buffer (MB, 20 mM 3-(*N*-morpholino)propanesulfonic acid (MOPS) pH 7.4, 5 mM MgCl$_2$, 0.1 mM EGTA, 1 mM DTT): MB-PVP, 0.1% polyvinylpyrrolidone (Sigma-Aldrich, PVP10); MB-anchor, 0.1 mg ml$^{-1}$ mouse monoclonal anti-GFP antibody (Sigma-Aldrich, G6539); MB-myosin, 0.02 μM myosin V and 0.08 μM myosin VI; MB-ATP, 1 mM ATP and 0.01% Nonidet P-40 Substitute (NP-40, Roche); MB-no-ATP, 0.01% NP-40. Solutions were clarified by ultracentrifugation at 50,000 rpm (108,726$g$) in a Beckman TLA-100 rotor, then brought to room temperature immediately prior to grid preparation.

Grids were untreated prior to sample preparation, as we found plasma cleaning and other treatments reduced the activity of myosins. First, 6 μl of MB-anchor was applied to each side of the grid and incubated for 2 min. Six microlitres of MB-myosin was then applied to each side of the grid and incubated for an additional 2 min. Twenty microlitres of MB-PVP was then applied to the top (shiny) side of the grid and incubated to block the surface for 1 min. Six microlitres of F-actin was then applied and incubated for 40 s. The grid was then washed in a 500 μl reservoir of MB in a microcentrifuge tube, then placed in the chamber of a Leica EM-GP plunge-freezing device operating at 25 °C and 100% humidity. Six microlitres of either MB-ATP (+force generation) or MB-no-ATP (no force generation) was then applied and incubated for 40 s. The grid was then blotted for 5 s and plunged into liquid ethane. For each grid preparation session, samples with F-actin concentrations ranging between 0.4–1.0 μM were prepared and screened, and specimens featuring optimal filament density were selected for further analysis.

For the barbed-end-directed and pointed-end-directed single-motor conditions, reagents were prepared as above, with the following modifications: MB-anchor was supplemented with 500 nM streptavidin (VWR, S000-01), and two separate myosin buffers were prepared, MB-myosin V (0.02 μM myosin V in MB) and MB-myosin VI (0.08 μM myosin VI in MB). To prepare 25% biotin−F-actin seeds biotinylated actin monomers (Cytoskeleton, AB07-C) were co-polymerized at a 1:4 molar ratio with unlabelled monomers with a total concentration of 10 μM for 1 h, then sheared through a 30G needle. F-actin was then extended by mixing seeds with unlabelled monomers at a 1:5 molar ratio and polymerizing for an additional 1 h. Samples were then prepared as

above, using either MB-myosin V (barbed-end-directed) or MB-myosin VI (minus-end-directed).

Samples for the force-activated α-catenin–F-actin complex were prepared identically as for the dual motor condition, with the following modifications. After mounting in the plunge-freezing apparatus, 3 µl of MB-ATP supplemented with α-catenin ABD was applied and incubated for 30 s, then the grid was blotted for 5 s and plunged into liquid ethane. Grids were prepared with varying F-actin/α-catenin ABD concentrations, and 0.6 µM F-actin/1 µM α-catenin ABD was found to be optimal after screening. All grids were stored in liquid nitrogen prior to cryo-EM data collection.

### Fluorescence imaging of myosin activity

For data presented in Supplementary Videos 2–4, samples were prepared as described above, except 10% rhodamine G-actin (Cytoskeleton, AR05-B) was included in all polymerization reactions (F-actin and biotinylated seeds). Instead of plunge freezing, grids were sandwiched between two No. 1.5 24 × 60 mm glass coverslips (Corning) in a drop of MB-ATP supplemented with oxygen scavengers (20 µg ml⁻¹ catalase, 100 µg ml⁻¹ glucose oxidase, 125 mg ml⁻¹ glucose, 50 mM DTT). Epifluorescence image sequences (movies) were collected at room temperature using a 60× 1.40 NA PlanApo oil-immersion objective lens (Nikon) on an inverted TE2000-E microscope (Nikon) equipped with a solid-state white light illumination system (Lumencor SOLA SE) controlled through the Metamorph software package (Molecular Devices). Images were collected on a Photometrics Coolsnap HQ2 CCD camera with a frame rate of 1 exposure every 2–3 s, using filters for visualizing rhodamine (excitation 650/25 nm, emission 607/36 nm). A single image (either bright-field to directly visualize the cryo-EM grid, or fluorescence using filters to visualize anchored, GFP-tagged myosins: excitation 485/20 nm, emission 525/30 nm) was taken before and after each image sequence to visualize the grid substrate and verify the lack of substantial drift.

### Low-magnification cryo-EM

For data and analysis presented in Fig. 1d–f and Extended Data Fig. 2a–d, samples were imaged on a Tecnai F20 transmission electron microscope operating at 120 kV using a Gatan 626 cryo-transfer holder, operated with the Leginon software package[87]. Exposures were collected at 29,000× magnification (corresponding to a calibrated pixel size of 3.2 Å per pixel) on a Gatan Ultrascan 4000 CCD camera with −4 µm underfocus. Images were low-pass filtered to 20 Å and binned by 4 for visualization and analysis. Initial quantification of sinusoidal filament region wavelengths (presented in Fig. 1f and Extended Data Fig. 2a) was performed manually with FIJI[88] by measuring the distance between peaks along segments with clear oscillating curvature.

### Filament annealing refractory condition

For data and analysis presented in Extended Data Fig. 2e,f, 50% PEG-biotin coated coverslips were prepared by adapting a previously described protocol[30]. No. 1.5H 24 × 60 mm glass coverslips (Marienfeld) were sequentially sonicated in 2% Hellmanex III (Hellma), acetone, and 1 M KOH. Coverslips were then sonicated in MilliQ water three times, followed by treatment with Nano-Strip 2X (Cyantek), then rinsed with MilliQ water. Coverslips were then coated in 0.5 mg ml⁻¹ mPEG silane MW 5 K (Laysan Bio) and 0.5 mg ml⁻¹ biotin–mPEG silane MW 5 K (Laysan Bio) in 96% ethanol and 9.6 mM HCl. PEG-coated coverslips were then washed in 100% ethanol, rinsed with MilliQ water, and dried before immediate use or storage at 4 °C.

For total internal reflection fluorescence (TIRF) microscopy experiments, 20% biotin-labelled F-actin seeds were prepared by co-polymerizing 0.6 µM of unlabelled purified G-actin with 0.2 µM biotin-labelled rabbit skeletal muscle G-actin (Cytoskeleton) and 0.2 µM ATTO-488-labelled rabbit skeletal muscle G-actin (Hypermol) in the presence of G-Mg + KMEI buffer at room temperature for 1 h.

The biotinylated F-actin solution was sheared 5 times with a 26G needle, then mixed with G-Mg + KMEI buffer containing 2.4 µM unlabelled G-actin, 0.8 µM biotin-labelled rabbit skeletal muscle G-actin (for anchoring filament extensions to glass for imaging), 0.8 µM rhodamine-labelled rabbit skeletal muscle G-actin (Cytoskeleton) and 100 µM profilin (purified as previously described[30,89]) at a 1:19 volume:volume ratio, then incubated at room temperature for an additional 1 h. Polymerization was subsequently halted by the addition of 0.3 µM capping protein (CapZ, Hypermol), producing capped F-actin extended from biotinylated seeds.

Imaging chambers were prepared by attaching CultureWell reusable PDMS gaskets (Grace Bio-Labs) to PEG-biotin coated coverslips. Each well was treated with the following sequence: incubation with 500 nM of streptavidin (Rockland) in MB supplemented to 3 mM DTT for 4 min, blocking with Blocking Buffer (MB-PVP supplemented with 1 mg ml⁻¹ κ-casein (Sigma) and 1 mg ml⁻¹ bovine serum albumin (BSA, Gemini)) for 1 min, incubation with capped F-actin extended from biotinylated seeds for 2 min, washing with MB, then immersion in MB + oxygen scavenger mixture (described above) for TIRF imaging.

Images were acquired at room temperature on a Nikon H-TIRF system through a CFI Apo 100× TIRF oil-immersion objective (NA 1.49), a quad filter (Chroma), and an iXon EMCCD camera (Andor) with the Perfect Focus System (Nikon) engaged. Illumination was provided by 488 and 561 nm laser lines (Agilent) switched by an acousto-optic tunable filter. Images were captured at a depth of 16-bit using the NIS-Elements software (Nikon).

### Optimized pointed-end force grids

To prepare cryo-EM specimens of the pointed-end-directed force condition using capped F-actin extended from biotinylated seeds (Extended Data Fig. 2g,h), we implemented an improved procedure by using an anti-GFP nanobody conjugated to fibrinogen (adapted from the protocol of Watson et al.[90]) to anchor myosin VI to the surface. This enabled us to maintain myosin activity after plasma cleaning the grids, improving sample quality.

To prepare the fibrinogen-coupled nanobody, we expressed and purified anti-GFP nanobody[91] LaG 94-10 and chemically conjugated it to fibrinogen (MilliporeSigma, 341576). Periplasmic expression of the nanobody from the pET-21b vector was induced in BL21(DE3) E. coli cells at 0.6 OD using 0.1 mM IPTG, which were collected after overnight culture at 12 °C, flash frozen in liquid nitrogen, and stored at −80 °C. Cell pellets were resuspended in TES buffer (0.2 M Tris-HCl pH 7.5, 0.5 mM ethylenediaminetetraacetic acid (EDTA), 0.5 M sucrose), then lysed by osmotic shock on ice for 20 min through the addition of a 1:4 volume:volume ratio mixture of TES and MilliQ water in 1.5 fold excess over the original resuspension volume. Protoplasts were separated from the periplasmic fraction by centrifugation at 4,000g for 10 min at 4 °C. The supernatant was collected, then centrifuged at 48,000g for 30 min at 4 °C to remove insoluble material present in the periplasm lysate[92].

The soluble fraction was then mixed with an equal volume of Nanobody Wash Buffer (20 mM sodium phosphate pH 8.0, 1 M NaCl, 20 mM imidazole) and incubated with Ni-NTA resin for 20 min at 4 °C. The resin was washed with 25 column volumes of Nanobody Wash Buffer before elution with Nanobody Elution Buffer (20 mM sodium phosphate pH 8.0, 0.5 M NaCl, 0.3 M imidazole). The eluted nanobody was buffer exchanged into fibrinogen buffer (0.1 M sodium bicarbonate pH 8.3, 0.5 mM EDTA) supplemented with 1 mM tris(2-carboxyethyl) phosphine (TCEP) and concentrated to 15 mg ml⁻¹ using a 3 kDa MWCO centrifugal filter (Amicon).

Fibrinogen was resuspended in fibrinogen buffer, then reacted with 25-fold molar excess of maleimide-PEG8-succinimidyl ester (Millipore Sigma, 746207) while rocking at room temperature for 1 h. Maleimide coupled fibrinogen was then separated from unreacted maleimide-PEG8-succinimidyl ester by two rounds of precipitation

with 25% mass/volume ammonium sulfate while rocking at room temperature for 30 min, centrifugation at 17,000g for 10 min, and resuspension in fibrinogen buffer. Maleimide coupled fibrinogen was then reacted with 5-fold excess of freshly purified anti-GFP nanobody overnight at 4 °C. The coupling reaction was quenched by the addition of 50 mM L-cysteine and incubation at room temperature for 30 min, followed by two additional rounds of ammonium sulfate precipitation to isolate the nanobody-fibrinogen conjugate. The conjugate was reconstituted to a final concentration of 5 mg ml[−1] in fibrinogen buffer, flash frozen in liquid nitrogen, and stored at −80 °C.

For cryo-EM sample preparation, reagents were prepared as described above for reconstituting myosin activity on grids, with the following modifications: MB-anchor was replaced with 0.5 mg ml[−1] anti-GFP nanobody-fibrinogen conjugate and 0.5 µM streptavidin prepared in MB, and the concentration of myosin VI in MB-myosin VI was increased to 0.5 µM. Capped F-actin extended from biotinylated seeds was prepared as described above while omitting both fluorescently labelled G-actins (which were substituted by equivalent amounts of unlabelled purified G-actin). Biotinylated actin was also omitted when filaments were extended from seeds. For sample preparation, grids were plasma cleaned using a Gatan Solarus in the presence of $H_2$ and $O_2$ for 10 s, then placed in CultureWell PDMS gaskets affixed to untreated coverslips. Grids were then treated sequentially with 30 µl of each solution as described above for preparing single-motor samples. For the no-ATP control, 200 µl of capped F-actin extended from biotinylated seeds was pre-mixed with 0.2 µl of apyrase (NEB M0398S), and specimens were incubated at room temperature for an additional 5 min upon addition of this F-actin to the grid. Prior to freezing, 3 µl of MB-ATP (+ATP condition) or MB-no-ATP (−ATP, +apyrase condition) was applied. Grids were incubated for 40 s, then blotted for 6 s and plunged into liquid ethane.

### Single-particle cryo-EM data acquisition

For the dual motor condition and filament annealing refractory pointed-end-directed force condition, cryo-EM data were recorded on a Titan Krios (ThermoFisher/FEI) operated at 300 kV equipped with a Gatan K3 camera, BioQuantum energy filter and spherical-aberration (Cs) corrector. SerialEM[70] was used for automated data collection. Movies were collected at a magnification of 64,000× in super-resolution mode resulting in a calibrated pixel size of 1.08 Å per pixel (super-resolution pixel size of 0.54 Å per pixel). For the dual motor condition, a defocus range of −1.5 to −3.5 µm underfocus was used. Sixty-three frames were recorded over 2.5 s of exposure at a dose rate of 27.6 e[−] pixel[−1] s[−1] (23.7 e[−] Å[−2] s[−1]) for a cumulative dose of 59.2 e[−] Å[−2]. Three datasets were collected with these imaging conditions consisting of 12,453 movies, 10,842 movies and 8,504 movies, respectively. The first dataset was used for preliminary processing, and all three datasets were pooled to generate the final reconstruction. A single no-ATP control dataset was collected under the same imaging conditions, consisting of 3,749 movies. For the filament annealing refractory pointed-end-directed force condition, a defocus range of −0.8 to −2.5 µm underfocus was used. Forty frames were recorded over 2 s of exposure at a dose rate of 30 e[−] pixel[−1] s[−1] (25.7 e[−] Å[−2] s[−1]) for a cumulative dose of 51.4 e[−] Å[−2]. 188 movies were analysed from the +ATP sample and 110 movies were analysed from the −ATP, +apyrase control.

For the force-activated α-catenin−F-actin complex, cryo-EM data were recorded on a Titan Krios (ThermoFisher/FEI) operated at 300 kV equipped with a Gatan K2 Summit camera. SerialEM[70] was used for automated data collection. Movies were collected at a nominal magnification of 22,500× in super-resolution mode, resulting in a calibrated pixel size of 1.33 Å per pixel (super-resolution pixel size of 0.665 Å per pixel), over a defocus range of −1.5 to −3.5 µm underfocus. Fifty frames were recorded over 10 s of exposure at a dose rate of 10 e[−] pixel[−1] s[−1] (e[−] Å[−2] s[−1]) for a cumulative dose of 56.5 e[−] Å[−2]. Three datasets were collected with these imaging conditions consisting of 1,909 movies, 1,792

movies and 754 movies, respectively. These datasets were pooled prior to processing.

### Micrograph pre-processing

Movies were aligned with MotionCor2 using 5 × 5 patches[72], and dose-weighting sums[93] were generated from twofold binned frames with Fourier cropping, resulting in a pixel size of 1.08 Å in the images collected in the dual motor condition datasets (±ATP) and filament annealing refractory pointed-end force datasets (+ATP or −ATP, +apyrase) and 1.33 Å for the force-activated α-catenin−F-actin complex datasets. Non-dose-weighted sums were used for CTF parameter estimation using CTFFIND4 (ref. 73).

### Two-dimensional quantification of cryo-EM images

To manually annotate micrographs, we implemented a custom Napari[94] plugin. For analysis of the dual motor condition (presented in Extended Data Fig. 2b), data were binned to a pixel size of 12.82 Å per pixel. For analysis of the pointed-end-directed force condition prepared with the filament annealing refractory protocol, data were binned to a pixel size of 4.32 Å per pixel. Micrographs were manually annotated by tracing individual filaments within holes using a stylus; each annotated micrograph was saved as an mrcs stack with a separate channel for each filament. This enabled each filament trace to span the entire observed filament length without discontinuities at intersections with other filament traces.

Filament traces were converted to spline representations by thresholding and skeletonizing each mrcs slice. Filament traces with a contour length of at least 325 nm were retained for analysis. To minimize the effects of long-range filament curvature and to quantify filament properties at the same length scale as the in vitro tomographic analysis (Fig. 2c,d), filament traces were analysed using a 300 nm sliding windows sampled every 25 nm. For each 300 nm-long window, principal component decomposition was performed: PC1 established the filament region's central axis, and PC2 corresponded to deviation of the filament's centre from this central axis. Peaks and troughs along the PC1/2 decomposition were identified, and the largest difference in the PC2 value between neighbouring peaks and troughs was recorded as the filament region's peak-to-peak amplitude. The highest amplitude score along a filament was retained as well as the average absolute curvature of the filament within that window. For the dual motor data, 912 filaments from 100 micrographs were traced for the +ATP condition, of which 857 filaments were longer than 325 nm. For the no-ATP condition, 745 filaments from 46 micrographs were traced, of which 712 filaments were longer than 325 nm. For the filament annealing refractory pointed-end-directed force data, 396 filaments from 188 micrographs were traced for the +ATP condition, of which 312 filaments were longer than 325 nm. For the −ATP, +apyrase condition, 414 filaments from 110 micrographs were traced, of which 320 filaments were longer than 325 nm.

### Cryo-ET of reconstituted specimens

Grids were clipped and transferred to Talos Arctica TEM (ThermoFisher) operating at 200 kV with a Gatan K2 Summit camera. Tomograms were collected using a dose-symmetric scheme from −60° to +60° with 3° angular increments. Each tilt image was fractionated into 10 frames with a total exposure dose of 2.4 e[−] Å[−2] (0.24 e[−] Å[−2] per frame) and a total exposure time of 1 s. The defocus range was between −1.5 µm and −5 µm underfocus. The acquisition magnification was 17,000× with a pixel size of 2.4 Å. Frames were aligned with MotionCor2 (ref. 72).

Tomograms were reconstructed using the Appion-Protomo software[95,96]. Images were aligned and dose-weighted, then reconstructed using the simultaneous iterative reconstruction technique (SIRT) as implemented in Tomo3D[97]. Reconstructions were binned by 4 resulting in a final pixel size of 9.6 Å.

## Synthetic cryo-ET data generation

In vitro tomograms were processed using a denoising autoencoder (DAE) approach to enable direct, confident tracing of F-actin filaments in 3D. A DAE with a similar architecture to that reported in our previous work[53] was trained. In brief, synthetic datasets approximating subtomograms were first generated. Twenty-four thousand synthetic noiseless subtomograms were prepared which contained between zero and five in silico actin filament models (PDB: 7R8V) of varying curvature. All of these filament models were bent in a single plane (to prevent supercoil model bias), then converted to volumetric data using the pdb2mrc procedure in EMAN2 (ref. 98). Each actin filament was rotated about the rot and psi angles by random, uniformly sampled values between 0° and 359°, while the tilt rotation angle was sampled from a Gaussian probability distribution centred at 90° with a standard deviation of 20°. Each actin filament was translated in the box along each dimension by random values uniformly sampled within the range of ±196 Å.

Matched synthetic noisy volumes were generated by first projecting the sum of the oriented filaments at integer angular increments (held constant for a particular subtomogram) uniformly sampled between 1° and 5°, inclusive. The maximum tilt was sampled from a random integer between 45° and 60°, yielding tilt angle ranges from ±45° to ±60°. These projections were corrupted by a CTF, using parameters matching those of the microscope and a defocus range of −3 μm to −11 μm underfocus, then reconstructed back into a 3D volume using reconstructor class functions as implemented in the EMAN2 python package[98]. Empirical noise was modelled by extracting 1,000 subtomograms of 96 × 96 ×96 voxels from each experimental tomogram and computing the per tomogram average Fourier transform. For each synthetic volume, one of these empirical noise boxes was randomly selected, multiplied element-wise by a white noise box of the same dimensions, normalized, and scaled by a random scale factor that modulated the signal-to-noise ratio of each synthetic subtomogram. The synthetic volume was then summed with its noise volume in Fourier space. To account for interpolation artefacts from CTF application or noise addition in Fourier space, the synthetic volumes were then cropped to 48 voxels in real space. After these operations were complete, each noisy particle was paired with its corresponding noiseless ground truth particle.

## Denoising in vitro tomograms

A DAE featuring adversarial training was constructed and trained using an approach we have recently reported[53]. Pre-training of this DAE consisted of 3D convolutional layers in a U-net architecture, and it was performed using a single NVIDIA A100 GPU with 80 GB of VRAM, using a learning rate of 0.00001. Training was run on the 24,000 pairs of synthetic noisy and noiseless volumes, with a 90:10 training:validation split, for 30 epochs. This model had a cross-correlation coefficient validation loss of 0.9147. After pre-training, an additional network head was added for domain classification by forking the network output after the feature extraction layers. The domain classification head consists of a gradient reversal layer and additional 3D convolutional layers, which are followed by flattening to a dense layer, then a binary classification layer with a sigmoid activation function. 1,000 synthetic volumes and 1,000 volumes extracted from the experimental tomograms were used as the training set for the domain classifier. Adversarial training was performed by alternatively passing these data through the domain classifier head, followed by re-training the feature extractors with the denoising head using only the 1,000 synthetic volumes. This adversarial training was run for ten iterations, and the final iteration of adversarial training was used for denoising.

The adversarial-trained neural network was then used to denoise empirical tomograms, using A100 GPUs. 48-voxel tiles sampled every 16 voxels in each dimension were extracted, normalized, and passed as inputs to the neural network. The network outputs were stitched together via maximum intensity projection.

## Quantification of in vitro tomograms

Denoising the in vitro tomograms enabled 3D contour tracing of F-actin. Denoised tomograms were visualized in UCSF Chimera[99], and each filament's central axis was manually traced using Chimera's marker set tool. Univariate splines were fit through the data with a fixed smoothing factor of 150 to minimize jitter from manual picking. The spline fits were resampled evenly with 9.6 Å steps and visually inspected to ensure trace quality.

The cartesian coordinates of these traces were subsequently analysed. The instantaneous curvature and torsion along each filament trace was inspected to identify filament segments with potential oscillatory character. Filament segments with high, oscillating curvature and torsion that were between 350 nm and 500 nm in length were identified and selected. PCA was performed on these selected filament segments, and the decompositions were aligned such that the maximum amplitude along the second principal component was at the centre point and the second derivative of the second principal component at this point was positive. If a filament's principal component decomposition was inverted from the other traces, the starting end of the filament spline was flipped. One limitation of this analysis is that filament polarity could not be assigned in the denoised tomograms. The aligned traces were analysed within a common 350 nm window to produce average traces, as well as peak-to-peak amplitude and wavelength measurements (Fig. 2c,d).

To assess whether the supercoil filament segments exhibited non-uniform orientation relative to the ice layer, the second principal component's angle relative to the ice layer was analysed. The tomograms contained protein aggregates at the air–water interface, which appeared as amorphous density in the denoised tomograms. Chimera's marker set tool was used to position centroids within these densities. PCA was then performed on these points, where the first and second principal components defined the ice plane. The angle between the second principal component of the oscillatory region (that is, the major axis of each supercoil's elliptical cross-section) and the ice plane was measured by projecting the ice plane's normal vector along the first principal component of the oscillatory domain and computing the angle between these vectors. A Kolmogorov–Smirnov test as implemented in Scipy[100] was performed to assess whether the observed major axis angle distribution was significantly different from a normal distribution, and no significant difference was found ($P = 0.27$).

## Coarse-grained molecular dynamics

**Parameterization.** Coarse-grained molecular dynamics simulations of individual actin filaments under force were performed using the software package ESPResSo[101] and custom Python scripts. Each subunit was mapped to its centroid based on the structure of canonical F-actin[48], which consists of two right-handed strands intertwining to form a left-handed helix with a rise of 27.8 Å and a twist of −166.67°. The filament was modelled as a 3D spring network, with each subunit (positioned 1.6 nm radially from the filament axis) connected to its four nearest neighbours (Extended Data Fig. 3a). Implicit solvent effects were simulated using the Langevin thermostat. The filament's energetics were described by a harmonic bond potential (equation 1), harmonic angle potential (equation 2), and dihedral potential (equation 3):

$$V(l) = \frac{1}{2}k_l(l - l_0)^2 \qquad (1)$$

Where $k_l$ is the harmonic bond stiffness, and $l_0$ is the diagonal or longitudinal bond length in equilibrium configuration.

$$V(\theta) = \frac{1}{2}k_\theta(\theta - \theta_0)^2 \qquad (2)$$

Where $k_\theta$ is the harmonic angle bending constant, and $\theta_0$ is the angle formed by neighbouring triplets in equilibrium configuration. Each particle is associated with six different harmonic angles.

$$V(\phi) = k_\phi[1 - \cos(n\phi - \phi_0)] \qquad (3)$$

Where $k_\phi$ is the dihedral angle bending constant, $\phi_0$ is the dihedral angle formed by neighbouring quadruplets in equilibrium configuration, and $n$ is the multiplicity of the potential (number of minima). Each particle is associated with two different dihedral angles; $n$ was set to 1 for simplicity.

The geometric parameters of the equilibrium configuration, $l_0$, $\theta_0$ and $\phi_0$ were derived from the canonical actin filament structure[48]. For simplicity, deformations of individual subunits were ignored. The effects of actin nucleotide state transitions on filament mechanics were also neglected.

The stiffness and bending constants, $k_l$, $k_\theta$ and $k_\phi$, have not been experimentally measured; nonetheless, they can be calibrated through matching to previously reported data, as demonstrated by Yogurtcu et al.[102]. Moreover, Schramm et al. previously developed a mesoscale model to investigate actin and cofilactin filament buckling and fragmentation under mechanical strain[103]. Their approach began with all-atom molecular dynamics simulations, from which they constructed elastic network models by mapping each actin subunit to its centre of mass and connecting it to four neighbours via elastic bonds. Bond stiffnesses were iteratively optimized to reproduce molecular dynamics-derived fluctuations and then incorporated into a mesoscale model, where each subunit was approximated as an ellipsoid. Harmonic bonds were randomly distributed across subunit interfaces based on buried solvent-accessible surface area.

By contrast, our model describes filament energetics using harmonic bond, angle, and dihedral potentials. Each interaction interface is represented by a single bond. Interaction potentials were parameterized sequentially and iteratively until convergence, as detailed below.

The harmonic bond potential captures F-actin's tensile rigidity and was calibrated to match a force-extension relationship previously reported by Chu and Voth[104]. Using coarse-grained molecular dynamics simulations, they probed the force-extension behaviour of actin filaments, reporting stretch stiffness values of 37 pN nm$^{-1}$ for ATP F-actin and 31 pN nm$^{-1}$ for ADP F-actin[104]. To calibrate $k_l$, parameter scans were performed by simulating an actin filament composed of 39 subunits (the same number of subunits modelled by Chu and Voth) under tension. Simulations were conducted with a low friction coefficient at extremely low temperature (0.298 K) to minimize stochastic deviations. The stretch stiffness was extracted by plotting the pulling force against the filament extension at equilibrium (Extended Data Fig. 3b). The target value for calibration was 31 pN nm$^{-1}$, corresponding to ADP F-actin to match our cryo-EM sample conditions.

The harmonic angle potential reflects F-actin's bending rigidity and was optimized to reproduce the 9 µm persistence length of ADP F-actin[57]. To calibrate $k_\theta$, parameter scans were performed by simulating thermal fluctuations of an actin filament composed of 100 subunits using the friction coefficient of water at room temperature. To compute the persistence length, a centre axis approximating a linear polymer was constructed as follows: for each subunit $i$, an anchor point on the opposite strand was calculated as the midpoint between subunits $i-1$ and $i+1$. The midpoint between subunit $i$ and this anchor, denoted $c_i$, was used to trace the filament's centre axis. The persistence length $P$ was determined using the relation:

$$\ln\langle\cos\alpha_{i,\,i+n}\rangle = -\frac{L_n}{P} \qquad (4)$$

where $\alpha_{i,\,i+n}$ is the angle between the segments $(c_i - c_{i-1})$ and $(c_{i+n} - c_{i+n-1})$, and $L_n$ is the contour length defined as the sum of segment lengths $|c_{i+1} - c_i|$ from $c_i$ to $c_{i+n}$. The persistence length was extracted from the slope of the linear fit (Extended Data Fig. 3c).

The dihedral potential captures F-actin's twisting rigidity and was optimized to reproduce the dependence of cumulative twist variance on subunit index, as established from cryo-EM data by Bibeau et al.[105]. To calibrate $k_\phi$, parameter scans were performed by simulating thermal fluctuations of an actin filament under the same conditions used for calibrating $k_\theta$. Subunit twist was calculated using the centre axis defined as described above. For each pair of consecutive subunits $i-1$ and $i$, their positions were projected onto the segment $(c_i - c_{i-1})$ along the centre axis, and the twist angle was determined from the resulting projection vectors. At each time point, the deviation of each subunit's twist from the canonical F-actin value was computed. A zero-mean normal distribution was then fitted to the distribution of these deviations across the simulation for each subunit, and the variance was extracted. The twist variance was plotted against subunit index (Extended Fig. 3d). To avoid edge effects, the first and last 20 subunits were excluded, resulting in the same number of analysed subunits ($n = 60$) as in the study by Bibeau et al.[105].

**Unit derivations.** ESPResSO does not predefine units. Instead, the user defines the fundamental units of length and energy, from which all other units can be derived.

In our simulations, we set the length unit as:

$$[l] = 1\text{ nm}$$

We simulate the system at room temperature (298 K), so the energy unit is defined as:

$$[E] = k_B \cdot 298 \text{ K} = 4.12 \times 10^{-21}\text{ J}$$

By default, ESPResSO also has no specification of particle mass. We thus define the mass unit using the molecular mass of G-actin ($M = 42$ kg mol$^{-1}$) and Avogadro's constant ($N_A = 6.02 \times 10^{23}$ mol$^{-1}$):

$$[m] = \frac{M}{N_A} = \frac{42\text{ kg mol}^{-1}}{6.02 \times 10^{23}\text{ mol}^{-1}} = 6.97 \times 10^{-23}\text{ kg}$$

Using the kinetic energy relation $E = \frac{1}{2}mv^2 = \frac{1}{2}m\left(\frac{l}{t}\right)^2$, the time unit is derived as:

$$[t] = [l] \cdot \sqrt{\frac{[m]}{2[E]}} = 1 \times 10^{-9}\text{ m} \times \sqrt{\frac{6.97 \times 10^{-23}\text{ kg}}{2 \times 4.12 \times 10^{-21}\text{ J}}} = 1 \times 10^{-10}\text{ s} = 0.1\text{ ns}$$

Based on the relation $F = \frac{E}{l}$, the force unit is derived as:

$$[F] = \frac{4.12 \times 10^{-21}\text{ J}}{1 \times 10^{-9}\text{ m}} = 4.12 \times 10^{-12}\text{ N} \approx 4\text{ pN}$$

Therefore, in our simulations, a force value of 1.5 corresponds to a physical force of approximately 6 pN. All final parameters in both simulation units and physical units are reported in Extended Data Table 2.

**Force perturbations.** The dynamics of a 400-subunit actin filament were then simulated. Thermal fluctuations were simulated at room temperature with the friction coefficient of water. The filament force response was then simulated by fixing the initial five subunits ($i = 1:5$) and applying compressive or tensile force on the terminal five subunits ($i = 396:400$), with their orientations constrained to prevent rotation.

To mimic the mechanical fragmentation behaviour of a physical filament, filament ruptures were implemented by enabling bond breaking when the length of a diagonal or longitudinal bond exceeded defined critical values. These critical lengths were estimated based on the

helical parameters measured from the cryo-EM density map of super-coil F-actin (Fig. 3a,b), using the following equations:

$$l_{diag} = \sqrt{2r^2(1 - \cos|\beta|) + h^2} \tag{5}$$

$$l_{long} = 2\sqrt{(r\sin|\beta|)^2 + h^2} \tag{6}$$

Where $l_{diag}$ and $l_{long}$ are the diagonal and longitudinal bond length, respectively; $h$ is the helical rise, $\beta$ is the helical twist, and $r$ is the distance from the subunit to the filament's centre axis.

Based on the measured ranges $h \in [2.5\ nm,\ 3.2\ nm]$ and $\beta \in [-177°, -154°]$ (Fig. 3b), the maximum diagonal bond length is 4.5247 nm (at $h = 3.2$ nm and $\beta = -177°$), corresponding to a 7.2% increase over the canonical value of 4.2226 nm. The maximum longitudinal bond length is 6.5519 nm (at $h = 3.2$ nm and $\beta = -154°$), corresponding to a 16.8% increase over the canonical value of 5.6087 nm.

Bond breaking was treated as irreversible; thus, subunit reincorporation and re-annealing of filament fragments were not permitted.

Actin filaments under tension or compression were simulated at a range of forces (10 pN, 20 pN, 30 pN and 40 pN total force) to investigate the effect of force magnitude on the resulting supercoil structures. For each replicate ($n = 5$) of each condition, a random seed was selected, then an initial simulation was performed under constant force with bond breaking enabled (Extended Data Fig. 4b), and the simulation step at which the filament ruptured was recorded. To maintain filament integrity for supercoil parameter analysis, a second simulation (with bond breaking disabled) was performed where force was applied until the previously observed rupture simulation step, after which force was released, followed by 5,000 additional simulation steps. Analyses for the 'constant force' (Fig. 2e) conditions were performed on simulation frame 0 through the frame preceding force release (Fig. 2g and Extended Data Fig. 3e,f), and analyses for the 'apply force and release' conditions (Extended Data Fig. 4c) were performed on all frames following force release (Fig. 2h and Extended Data Fig. 3e,f).

To mimic the asynchronous activity of myosin motors, a stochastic motor-firing scheme was implemented by randomly selecting five subunits from the lower half of the filament ($i \in [200, 400]$), with each subjected to a 6 pN pulling or pushing force for a randomly assigned duration (Extended Data Fig. 4d). Bond breaking was disabled in these simulations.

Actin filaments under torque were simulated by fixing the initial two subunits ($i = 1,2$) and applying either under-twisting or over-twisting torque to the terminal two subunits ($i = 399,400$). The rotation axis was defined by two midpoints: one between the last ($i = 400$) and second-to-last ($i = 399$) subunits, and the other between the second-to-last ($i = 399$) and third-to-last ($i = 398$) subunits. Vectors for torque application (levers) were calculated by projecting the positions of the last two subunits onto the rotation axis, and the direction of the applied force was determined by the cross product of the lever and the rotation axis. Forces of 80 pN and 800 pN were applied to each of the terminal two subunits ($i = 399,400$) to probe the effects of torque magnitude. Simulations were performed both with and without bond breaking enabled (Extended Data Fig. 4e).

To investigate the effects of helical parameters on supercoil architecture, simulations were performed with a 200-subunit filament (to maintain reasonable computation times) where F-actin's helical rise and twist were systematically varied around their canonical values. Separately, to examine the influence of flexural rigidity, the harmonic angle bending constants were adjusted to produce a range of persistence lengths, calculated using the following equation:

$$P = \frac{B_s}{k_B T} \tag{7}$$

Where $P$ is persistence length, $B_s$ is flexural rigidity, $k_B$ is the Boltzmann constant, and $T$ is absolute temperature (n.b. persistence length scales linearly with flexural rigidity, rendering it a suitable proxy). In both cases, simulations were performed by fixing the initial five subunits ($i = 1:5$) and applying a 6 pN compressive force to each of the terminal five subunits ($i = 196:200$), with their orientations constrained to prevent rotation. Bond breaking was disabled in these simulations.

All mechanical perturbations and thermal fluctuations were simulated using the friction coefficient of water at room temperature.

**Supercoil analysis.** To compute the amplitude of a supercoil, the filament's centre axis was traced at each time point as described above. Analogous to our quantification of sinusoidal filament regions in cryo-EM images, a sliding window with a contour length of 300 nm was applied along the axis using a 25 nm step size. Within each window, PCA was performed on the segment, and PC2 was plotted against PC1. Peaks and troughs were identified, and the maximum difference along PC2 between neighbouring peaks and troughs was recorded for each window. The largest such difference across all windows of a given simulation frame was used to determine the instantaneous supercoil peak-to-peak amplitude for that frame.

To identify actin filaments that adopted supercoil configurations, a peak-to-peak amplitude cut-off of 16 nm was applied. For filaments meeting this criterion, the supercoil pitch was determined using the distance along PC1 between the neighbouring peak and trough corresponding to the recorded amplitude; the pitch was calculated as twice this distance.

To enable comparison with experimental measurements (Fig. 1f and Extended Data Fig. 2a), lower and upper pitch cut-offs of 75 nm and 350 nm, respectively, were applied when analysing thermal fluctuations, and constant force and force and release conditions (Fig. 2e–h and Extended Data Fig. 3e,f). For simulations exploring the effects of helical parameters and flexural rigidity (Extended Data Fig. 3g,h), pitch analysis was conducted on all frames until the filament reached a 50% reduction in end-to-end length, with no cut-offs applied.

Eccentricity (Extended Data Fig. 3i) was calculated from ellipses fitted to the PC3 versus PC2 projections of the maximal peak-to-peak amplitude window of each frame as described above.

All analyses were performed using trajectories from five independent simulations per condition ($n = 5$).

## Neural network particle picker training

To analyse single-particle data, a DAE featuring a previously described architecture[48] was trained to learn features of actin filaments in cryo-EM projection images (Extended Data Fig. 5a). In brief, the DAE consisted of an encoder and decoder. The encoder was composed of nine convolutional layers followed by three dense, fully connected layers of decreasing size. The decoder was composed of two dense, fully connected layers of increasing size, which then connected to nine convolutional layers, the last of which produced the denoised image.

Synthetic projection images to train the DAE and an accompanying semantic segmentation network were generated as outlined previously[48]. To improve network performance for semantic segmentation of the experimental data presented in this study, two modifications were made: incorporation of known background picks, and improvement of the noise model. Our data collection scheme resulted in a substantial amount of carbon in the micrographs. To prevent the semantic segmentation network from picking the edges of holes or actin filaments over carbon, 15,208 manual picks of hole edges and thick carbon areas were selected from micrographs in the dataset using RELION[106]. These picks were extracted at a box size of 512 pixels, then binned by 4 to a box size of 128 pixels and a pixel size of 4.32 Å per pixel. These picks were then integrated into the training dataset of the fully convolutional neural network for semantic segmentation, paired with targets composed entirely of background with no signal.

In our previously reported neural network training schemes, EMAN2's pink noise generation function[98] was used to produce realistic-looking projection images. However, this noise model proved insufficient to pick on the current datasets, possibly due to ice thickness or different microscope or detector parameters. To improve the approximation of the synthetic data to real micrographs, an empirical pink noise model was used. The synthetic 2D particle image generation model can be summarized by the following equation:

$$X_i = C_i \sum_{k=0}^{K} P(\phi_k) V_k + \eta_i \tag{8}$$

The following procedure was used to generate a particle image $X_i$: first, a synthetic volume was randomly chosen from a library of 135 bent actin filaments $V_k$, and the pose of the filament was transformed by a 3D rotation and translation before it was projected $P(\phi_k)$. This process was repeated for each filament in the image ($K$, up to four filaments per particle image), and the results were summed in real space to produce a noiseless projection image. Then, the image was corrupted by a CTF with the sampled defocus range, $C_s$, amplitude contrast, and voltage matching those used for CTF estimation. Finally, noise was added in Fourier space $\eta_i$.

The noise model, $\eta$, was empirically determined by using RELION to manually pick and extract 2,177 empty regions within holes in the micrographs to yield empty noise boxes with pixel sizes of 4.32 Å per pixel and box sizes of 128 pixels. The average and standard deviation of the Fourier transform of each empty pick was computed to model the empirical noise distribution. For each synthetic particle image, a white noise box was generated, and the Fourier transform of the white noise box was multiplied element-wise by the sum of the empirical noise average and a Gaussian sampling of the empirical noise standard deviation. This pink noise box, $\eta_i$, was added to the Fourier transform of the image after CTF corruption, and the inverse Fourier transform was computed to yield the final noisy projection image.

To produce the semantic segmentation target, the noiseless projection image was low-pass filtered to 40 Å, binarized, and eroded by 8 pixels.

To train the DAE, 800,000 image pairs with a 90:10 training:validation split and a learning rate of 0.00005 were used to train the network. Cross-correlation coefficient was used as the loss function. The network was trained until validation loss did not improve for three epochs; then the best network weights were restored and saved. The validation cross-correlation coefficient of the trained DAE was 0.9887.

The trained convolutional layers of the DAE were then used to initialize the weights of the initial layers of a fully convolutional neural network for semantic segmentation, featuring a previously described architecture[48]. In brief, the trained convolutional encoding layers of the DAE were copied as a separate neural network, and additional convolutional layers were added to form a fully convolutional neural network for semantic segmentation[107]. The final layer of this network consisted of two channels with sigmoid activation and default initializations. To train the network for semantic segmentation, 75,000 image pairs (60,000 of which contained synthetic projections and 15,000 of which were carbon picks, as described below) with a 90:10 training/validation split and a learning rate of 0.001 were used to train the network. Binary cross-entropy (BCE) was used as the loss function. The network was trained until validation loss did not improve for three epochs; then the best network weights were restored and saved. The validation BCE of the trained DAE was 0.0444. The architectures described above have since been superseded by a U-net architecture, which we found produces better segmentation loss with a smaller training set in a shorter time[108].

## Particle picking with trained network
A custom Python script was used to pass images to a fully convolutional neural network for semantic segmentation and execute curvature-sensitive filament picking, modified from our previously described method[48]. In brief, each micrograph was binned by 4 to a pixel size of 4.32 Å per pixel, then 128-pixel tiles featuring 32 pixels of overlap were extracted and passed as inputs to the network. The outputs were stitched together by maximum intensity projection at the overlaps, producing a semantic segmentation map of the micrograph. These maps were then binarized using a fixed, empirically determined threshold of 0.3 and skeletonized. An additional step of dilation by 8 pixels after the skeletonization was performed to link filament ends, which we found were often disjointed in these datasets. Another round of skeletonization was then performed. Branches shorter than 8 pixels were pruned, and pixels within a radius of 16 pixels from filament intersections were removed. Continuous filaments were then identified by matching tracks with common end points, and 2D splines were fit through the filaments for curvature estimation. To prevent spuriously high curvature values due to edge effects, the terminal 50 pixels of the spline were omitted from picking. From the remaining filament sections, the instantaneous curvature was measured along the spline at three-pixel intervals and used for segment selection. This resulted in substantial filament segment overlap during initial picking; these overlapping segments were retained for initial alignment before subsequent duplicate removal. Segments from the same filament were flagged in the output metadata (a RELION-formatted STAR file).

To be assigned as a supercoil segment, substantial curvature in both positive and negative directions along the same filament were required. An identified filament segment was considered a supercoil if it contained a segment at least 200 Å in length (approximately the length of 7 subunit rises) where the instantaneous curvature was always greater than or equal to 1.5 $\mu m^{-1}$, as well as another segment of the same length where the instantaneous curvature was always less than or equal to −1.5 $\mu m^{-1}$ (Extended Data Fig. 5b). These criteria exclude straight filaments and filaments with substantial uniplanar curving in one direction. All filament segments assigned as supercoils were retained for subsequent image processing.

## Modelling of curvature distributions
The curvature distribution of free, unloaded F-actin would be expected to follow a Boltzmann distribution set by the F-actin's persistence length and the filament segment length, as we modelled in our previous study[48]. In this scheme, the energy to bend F-actin is defined as:

$$E = \frac{1}{2} k_B T L_p L \kappa^2 \tag{9}$$

Where is the $k_B$ is the Boltzmann constant, $T$ is absolute temperature, $L_p$ is persistence length in microns, $L$ is the filament segment length in micrometres, and $\kappa$ is curvature in $\mu m^{-1}$ (refs. 48,58). This corresponds to the probability function defined by:

$$P_{adjusted}(\kappa) = \frac{1}{Z_{adjusted}} e^{\frac{\alpha L_p L \kappa^2}{2}} \tag{10}$$

$$Z_{adjusted} = \sum_{k=0}^{\infty} e^{\frac{\alpha L_p L \kappa^2}{2}} \tag{11}$$

Where $\alpha$ is a multiplicative adjustment factor that serves as a proxy for differing persistence length in this simple approximation. $\alpha$ was both fixed at a value of one for a basic modelling (Extended Data Fig. 5c, grey curves) and fit to the experimental curvature distributions (Extended Data Fig. 5c, pink and blue curves).

## Supercoil F-actin reconstruction
For initial model generation, data from a single imaging session were used. Selected supercoil segments were extracted in RELION (v3.1.2)

with a box size of 512 × 512 pixels and pixel size of 2.16 Å per pixel (bin 2; Extended Data Fig. 5d,e). To avoid reference bias, these segments were imported to cryoSPARC[109] (v3.2.0) for successive rounds of ab initio initial model generation and subsequent 3D classification. The initial 148k picked segments were used for ab initio model generation with three classes as implemented in cryoSPARC. The subset of filament segments contributing to the best class were then selected and used for three more rounds of ab initio model generation with two classes; after each round only the segments belonging to the best class were retained. The reconstructions from the final round of ab initio model generation were then used as references for a two-class heterogeneous refinement in cryoSPARC, with the initial, unaligned 148k picked segments used as inputs. This heterogeneous refinement produced one class with clear protomer definition and one class with apparently mixed filament polarity. The segments from the good class were retained for subsequent homogeneous refinement in cryoSPARC and 3D auto-refinement in RELION. Focused 3D classification without segment alignment was then performed using three classes. Segments from a single class with poor protomer definition were rejected, and the remaining segments were refined using local 3D auto-refinement in RELION to produce the initial model for processing the full dataset.

For processing of the full dataset of 276k filament segments from three datasets, selected supercoil segments were extracted as described above. These segments were imported to cryoSPARC for 2D classification. After removing junk classes and exceptionally straight filament classes, 173k segments were retained. These segments were passed to a cryoSPARC homogeneous refinement for initial alignment. The previously obtained 3D reconstruction was low-pass filtered to 30 Å and used as the initial model. After exporting a RELION STAR file containing the cryoSPARC alignment parameters, the original filament tube IDs were restored using a custom python script. A second custom python script implementing per-filament non-maximum suppression was used to remove overlapping segments from the same filament that were closer than 768 pixels at bin 1 (approximately 830 Å), yielding 13,146 filament segments. These segments were re-imported to cryoSPARC and a non-uniform refinement[110] was performed to improve local resolution near the filament edges using a mask encompassing the filament segment without overlap (75% $z$-length). 3DVA[111] was also performed in cryoSPARC using default parameters except for the filter resolution, which was set to 8 Å.

Two independent asymmetric reconstructions from the no-ATP control dataset were generated using a similar approach (Extended Data Fig. 6). To obtain these reconstructions, 345k segments from the no-ATP dataset that did not meet the supercoil criteria were extracted in RELION (v3.1.2) with a box size of 256 × 256 pixels and pixel size of 4.32 Å per pixel (bin 4). These segments were imported to cryoSPARC for 2D classification. After removing junk, 315k segments remained, and they were then split into two equally sized subsets. Each subset was then subjected to the following processing steps. First, ab initio reconstruction of one class was performed, followed by homogeneous refinement. The aligned segments were then re-extracted in RELION at a box and pixel size matching that of the supercoil reconstruction. Overlapping segments within 830 Å were then removed, resulting in 3,436 segments in the first subset and 3,137 segments in the second subset. 3D auto-refinement with local angular searches was performed in RELION. These aligned segments were then imported to cryoSPARC for a final non-uniform refinement using a 75% $z$-length soft mask.

## Force-activated α-catenin complex

For the three pooled α-catenin datasets, particle picking was largely similar to the supercoil dataset (Extended Data Fig. 8). In brief, the micrographs were binned to a pixel size of 4.32 Å per pixel (bin 3.25) and picked using the same selection criteria to identify filaments that contained long stretches of high curvature in both directions. In total, 133,677 segments with no overlap within 3 Å were picked, then extracted in RELION with a box size of 384 × 384 pixels and pixel size of 2.66 Å (bin 2). These extracted segments were subjected to initial 2D classification in RELION to remove junk picks. In order to align the psi (in-plane rotation) angle to a common reference, the remaining 110k segments were subjected to 3D classification in RELION using the initial supercoil model low-pass filtered to 30 Å as a reference and an angular sampling interval of 3.7°. The resulting reconstruction deviated substantially from the reference map. Therefore, the Euler angles were reset such that the rot and tilt angles were removed, the tiltPrior was set to 90°, and the psi angle was set as a prior, while translations were retained. 3D classification was performed with global rot search, local tilt and psi searches with a 12° angular search range, and bimodal priors on the psi angle.

After segments were aligned into this one class, subsequent 3D classification was performed with three classes using a fine 1.8° angular sampling interval, a global search for rot, and a 9° angular search range for tilt and psi. This yielded one bare class, which was rejected from further processing, as well as two classes with α-catenin decorating one side of the filament. To retain the maximum number of segments, the aligned segments from both decorated classes were subjected to focused 3D classification with no alignment, using a mask for α-catenin binding sites along one side of the filament. This resulted in one decorated class and one undecorated class. The decorated class was passed to a 3D auto-refinement job, and the undecorated class was subjected to another round of focused 3D classification with no alignment using a mask for α-catenin binding sites along the other side of the filament. This resulted in one undecorated class and one decorated class. The decorated class was retained and subjected to 3D auto-refinement. Upon inspection, the two refined maps were indistinguishable after a shift and rotation of one short-pitch helical step. The constituent segments of one of the classes were therefore rotated and translated by one short-pitch helical step, then combined with the segments from the other class. Overlapping segments within 768 Å were removed and refined in cryoSPARC using non-uniform refinement.

The aligned segments from the refinement were then subjected to 3DVA as implemented in cryoSPARC[111]. In order to detect large changes, a filter resolution of 20 Å was used. Large changes in filament curvature and α-catenin occupancy were observed along the first variability component. Filament segments from the 0th to the 25th percentile, featuring high α-catenin occupancy, were selected and refined in RELION to produce an asymmetric reconstruction.

## Helical parameter measurements

Measurements were performed using our previously described approach[48]. In brief, canonical F-actin protomers (PDB: 8D13) were rigid-body fitted into each reconstructed map and combined into a single model, and three copies of the model were stitched together to minimize edge artefacts. Only the central protomers from a single model were reported. A 3D spline is fit to the filament axis of each model. Rise is measured by computing the path length along the central axis between neighbouring protomers. Twist is determined using the Frenet-Serret frame of reference defined by the orthonormal basis of the unit tangent, unit normal, and unit binormal vectors along the length of the spline. The frame is then rotated along the normal-binormal plane. Twist is measured along the short-pitch helix between consecutive protomers. The models analysed in this work exhibited substantial lattice deformations that required minor adjustments to reliably measure the filaments' helical parameters. The measurement approach was updated to account for a variable radius of the filaments. During fitting of the filament axis spline, the radius was allowed to vary along the filament length and was fit using a separate univariate spline; this tended to suppress spurious twist deviations. Additionally, the modest resolution of the reconstructed maps, particularly at the filament edges, limited our confidence in protomer fitting. Consequently, for the consensus supercoil F-actin reconstruction, the no-ATP control reconstructions, and the 3DVA-sorted α-catenin-bound F-actin map, three independent

manual rigid-body fittings were performed to provide confidence intervals for the helical parameter measurements.

## Actin subdomain rearrangements

First, 25 copies of a canonical protomer model (PDB: 8D13) were rigid-body fit into each map. These 25 protomers were combined into a single PDB model for each map, and these combined models were subjected to Phenix geometry minimization with default parameters of 500 maximum iterations and 5 macro cycles in order to remove clashes. These minimized PDB models were then input to ISOLDE[112] and hydrogen atoms were added. Secondary structure distance restraints were imposed on each actin subunit for the following residue ranges: 7–35, 35–72, 72–147, 340–377, 147–183, 272–340 and 183–272, based on previously defined subdomain boundaries[113]. Torsional and distance restrains were then imposed on the entire secondary structure of each protomer. ISOLDE simulations were run with a weight of $30 \times 1,000$ kJ mol$^{-1}$ (map units)$^{-1}$ Å$^3$. The temperature was set to 100 K for 10 min of simulation, then slowly decreased to 0 K. Subdomain movement magnitudes and directions were determined as previously described[48]. In brief, protomers were aligned to a reference protomer model (PDB: 8D13) and the displacement vectors between $C_\alpha$s of the flexibly fit protomers and the reference protomer were computed. Finally, the average of the displacement vectors was calculated for each subdomain defined above.

## Molecular graphics and data analysis

Molecular graphics were prepared with ChimeraX[81]. Unless otherwise noted, plotting and statistical analysis was performed with GraphPad Prism. Python codes were prepared with the assistance of ChatGPT 4.0.

## Reporting summary

Further information on research design is available in the Nature Portfolio Reporting Summary linked to this article.

## Data availability

Cryo-EM density maps have been deposited in the Electron Microscopy Data Bank (EMDB) with the following accession codes: myosin force-evoked supercoil F-actin (EMD-46426); control myosin tethered F-actin no-ATP 1 (EMD-46427); control myosin tethered F-actin no-ATP 2 (EMD-46428); consensus force-activated α-catenin–F-actin complex (EMD-46429); and 3DVA-sorted force-activated α-catenin–F-actin complex (EMD-46431). Additional raw and processed data files are available at Zenodo (https://doi.org/10.5281/zenodo.18022755, https://doi.org/10.5281/zenodo.12702199 and https://doi.org/10.5281/zenodo.12702799 (refs. 114–116)). Cellular tomograms, segmentations and filament traces are available at https://doi.org/10.5281/zenodo.18022754 (ref. 114). Trained neural networks and denoised in vitro tomograms with filament traces are available at https://doi.org/10.5281/zenodo.12702199 (ref. 115). Trained neural networks used for single-particle analysis particle picking, flexibly fit PDB models, and variability analysis maps and models used for data analysis are available at https://doi.org/10.5281/zenodo.12702799 (ref. 116). Source data are provided for Figs. 1–4 and Extended Data Figs. 1, 2, 6 and 10. Additional source data are available at Zenodo (https://doi.org/10.5281/zenodo.17932417 and https://doi.org/10.5281/zenodo.17958929 (refs. 117,118)) for simulations (Fig. 2f–h and Extended Data Fig. 3; https://doi.org/10.5281/zenodo.17932417 (ref. 117)) and Extended Data Fig. 5 (https://doi.org/10.5281/zenodo.17958929 (ref. 118)). All reagents and resources reported in this study are freely available from the corresponding author. Source data are provided with this paper.

## Code availability

Custom code reported in this study is available at https://github.com/alushinlab/squiggle as open source.

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

**Acknowledgements** We thank M. Rout for the gift of the anti-GFP nanobody LaG 94–10; J. Hinshaw for use of the F20 electron microscope; C. Waterman for use of the epifluorescence microscope; and J. Sotiris, H. Ng and M. Ebrahim for assistance with cryo-fluorescence, cryo-ET and cryo-EM data collection. A.G.C. was supported by NIH T32 GM115327 and D.Y.Z.P. was supported by NIH/NHLBI fellowship F31HL165906. P.S.G. was supported by a Rockefeller University Women in Science Postdoctoral Fellowship, and K.H. was supported by a Rockefeller University Anderson Center for Cancer Research Postdoctoral Fellowship. A.J.N. was supported by NIH/NIGMS fellowship F32GM128303. This work was funded by grants from the NIH (R01GM141044 and 5DP5OD017885), the Alfred P. Sloan Foundation (G-2020-14047) and the Pew Charitable Trusts to G.M.A. and NIH grant HL004229 to J.R.S. This research was also supported by the Stavros Niarchos Foundation (SNF) as part of its grant to the SNF Institute for Global Infectious Disease Research at the Rockefeller University.

**Author contributions** D.Y.Z.P. and A.G.C. prepared and imaged cellular cryo-ET specimens, which were computationally processed and quantified by K.H. and G.M.A. P.S.G. initially developed the myosin reconstitution assay for cryo-EM studies with assistance from Y.T. and J.R.S. J.W.W. developed and prepared the anti-GFP nanobody coupled to fibrinogen. A.G.C., P.S.G., D.Y.Z.P. and X.S. prepared and imaged myosin reconstitution specimens, and L.M. prepared α-catenin specimens, which were imaged by A.G.C. and X.S. M.J.R. and G.M.A. performed quantification of sinusoidal filament regions using computational tools developed by M.J.R. A.G.C. and M.J.R. performed single-particle cryo-EM analysis and data interpretation, using computational tools developed by M.J.R. X.S. performed coarse-grained molecular dynamics simulations. A.G.C. performed in vitro cryo-ET studies with assistance from A.J.N., which were analysed by M.J.R. A.G.C., M.J.R., X.S. and G.M.A. analysed data and wrote the paper with input from all authors. G.M.A. supervised the study and conceived of the project.

**Competing interests** The authors declare no competing interests.

**Additional information**
**Correspondence and requests for materials** should be addressed to Gregory M. Alushin.

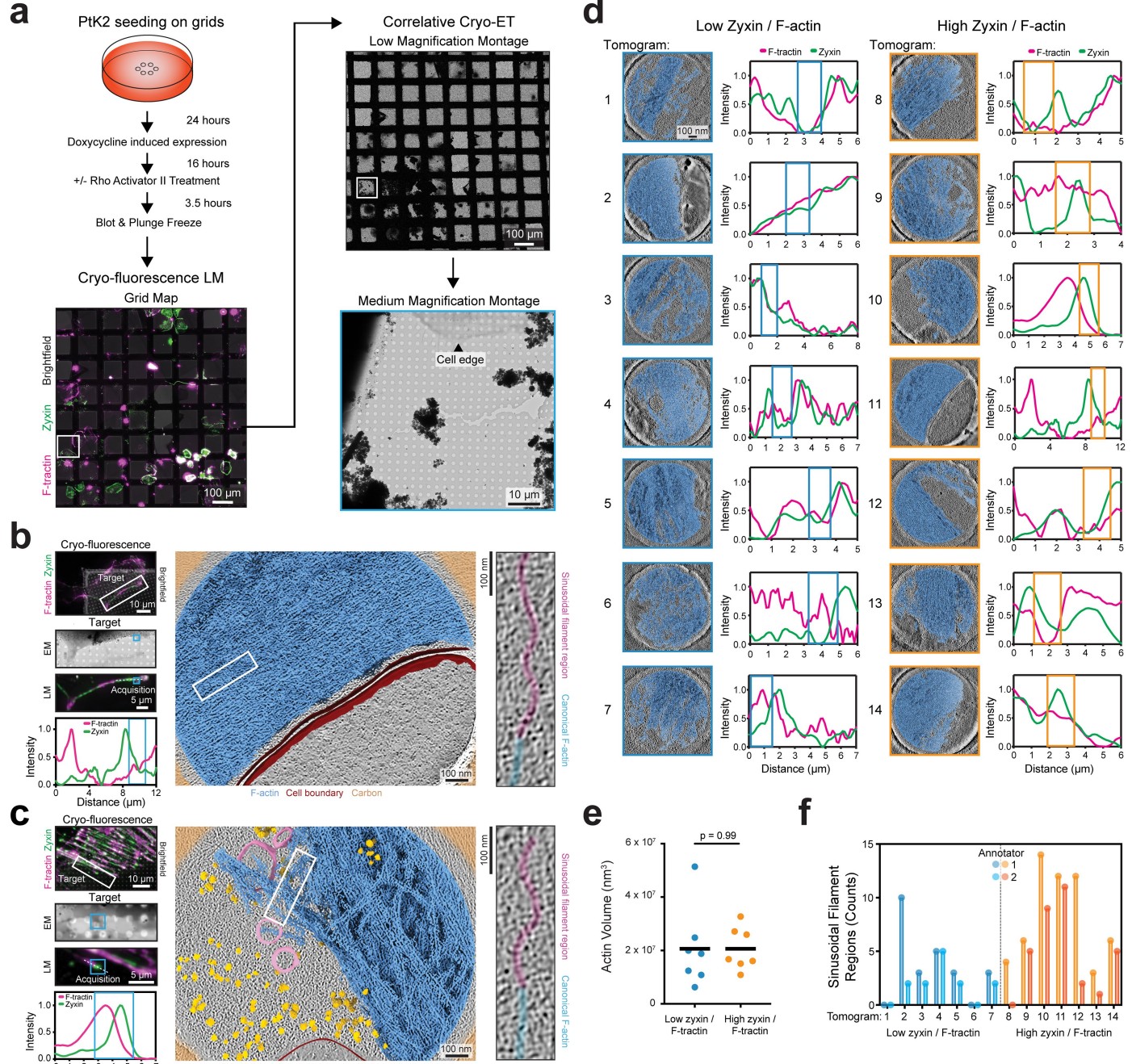

**Extended Data Fig. 1 | Correlative cryo-fluorescence and cryo-electron tomography. a**, Schematic of sample preparation and data collection workflow. **b and c**, Additional examples of sinusoidal filament regions imaged in zyxin-enriched adhesion sites from N = 2 independent experiments. The site in **b** features parallel bundled F-actin and was visualized in a cell not treated with Rho Activator II, while the site in **c** features more disorganized F-actin and was visualized in a cell treated with Rho Activator II. Correlative LM/EM and tomograms/segmentations are presented as in Fig. 1a,b. **d**, F-actin

segmentations (left) and fluorescence intensity line scans (right) of all quantified tomograms, binned by low versus high zyxin/F-actin assignment. Tomograms 11 and 10 correspond to those presented in b and c, while tomogram 9 is presented in Fig. 1b. **e**, Quantification of total F-actin captured by tomogram segmentations, which is indistinguishable between low (n = 7) and high (n = 7) zyxin/F-actin tomograms. Bars represent means; conditions were compared by two-tailed unpaired t-test. **f**, Quantification of sinusoidal filament regions across tomograms by each blinded annotator.

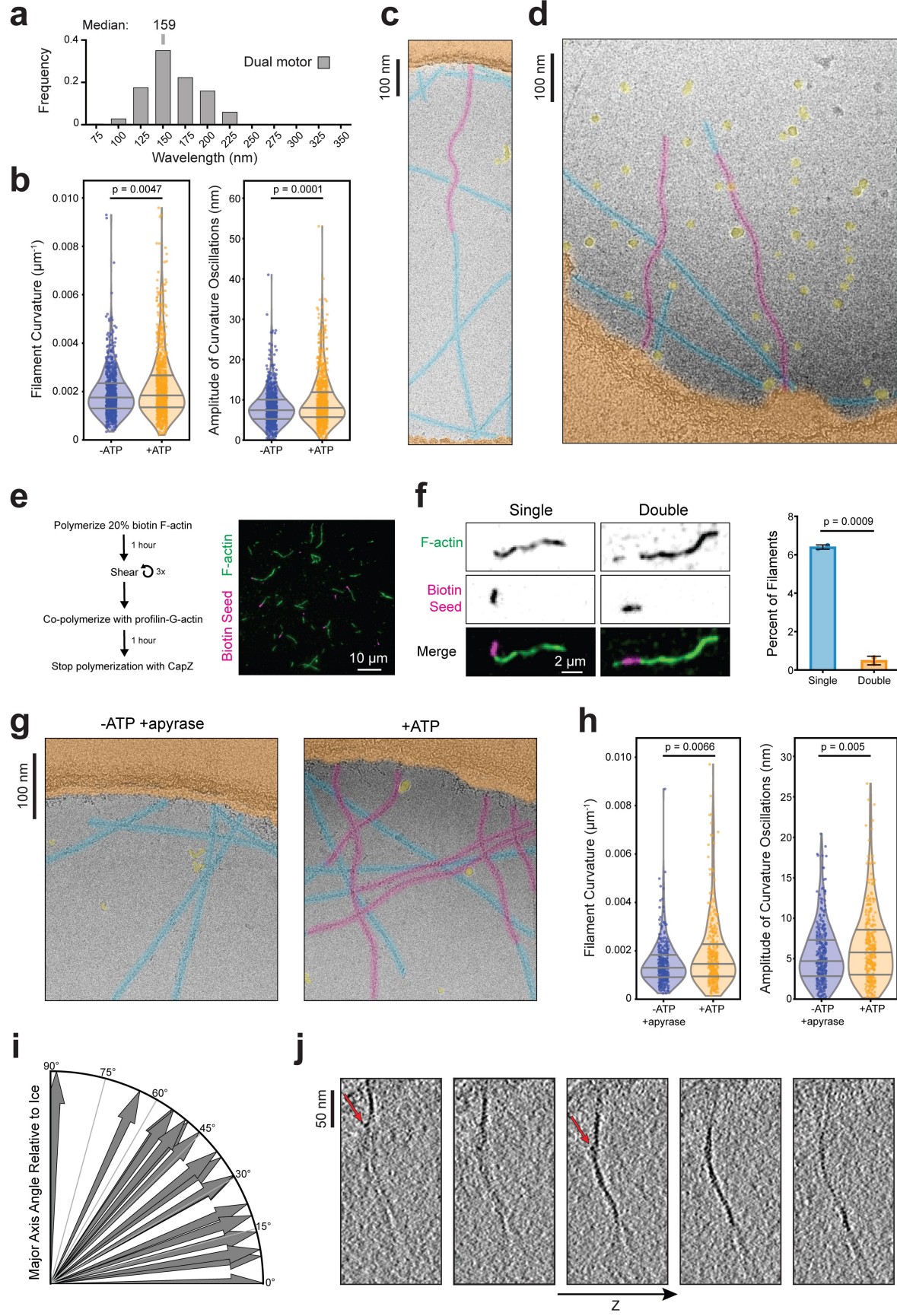

**Extended Data Fig. 2** | See next page for caption.

**Extended Data Fig. 2 | Analysis of F-actin sinusoidal region morphology.**
**a**, Quantification of sinusoidal region wavelengths in the dual motor condition; n = 251 from N = 3 independent experiments. **b**, Quantification of filament curvature (left) and peak-to-peak amplitude of curvature oscillations (right) in the dual motor condition in the absence (n = 712) and presence (n = 857) of ATP. Bars represent 25th, 50th, and 75th percentiles; conditions were compared by two-sided Mann Whitney U-test. **c** and **d**, False-colored cryo-EM images of the dual motor condition in the presence of ATP, displaying sinusoidal regions in a hole-spanning filament (**c**) and in a pair of broken filaments (**d**). Sinusoidal regions, magenta; canonical F-actin, blue; carbon film, orange; ice contamination, yellow. **e**, Preparation workflow (left) and TIRF image (right) of F-actin polymerization from biotinylated seeds in conditions refractory to filament annealing. **f**, Left: detail view of biotinylated seeds featuring F-actin extensions from one ("single") or two ("double") ends. Right: quantification of single versus double extensions (n = 7,466 filaments) across N = 2 trials. Bars indicate means ± s.d.; conditions were compared by two-tailed unpaired t-test. **g**, False-colored cryo-EM images of the pointed-end directed force condition using filaments prepared as in e. Specimens were prepared in the presence of either apyrase (to remove nucleotide, left) or ATP (right). **h**, Quantification of filament curvature (left) and peak-to-peak amplitude of curvature oscillations (right) in data from g, in the presence of apyrase (n = 320) or ATP (n = 312). Bars represent 25th, 50th, and 75th percentiles; conditions were compared by two-sided Mann Whitney U-test. **i**, Polar arrow plot of supercoil major oscillation axis (PC2) orientation relative to the ice plane from n = 16 tomographic observations, where 0° corresponds to the major axis being parallel to the ice plane. **j**, Serial Z slices through a supercoil F-actin tomogram (barbed-end directed force condition) featuring protruding densities consistent with subunit dislocations (red arrows).

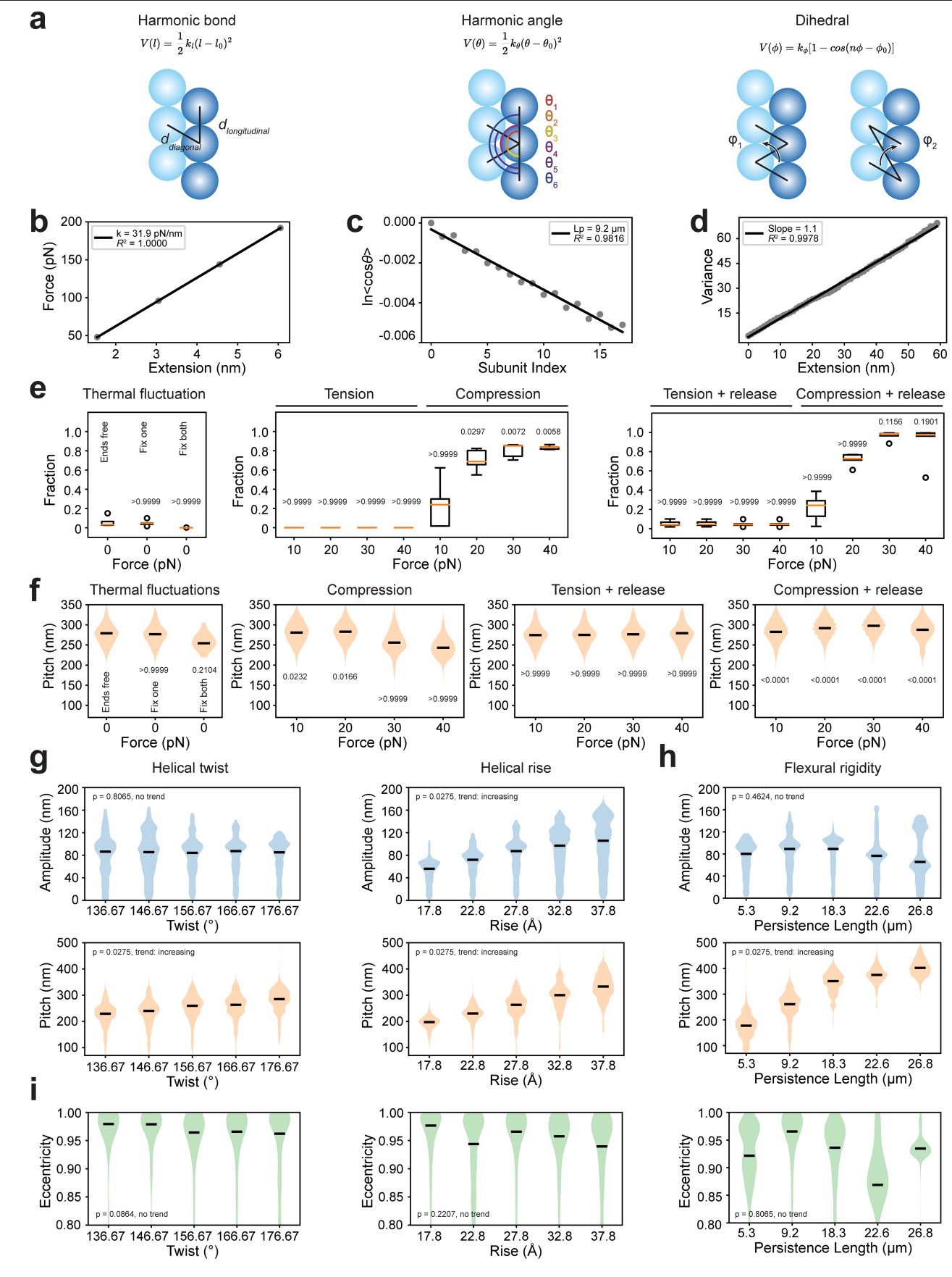

**Extended Data Fig. 3** | See next page for caption.

**Extended Data Fig. 3 | Calibration and quantification of coarse-grained molecular dynamics simulations. a**, Schematic of potentials used in coarse-grained molecular dynamics (MD) simulations. **b**, Calibration of the F-actin force-extension curve to determine the harmonic bond constants. **c**, Calibration of the F-actin persistence length to determine the harmonic angle potential constants. **d**, Calibration of cumulative twist variance to determine the dihedral potential constants. **e**–**f**, Quantification of additional parameters from simulations displayed in Fig. 2f–h (N = 5 simulation runs per condition). **e**, Quantification of the fraction of simulation frames containing a supercoil using a 3D peak-to-peak amplitude cutoff of 16 nm, a minimum pitch cutoff of 75 nm, and a maximum pitch cutoff of 350 nm. Bars represent medians, and lower and upper bounds of boxes represent 25th and 75th percentile, respectively. Whiskers indicate range, while circles represent outliers. P values are displayed for comparisons by Friedman test followed by post hoc Dunn's multiple comparisons test, versus controls as described in Fig. 2f–h; N = 5. **f**, Quantification of supercoil pitch of evenly sampled simulation frames selected using the cutoffs described in e. Bars represent medians. P values are displayed for comparisons by Kruskall-Wallis test followed by post hoc Dunn's multiple comparisons test versus controls as described in Fig. 2f–h. For thermal fluctuations, p > 0.9999 and p = 0.2104 for fixing one and both ends compared with ends free, respectively. For constant compression, p = 0.0232, p = 0.0166, p > 0.9999, and p > 0.9999 for 10 pN, 20 pN, 30 pN, and 40 pN force, respectively. For release after tension, p > 0.9999 for all forces tested. For release after compression, p < 0.0001 for all forces tested. No supercoils were observed in the presence of constant tension, and thus this condition is not displayed. In the two-end fixed thermal fluctuation condition, only n = 15 supercoils were observed. For all other conditions, 1,184 ≤ n ≤ 23,258. **g** and **h**, Quantification of 3D peak-to-peak amplitude and pitch from constant compression simulations where F-actin's helical parameters (15,003 ≤ n ≤ 24,657; **g**) and persistence length (12,894 ≤ n ≤ 63,011; **h**) were systematically varied. Pitch analysis was performed with a 3D peak-to-peak amplitude cutoff of 16 nm. Bars represent medians. Two-sided Mann-Kendall test was applied to test for monotonic trends. **i**, Quantification of supercoil cross-section eccentricity from simulation conditions described in g and h, using a peak-to-peak amplitude cutoff of 16 nm (8,728 ≤ n ≤ 46,008). N = 5 simulation runs per condition. Bars represent medians; two-sided Mann-Kendall test was applied to test for monotonic trends.

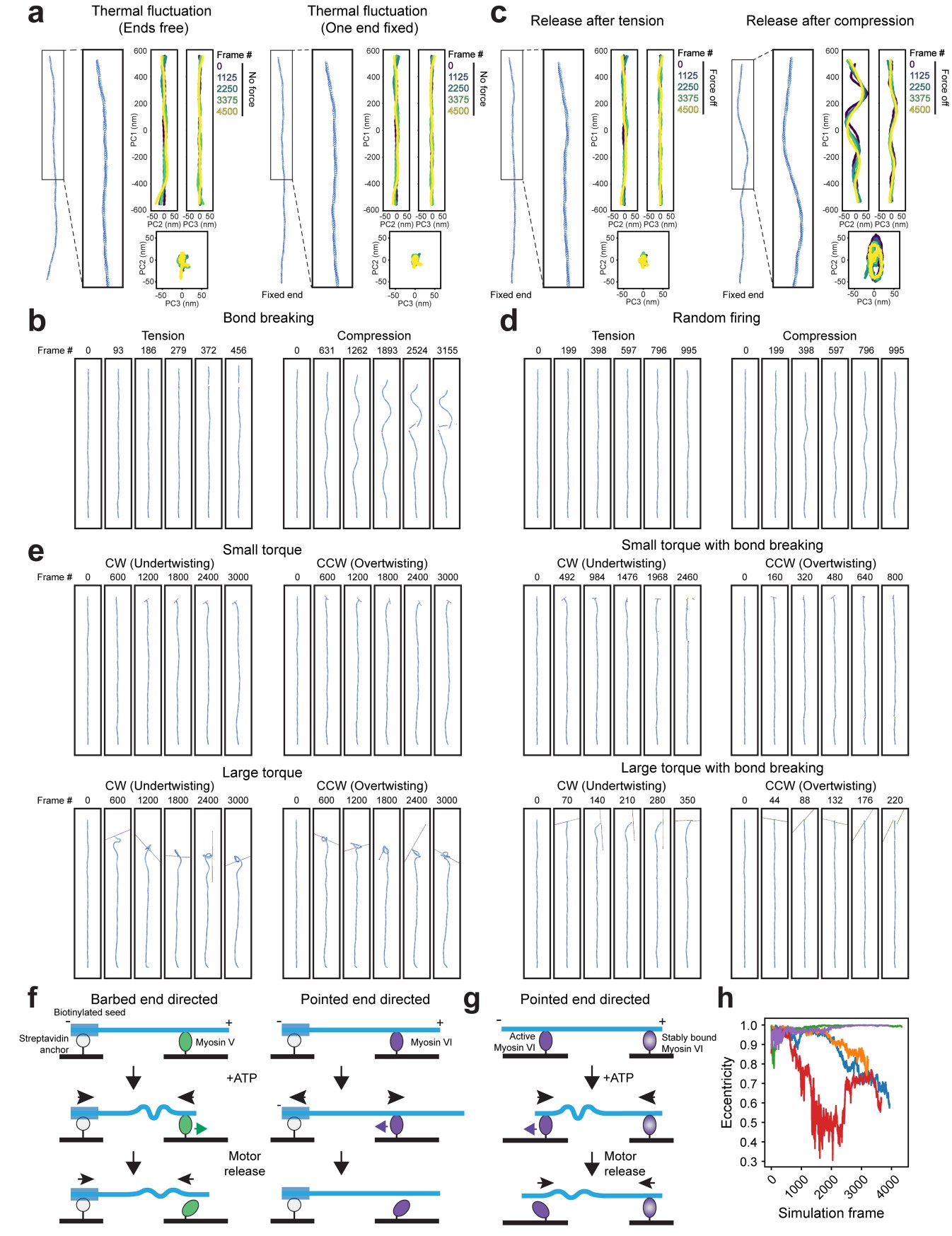

**Extended Data Fig. 4** | See next page for caption.

**Extended Data Fig. 4 | Additional coarse-grained MD simulations support axial compression as the primary mechanism of F-actin supercoil formation.** **a**, Simulation snapshots and principal component analyses of key frames in indicated thermal fluctuation conditions. **b**, Montages of constant tension/compression simulations where bond breaking is enabled. Break sites are highlighted in red and displaced subunits are highlighted in green. **c**, Simulation snapshots and principal component analyses of key frames when tension/compression is applied then released. **d**, Montages of simulations mimicking 5 randomly firing motors which apply either tensile or compressive force. **e**, Montages of simulations where clockwise (CW) or counterclockwise (CCW) torque is applied in either the absence (left) or presence (right, colored as in b) of bond breaking. Site of torque application and magnitude is indicated by vectors. **f**, Cartoon of F-actin's response to barbed-end directed (left) and pointed-end directed (right) forces applied towards a pointed-end localized biotinylated anchor. While motors are schematized as single heads for simplicity, dimeric motors capable of processive runs were employed in experiments. Supercoils are only observed in simulations mimicking the barbed-end directed force. **g**, Cartoon of alternative mechanism for supercoil formation in the pointed-end directed force condition. In this scenario, asynchronous myosin VI motors can serve as a barbed-end localized anchors during the strongly-bound phase of their mechanochemical cycles, enabling the buildup of compression. **h**, Plot of supercoil cross-section eccentricity versus simulation frame for each of the five replicates in the constant compression condition displayed in Fig. 2e.

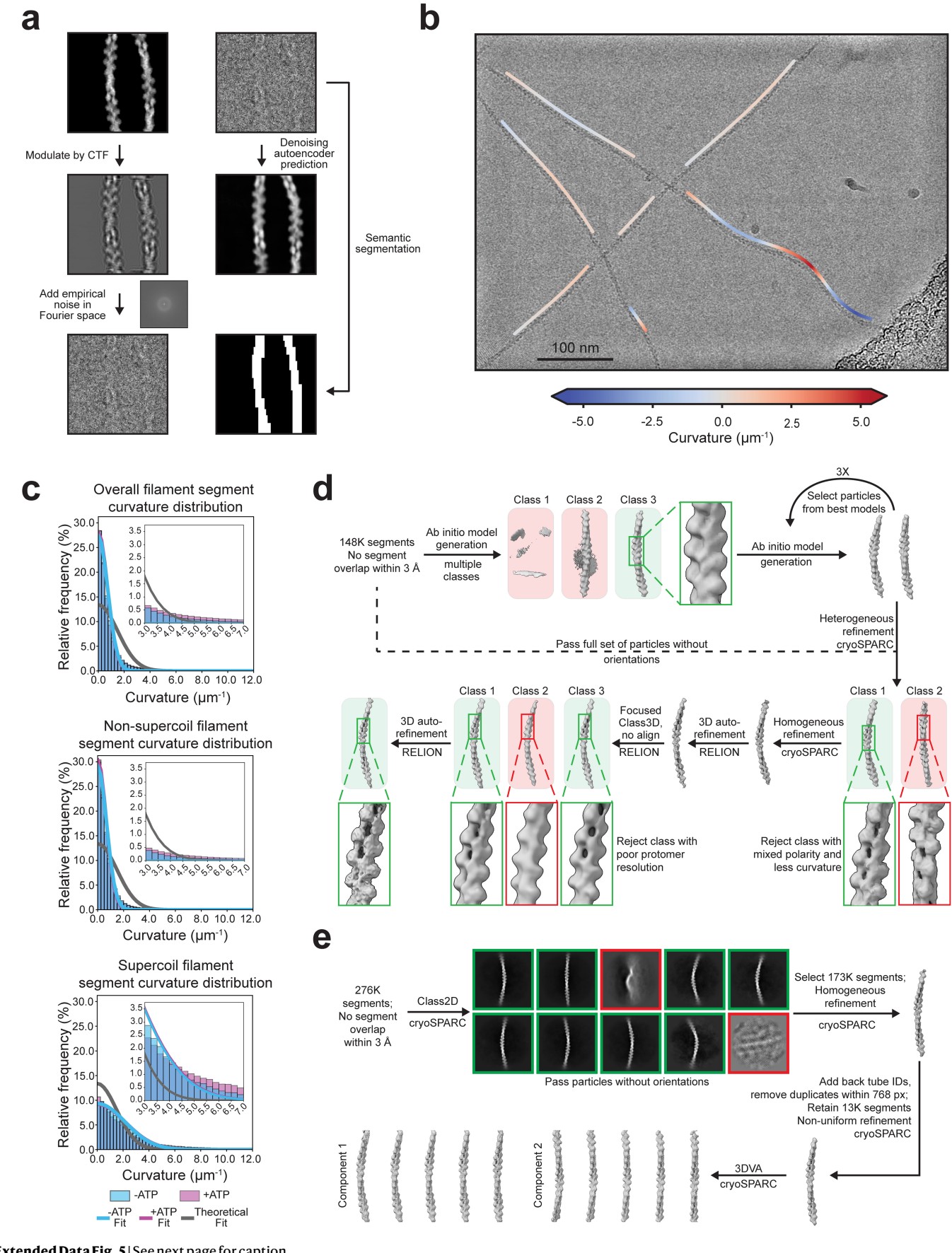

**Extended Data Fig. 5** | See next page for caption.

**Extended Data Fig. 5 | Adaptation of neural network picker and supercoil structure determination workflow. a**, Workflow for network training and subsequent picking. A synthetic particle dataset is generated by projecting PDB models of filaments featuring different computationally-generated curvatures, which are modulated by the CTF followed by the addition of a pink noise box (top). The network is then trained to denoise these noisy particles, which is then used as an input for semantic segmentation. After training with synthetic data, the network can be used to denoise and segment real data. **b**, Representative micrograph (from 31,499 collected in the dual motor +ATP condition) containing both straight filaments and filaments featuring oscillating curvature. Filaments are assigned estimated signed curvature values and categorized for subsequent selection. Traces are offset by one F-actin width for visualization. **c**, Curvature distributions of filaments identified in the dual myosin-motor evoked force condition (pink) or the −ATP control (blue). Measured curvature distributions of all picked filament segments (left), non-supercoil segments (middle), and supercoil segments (right) are shown as histograms, and modeled thermal bending fluctuation distributions are overlaid as blue, pink, and grey curves. The gray curve corresponds to a simple bending model described by Equations 9–11 (Methods) using actin's experimentally determined persistence length of 9 μm (ref. 57), while the pink and blue curves correspond to a model fitted with a multiplicative adjustment factor as described in detail in the Methods. **d**, Initial single particle cryo-EM data processing workflow for supercoil F-actin, using a single dataset collected for the dual motor condition. Transparent red and green boxes indicate rejected and accepted classes, respectively. **e**, Final processing workflow, incorporating data from two additional datasets to boost particle number and enhance the quality of the final map. Additionally, 3DVA variability analysis is displayed, highlighting the presence of continuous structural variability despite extensive classification.

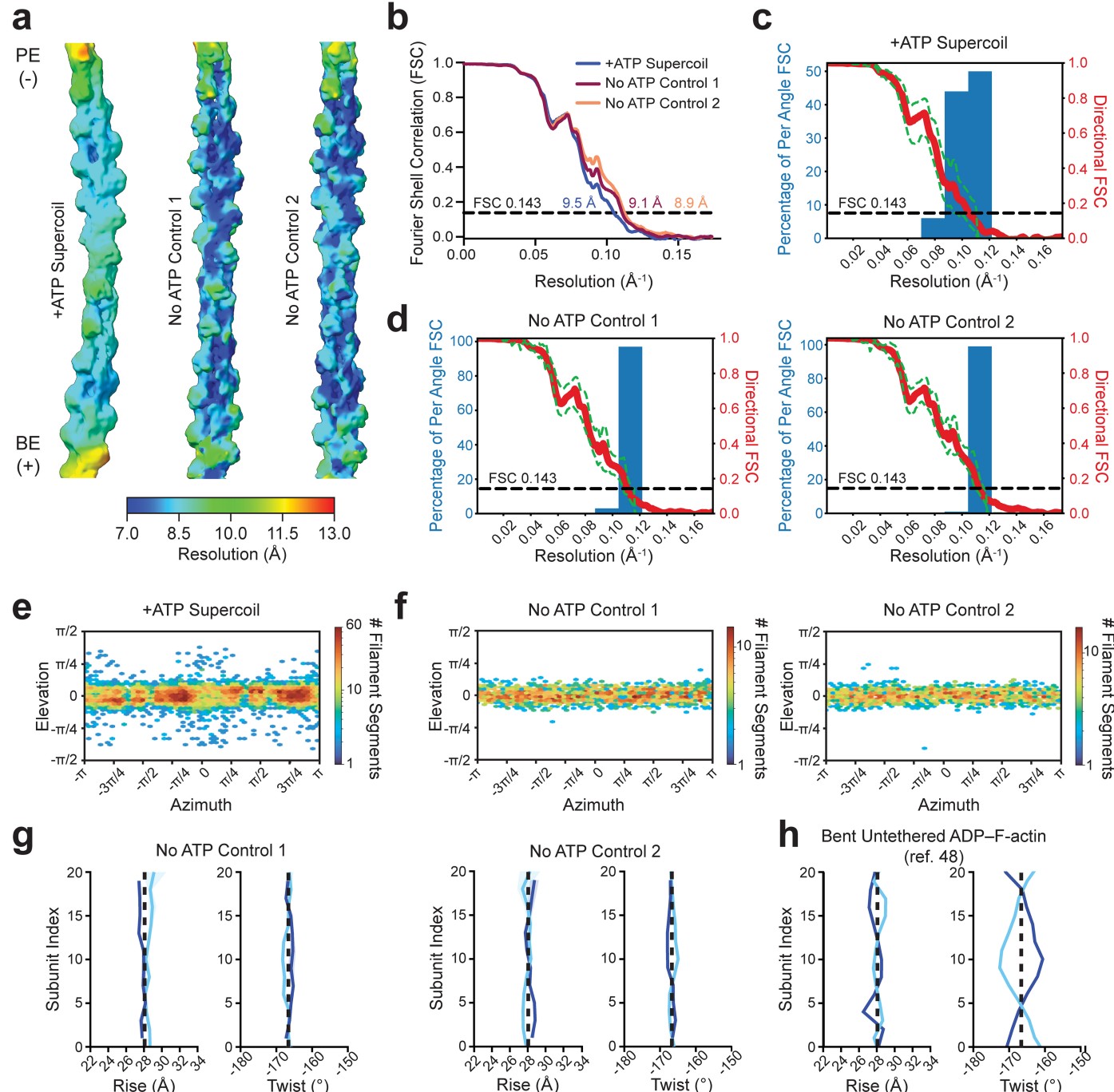

**Extended Data Fig. 6 | Resolution assessment and helical parameter analysis of supercoil and control F-actin. a**, Local resolution of +ATP myosin-force evoked supercoil F-actin (left) and two independently reconstructed −ATP control maps (right). BE: barbed end; PE: pointed end. **b**, Global Fourier Shell Correlation (FSC) curves. **c** and **d**, 3DFSC curves for +ATP supercoil F-actin (**c**) and −ATP control reconstructions (**d**). **e** and **f**, Particle orientation distribution plots for +ATP supercoil F-actin (**e**) and −ATP control reconstructions (**f**). **g**, Instantaneous helical parameters of −ATP control reconstructions, colored as in Fig. 3b. Solid lines represent means and shaded regions represent 95% CI from 3 independent analyses. **h**, Instantaneous helical parameters of bent, untethered ADP−F-actin with 5.4 μm⁻¹ curvature, from ref. 48. Vertical dashed lines indicate parameters of canonical F-actin (ref. 48).

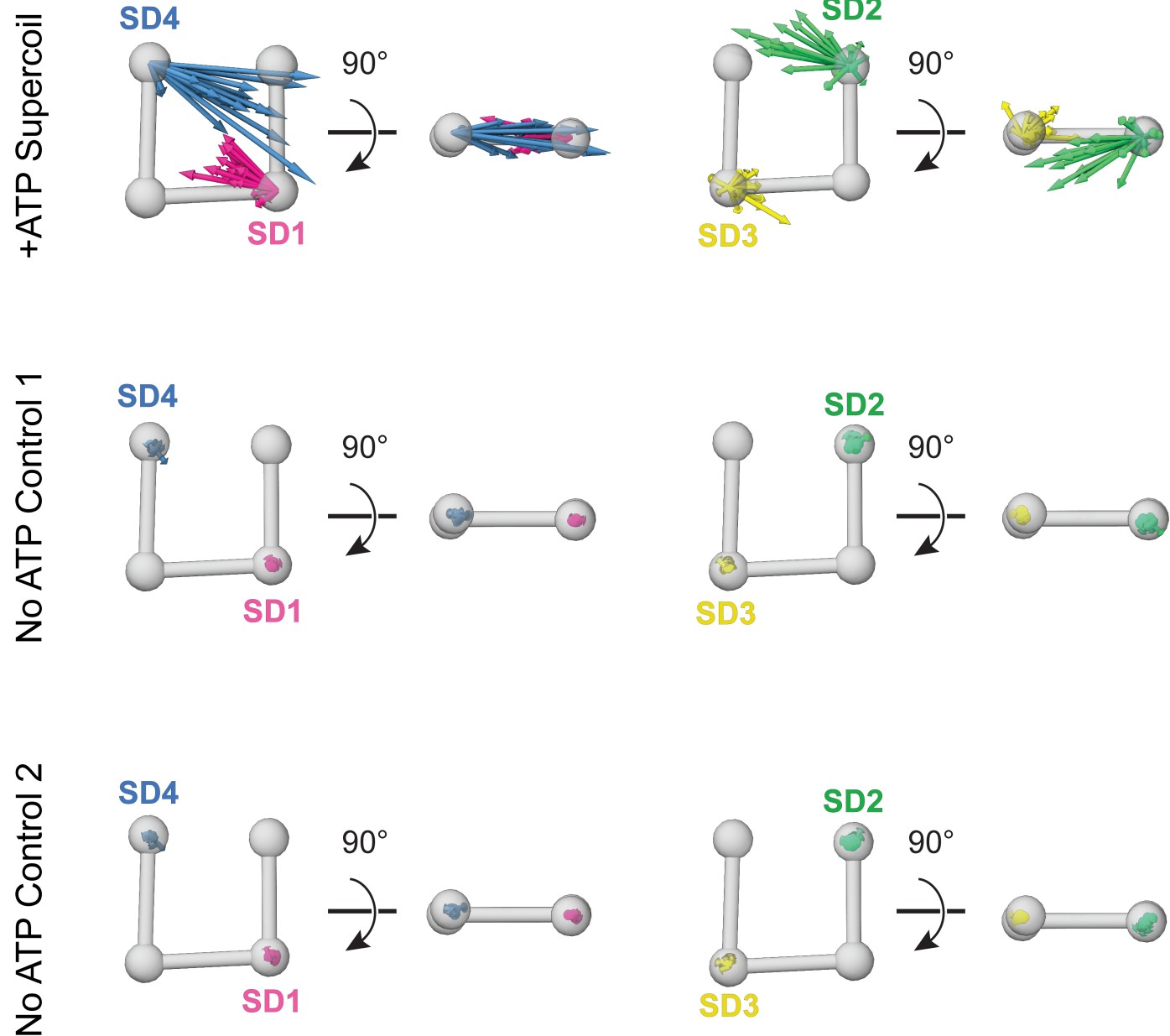

**Extended Data Fig. 7 | Subdomain displacements in supercoil F-actin.** Superimposed subdomain displacement vectors from all protomers after MDFF analysis of +ATP myosin force-evoked supercoil F-actin (top) and –ATP control reconstructions (bottom). Subdomains 1 and 4 versus 2 and 3 are displayed separately for clarity, and vectors are scaled 15X for visualization. The averages of these vectors are displayed in Fig. 3c.

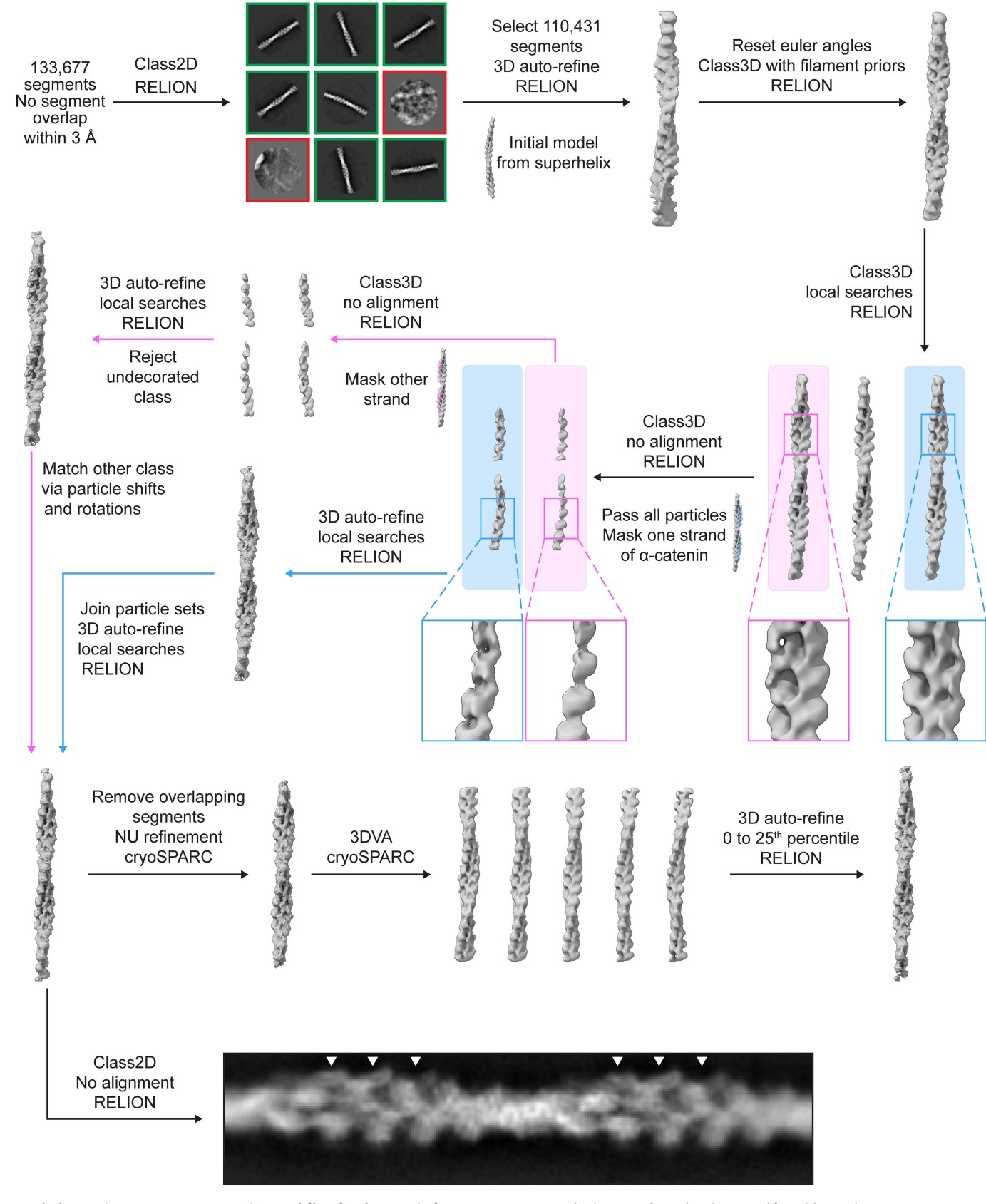

**Extended Data Fig. 8 | Cryo-EM processing workflow for the myosin force-activated α-catenin–F-actin complex.** Top: Cryo-EM processing workflow for visualizing the force-activated α-catenin-F-actin complex, from a specimen prepared in the dual motor condition. Green and red boxes represent 2D class averages which were selected and rejected for additional processing, respectively. Bottom: a magnified 2D class average is displayed, highlighting preferential binding of α-catenin along one side of the filament (arrowheads) on alternating F-actin strands.

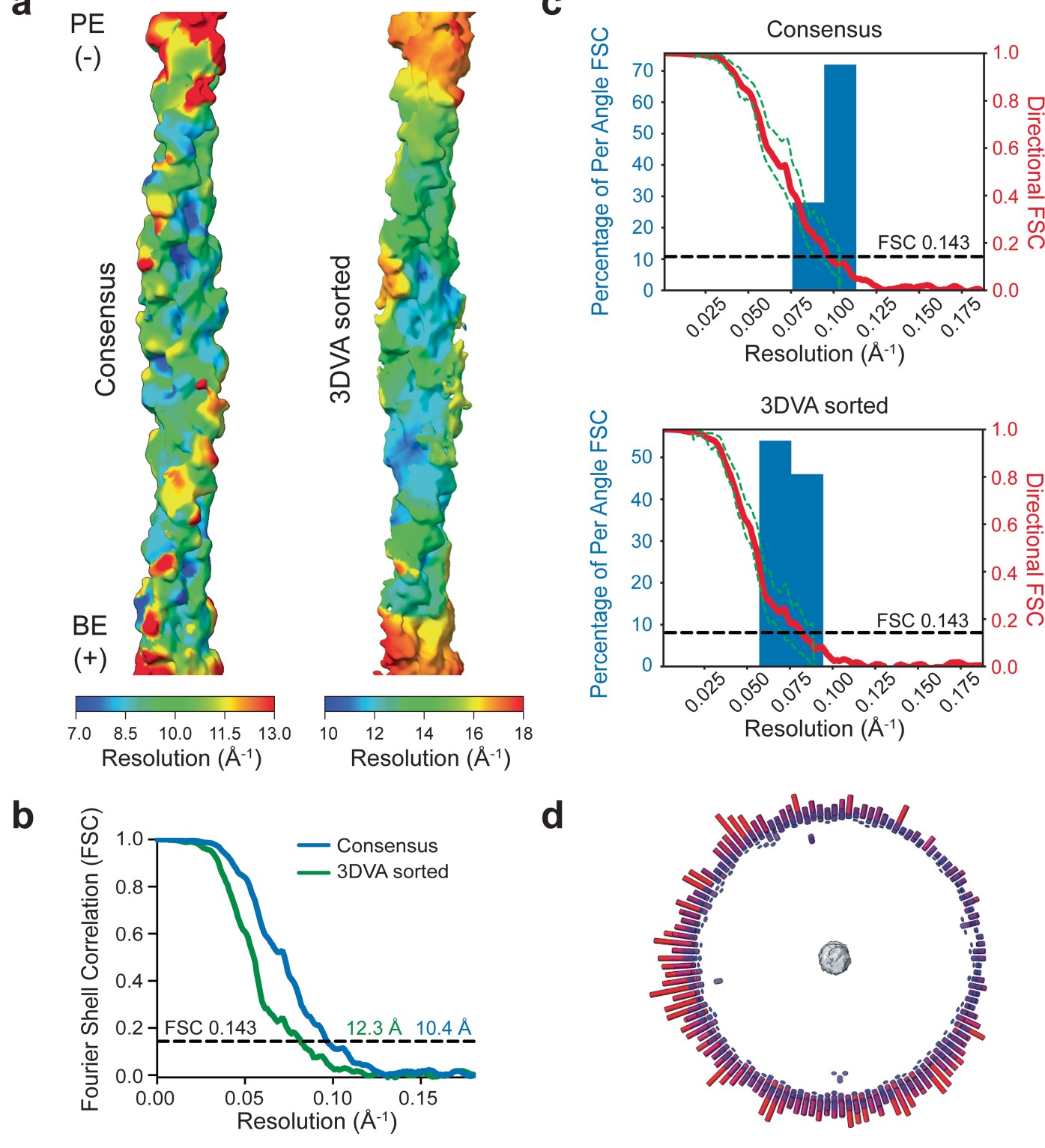

**Extended Data Fig. 9 | Resolution assessment of force-activated α-catenin–F-actin complex reconstruction. a**, Local resolution of final consensus (left) and 3DVA sorted (right) α-catenin–F-actin complex reconstructions.

BE: barbed end; PE: pointed end. **b**, Global FSC curves. **c**, 3DFSC curves. **d**, Particle orientation distribution plot.

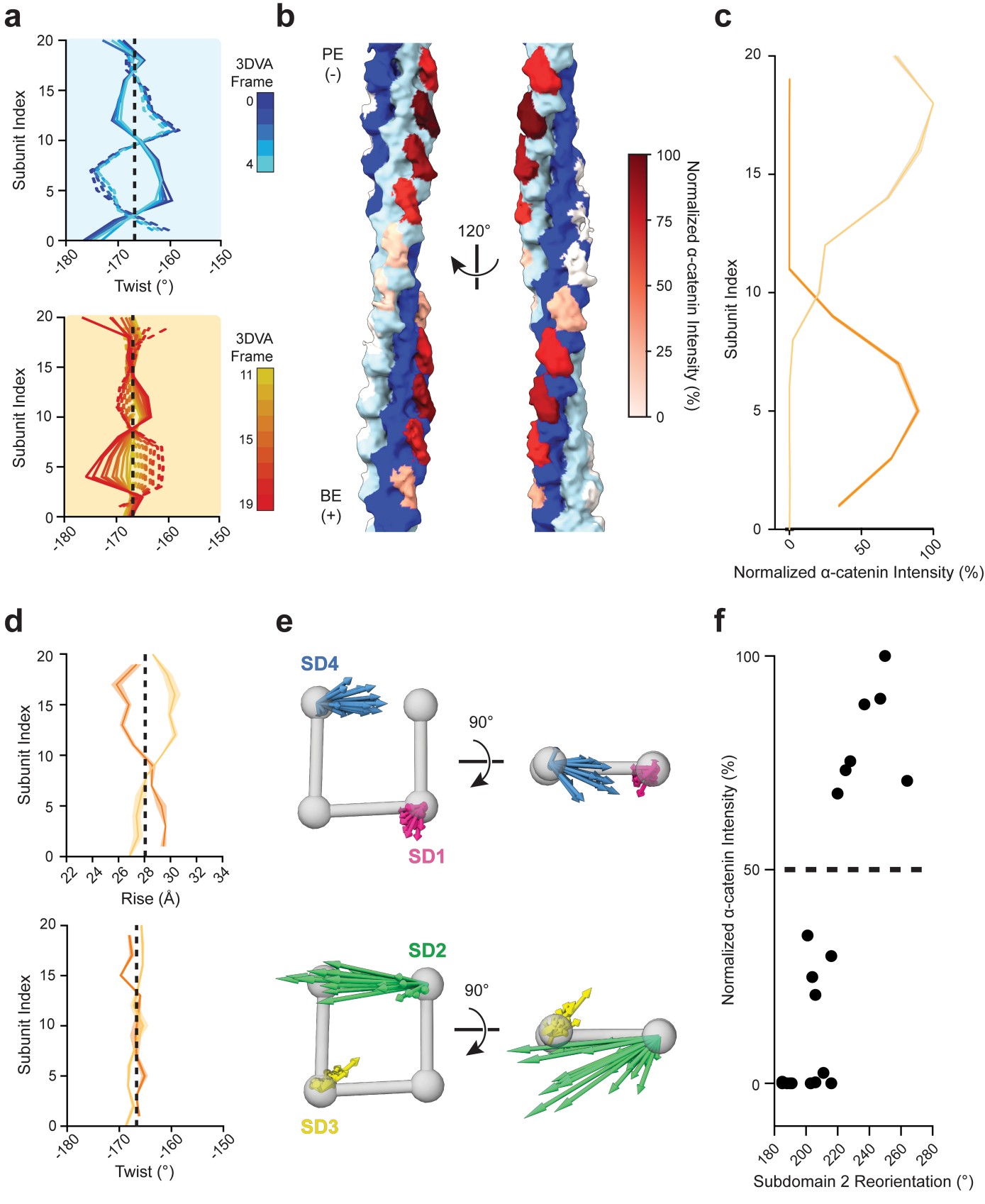

**Extended Data Fig. 10** | See next page for caption.

**Extended Data Fig. 10 | Additional analysis of the force-activated α-catenin–F-actin complex structure. a**, Instantaneous helical twist of selected 3DVA frames from Fig. 4b,c. Vertical dashed lines indicate canonical F-actin twist. **b**, Orthogonal views of the post-3DVA α-catenin–F-actin complex map, colored as in Fig. 4a. **c**, Quantification of α-catenin density intensity in post-3DVA map. Yellow corresponds to light blue strand from a, and orange corresponds to dark blue strand. Solid lines represent means and shaded regions represent 95% CI from 3 independent analyses. **d**, Instantaneous helical parameters of consensus map, colored as in b. Solid lines represent means and shaded regions represent 95% CI from 3 independent analyses. Vertical dashed lines indicate parameters of canonical F-actin. **e**, Superimposed subdomain displacement vectors versus a canonical F-actin subunit from all protomers after MDFF analysis of the consensus α-catenin–F-actin complex map, displayed as in Extended Data Fig. 7. The averages of these vectors are displayed in Fig. 4f, top. **f**, Quantification of actin subdomain 2 reorientation versus α-catenin intensity. Dashed line indicates 50%.

**Extended Data Table 1 | Cryo-EM data collection and processing statistics**

| | Myosin force-evoked supercoil F-actin (EMD-46426) | Control myosin tethered F-actin -ATP 1 (EMD-46427) | Control myosin tethered F-actin -ATP 2 (EMD-46428) | Consensus force-activated α-catenin–F-actin complex (EMD-46429) | 3DVA sorted force-activated α-catenin–F-actin complex (EMD-46431) |
|---|---|---|---|---|---|
| **Data collection and processing** | | | | | |
| Microscope | Titan Krios | Titan Krios | Titan Krios | Titan Krios | Titan Krios |
| Detector | K3 | K3 | K3 | K3 | K2 Summit |
| Magnification | 64,000 | 64,000 | 64,000 | 64,000 | 22,500 |
| Voltage (kV) | 300 | 300 | 300 | 300 | 300 |
| Electron exposure (e–/Å$^2$) | 59.156 | 59.156 | 59.156 | 59.156 | 56.53 |
| Exposure rate (e–/pixel/s) | 23.66 | 23.66 | 23.66 | 23.66 | 5.65 |
| Defocus range (μm) | -1.5 to -3.5 | -1.5 to -3.5 | -1.5 to -3.5 | -1.5 to -3.5 | -1.5 to -3.5 |
| Pixel size (Å) | 1.08 | 1.08 | 1.08 | 1.08 | 1.33 |
| Symmetry imposed | C1 | C1 | C1 | C1 | C1 |
| Micrographs (no.) | 31,799 | 3,749 | 3,749 | 4,455 | 4,455 |
| Initial particle images (no.) | 276,493 | 345,515 | 345,515 | 133,677 | 133,677 |
| Final particle images (no.) | 13,146 | 3,436 | 3,137 | 3,289 | 822 |
| Map resolution (Å) | 9.5 | 9.1 | 8.9 | 10.4 | 12.3 |
| FSC threshold | 0.143 | 0.143 | 0.143 | 0.143 | 0.143 |
| Map resolution range (Å) | 7.0 to 13.0 | 7.0 to 13.0 | 7.0 to 13.0 | 7.0 to 13.0 | 10.0 to 18.0 |
| | | | | | |
| **Refinement** | | | | | |
| Initial model used (PDB code) | - | - | - | - | - |
| Model resolution (Å) | - | - | - | - | - |
| FSC threshold | - | - | - | - | - |
| Model resolution range (Å) | | | | | |
| Map sharpening $B$ factor (Å$^2$) | -886.6 | -850.8 | -801.5 | -1151.9 | -100.0 |
| Model composition | | | | | |
| Non-hydrogen atoms | - | - | - | - | - |
| Protein residues | - | - | - | - | - |
| Ligands | - | - | - | - | - |
| $B$ factors (Å$^2$) | | | | | |
| Protein | - | - | - | - | - |
| Ligand | - | - | - | - | - |
| R.m.s. deviations | | | | | |
| Bond lengths (Å) | - | - | - | - | - |
| Bond angles (°) | - | - | - | - | - |
| Validation | | | | | |
| MolProbity score | - | - | - | - | - |
| Clashscore | - | - | - | - | - |
| Poor rotamers (%) | - | - | - | - | - |
| Ramachandran plot | | | | | |
| Favored (%) | - | - | - | - | - |
| Allowed (%) | - | - | - | - | - |
| Disfavored (%) | - | - | - | - | - |

Please see Methods for additional details about cryo-EM data collection and processing.

**Extended Data Table 2 | Coarse grained molecular dynamics simulation parameters**

| Parameter | Simulation units | Physical units |
|---|---|---|
| $k_{l\_diag}$ | 300 [$F$]/[$l$] | 1200 pN / nm |
| $k_{l\_long}$ | 20 [$F$]/[$l$] | 80 pN / nm |
| $k_{\theta 1}$ | 435 [$E$] | $1.74 \times 10^{-15}$ kJ |
| $k_{\theta 2}$ | 580 [$E$] | $2.32 \times 10^{-15}$ kJ |
| $k_{\theta 3}$ | 435 [$E$] | $1.74 \times 10^{-15}$ kJ |
| $k_{\theta 4}$ | 290 [$E$] | $1.16 \times 10^{-15}$ kJ |
| $k_{\theta 5}$ | 290 [$E$] | $1.16 \times 10^{-15}$ kJ |
| $k_{\theta 6}$ | 145 [$E$] | $5.80 \times 10^{-16}$ kJ |
| $k_{\phi 1}$ | 1100 [$E$] | $4.4 \times 10^{-15}$ kJ |
| $k_{\phi 2}$ | 55 [$E$] | $2.2 \times 10^{-16}$ kJ |

Please see Methods for additional details about molecular dynamics simulations.

# Reporting Summary

## Statistics

For all statistical analyses, confirm that the following items are present in the figure legend, table legend, main text, or Methods section.

| n/a | Confirmed | |
|---|---|---|
| ☐ | ☒ | The exact sample size (*n*) for each experimental group/condition, given as a discrete number and unit of measurement |
| ☐ | ☒ | A statement on whether measurements were taken from distinct samples or whether the same sample was measured repeatedly |
| ☐ | ☒ | The statistical test(s) used AND whether they are one- or two-sided *Only common tests should be described solely by name; describe more complex techniques in the Methods section.* |
| ☒ | ☐ | A description of all covariates tested |
| ☐ | ☒ | A description of any assumptions or corrections, such as tests of normality and adjustment for multiple comparisons |
| ☐ | ☒ | A full description of the statistical parameters including central tendency (e.g. means) or other basic estimates (e.g. regression coefficient) AND variation (e.g. standard deviation) or associated estimates of uncertainty (e.g. confidence intervals) |
| ☐ | ☒ | For null hypothesis testing, the test statistic (e.g. $F$, $t$, $r$) with confidence intervals, effect sizes, degrees of freedom and $P$ value noted *Give P values as exact values whenever suitable.* |
| ☒ | ☐ | For Bayesian analysis, information on the choice of priors and Markov chain Monte Carlo settings |
| ☒ | ☐ | For hierarchical and complex designs, identification of the appropriate level for tests and full reporting of outcomes |
| ☒ | ☐ | Estimates of effect sizes (e.g. Cohen's *d*, Pearson's *r*), indicating how they were calculated |

*Our web collection on statistics for biologists contains articles on many of the points above.*

## Software and code

Policy information about availability of computer code

| | |
|---|---|
| Data collection | Cryo-EM and Cryo-ET data were collected with SerialEM. Low-magnification cryo-EM data (displayed in Figure 1) were collected with Leginon. Epi-fluorescence movies were collected with MetaMorph (Molecular Dynamics). TIRF images were collected with Nikon NIS-Elements. Cryo-fluorescence images were collected with Leica's LAS X software. |
| Data analysis | For cellular cryo-ET analysis, direct detector frame series were aligned with MotionCor2, and CTF estimation was performed with CTFFIND4. Tilt series were aligned with AreTomo2, then reconstructed with IMOD. Reconstructed tomograms were denoised with cryoCARE and IsoNET. Segmentation was performed with Dragonfly and MemBrain v2.<br><br>For in vitro cryo-ET analysis, frames were aligned with MotionCor2. Tilt series were aligned with Appion-Protomo, then reconstructed with Tomo3D. Tomograms were denoised with a custom denoising autoencoder approach, implemented with TensorFlow and functions from the EMAN2 package in Python.<br><br>For cryo-EM analysis, direct detector frame series were aligned with MotionCor2, and CTF estimation was performed with CTFFIND4. Filament picking and curvature analysis were performed with a custom denoising autoencoder approach, implemented with TensorFlow and functions from the EMAN2 package in Python. Classification, reconstruction, and variability analysis were performed with cryoSPARC v3.2.0 and RELION v3.1.2. Molecular dynamics flexible fitting was performed with ISOLDE on atomic models idealized with Phenix.<br><br>Coarse-grained molecular dynamics simulations were performed with ESPResSO and analyzed with custom code in Python.<br><br>Fluorescence microscopy data were analyzed with FIJI. UCSF Chimera and ChimeraX were used for molecular graphics and structural analysis. Statistical analysis and plotting were conducted with GraphPad Prism, SciPy, and Matplotlib. |

Custom code was prepared with the assistance of ChatGPT 4.0.

All custom code is available at www.github.com/alushinlab/squiggle.

For manuscripts utilizing custom algorithms or software that are central to the research but not yet described in published literature, software must be made available to editors and reviewers. We strongly encourage code deposition in a community repository (e.g. GitHub). See the Nature Portfolio guidelines for submitting code & software for further information.

# Data

Policy information about availability of data

All manuscripts must include a data availability statement. This statement should provide the following information, where applicable:
- Accession codes, unique identifiers, or web links for publicly available datasets
- A description of any restrictions on data availability
- For clinical datasets or third party data, please ensure that the statement adheres to our policy

Cryo-EM density maps have been deposited in the EMDB with the following accession codes: Myosin force-evoked supercoil F-actin (EMD-46426); Control myosin tethered F-actin -ATP 1 (EMD-46427); Control myosin tethered F-actin -ATP 2 (EMD-46428); Consensus force-activated α-catenin–F-actin complex (EMD-46429); 3DVA sorted force-activated α-catenin–F-actin complex (EMD-46431). Additional raw and processed data files are available at Zenodo (refs. 114,115,116). Cellular tomograms, segmentations, and filament traces are available at https://doi.org/10.5281/zenodo.18022754. Trained neural networks and denoised in vitro tomograms with filament traces are available at https://doi.org/10.5281/zenodo.12702199. Trained neural networks used for single particle analysis particle picking, flexibly fit PDB models, and variability analysis maps and models used for data analysis are available at https://doi.org/10.5281/zenodo.12702799. Source data are provided for Figs. 1-4 and Extended Data Figs. 1, 2, 6, and 10. Additional source data are available at Zenodo (refs. 117,118) for simulations (Fig. 2f-h and Extended Data Fig. 3; https://doi.org/10.5281/zenodo.17932417) and Extended Data Fig. 5 (https://doi.org/10.5281/zenodo.17958929). All reagents and resources reported in this study are freely available from the corresponding author.

# Research involving human participants, their data, or biological material

Policy information about studies with human participants or human data. See also policy information about sex, gender (identity/presentation), and sexual orientation and race, ethnicity and racism.

| | |
|---|---|
| Reporting on sex and gender | N/A |
| Reporting on race, ethnicity, or other socially relevant groupings | N/A |
| Population characteristics | N/A |
| Recruitment | N/A |
| Ethics oversight | N/A |

Note that full information on the approval of the study protocol must also be provided in the manuscript.

# Field-specific reporting

Please select the one below that is the best fit for your research. If you are not sure, read the appropriate sections before making your selection.

☒ Life sciences          ☐ Behavioural & social sciences          ☐ Ecological, evolutionary & environmental sciences

For a reference copy of the document with all sections, see nature.com/documents/nr-reporting-summary-flat.pdf

# Life sciences study design

All studies must disclose on these points even when the disclosure is negative.

| | |
|---|---|
| Sample size | Sample sizes were not pre-determined. The amount of data collected were limited by our capacity to prepare high-quality specimens and the length of cryo-EM imaging sessions. Combining multiple datasets only modestly enhanced the resolution of our reconstructions, suggesting that we had reached the limitations imposed by the molecular heterogeneity of our samples. |
| Data exclusions | No data were excluded from analysis. |
| Replication | F-actin sinusoidal regions were observed in three independent low magnification cryo-EM experiments (dual motor, pointed-end directed force, and barbed-end directed force conditions), three high-magnification single particle cryo-EM datasets in the dual motor condition, and two in vitro cryo-ET datasets (pointed-end directed force and barbed-end directed force). Imaging with both cryo-ET and cryo-EM visualized highly similar morphological features. |
| Randomization | For Fourier Shell Correlation based resolution analysis, particles were randomly assigned to half-datasets for comparison. For other |

| Randomization | experiments in this study, as is customary we did not perform randomization, since biophysical experiments are performed under highly controlled experimental conditions where co-variates are minimal. Furthermore, specific individual cells and molecules cannot be randomly assigned to groups: instead, large numbers of molecules or cells from a population are subdivided into experimental conditions (e.g. aliquots from a tube) for comparison. As these divisions are not controlled, they effectively achieve a similar effect as intentional randomization of individual subjects. |
|---|---|
| Blinding | To avoid bias in manual analysis of cellular tomograms, blinding was performed for analysis presented in Fig. 1C / Extended Data Fig. 1f, as described in the Methods. Different investigators performed experiments and analysis; thus, those performing analysis were blinded to group allocation during data collection.<br><br>For other comparisons between experimental conditions (e.g. reconstitution experiments -/+ ATP), blinding was not performed. Analysis was performed with automated methods with minimal human intervention for these experiments, and thus blinding was not required. |

# Reporting for specific materials, systems and methods

We require information from authors about some types of materials, experimental systems and methods used in many studies. Here, indicate whether each material, system or method listed is relevant to your study. If you are not sure if a list item applies to your research, read the appropriate section before selecting a response.

## Materials & experimental systems

| n/a | Involved in the study |
|---|---|
| ☐ | ☒ Antibodies |
| ☐ | ☒ Eukaryotic cell lines |
| ☒ | ☐ Palaeontology and archaeology |
| ☒ | ☐ Animals and other organisms |
| ☒ | ☐ Clinical data |
| ☒ | ☐ Dual use research of concern |
| ☒ | ☐ Plants |

## Methods

| n/a | Involved in the study |
|---|---|
| ☒ | ☐ ChIP-seq |
| ☒ | ☐ Flow cytometry |
| ☒ | ☐ MRI-based neuroimaging |

## Antibodies

| Antibodies used | We used monoclonal anti-GFP antibody G6539, purchased from Sigma-Aldrich, to anchor GFP-tagged myosin motor proteins to the surface of cryo-EM grids. |
|---|---|
| Validation | This is a widely-used commercial anti-GFP antibody, which according to the manufacturer has been cited in 170 publications. The manufacturer provides a certificate of analysis for each batch of the antibody, validated through its performance in western-blotting with appropriate controls. Given its widespread use, we did not independently validate the antibody, beyond its apparent success in anchoring myosin to grids in a manner which maintains motility (evident in Supplementary Video 2). |

## Eukaryotic cell lines

Policy information about cell lines and Sex and Gender in Research

| Cell line source(s) | PtK2 (#CCL-56) and HEK293T cells (#CRL-3216) were purchased from ATCC. |
|---|---|
| Authentication | ATCC validates cell lines by STR profiling. We did not further validate lines after purchase. |
| Mycoplasma contamination | Cells were monitored for mycoplasma contamination with the ATCC universal mycoplasma detection kit (a PCR-based assay), and tested negative. |
| Commonly misidentified lines<br>(See ICLAC register) | No commonly mis-identified cell lines were used in this study. |

## Plants

| Seed stocks | N/A |
|---|---|
| Novel plant genotypes | N/A |
| Authentication | N/A |

