## [Peer Review file · Nature]

Myosin forces remodel F-actin for mechanosensitive protein recognition

Corresponding Author: Professor Gregory Alushin

Version 0:

Reviewer comments:

Referee #1

(Remarks to the Author)

This manuscript by Carl et al. addresses a longstanding and very important question: how can mechanical forces applied to actin filaments, primarily by myosin motors in cells, change the conformation of the filaments and their ability to bind regulatory proteins? This question is of paramount importance since it appears that mechanical forces induce biochemical changes, thereby regulating the architecture and the dynamics of the actin cytoskeleton. However, until now, structural evidence of the putative mechanically-induced conformational changes is lacking, hindering our ability to understand this key mechanism. The main reason for this lack of data is technical, as there is no straightforward way to apply controlled mechanical forces to a large population of filaments on an electron microscope grid.

Here, the authors propose a way to do so, building on a previous assay developed in their lab, using two types of myosins motors. These in vitro experiments are complemented by observations in cells and by numerical simulations. They propose that compression and tension can both lead the filaments to adopt a new, supercoiled conformation.

This paper could be a landmark, and thus it would be terrible if flawed conclusions were published and were then to mislead scientists for years to come. Currently, I am afraid there is such a risk. I find the work exciting, and I hope that the authors will be able to alleviate my concerns.

My main problem is that I am not convinced by one of the main claims of the paper, that filaments under tension adopt a superhelical conformation, for the following reasons.

- First, it is not clear to me that filaments are only under tension in the "pointed end" experiments. Uncontrolled filament sticking to the surface could cause the pointed end motors to generate compressive forces. Control experiments should be performed to show that this is not the case. Also, based on the way the filaments are prepared (page 42) it seems likely that the pointed ends of some biotinylated segments anneal with barbed ends of non-biotinylated segments. This would also result in portions of filaments being compressed in the experiment with myosin VI. I think the experiments could be done differently (for example, capping the pointed ends of the biotinylated seeds) to ensure that this problematic annealing does not happen. Also, it would be informative to quantify the occurrence of the different types of filament segments (straight vs. oscillatory) in each experiment.

- Second, I do not find the simulations convincing. Based on the movies, it appears that the filaments are simply able to fluctuate thermally once they are released. A control filament, with no mechanical force applied (neither tension nor compression, neither constant nor released) is required to clarify this. For the filament under constant compression, I would expect buckling to readily occur since the filament is approximately 1 μm long and a 25 pN compressive force is applied. Is that what we are observing? If that simulation were to run a bit longer, it seems that the filament would continue to collapse, wouldn't it?

- Third, regardless of the issues mentioned in the previous paragraphs, I find it hard to compare the simulations with what is happening in the in vitro experiment and in cells. In the simulations, in order to observe an oscillatory shape when applied tension, a 25 pN force is applied and released. This would correspond to several motors (5 motors, according to the authors' estimation, page 49) pulling on the motor and simultaneously letting go. I have a hard time imagining this taking place in the experiments. Also, this would not correspond to the filaments that are observed with both ends attached to the surface.

Rather, I expect that, in the experiments, filaments are pulled on by several motors, with non-synchronized cycles. The applied tension would certainly fluctuate over time around an average value, but not drop rapidly to zero as it is currently done in the simulations.

Other points:

- The motors probably apply a significant mechanical torque to the filaments. This should be an important mechanical contribution that could lead to supercoiling. Why is this aspect ignored?
- The filaments in EM micrographs exhibit straight segments coexisting with oscillatory segments, with seemingly sharp transitions between the two. How can this be understood? The simulations (movie and Fig 2f) do not appear to show anything like this.
- I am not sure to understand the authors' point about the elliptical cross-section of the superhelical filaments. The way I see it, superhelical filaments with a circular cross-section would be deformed (because of thermal fluctuations, at least) and their cross-sections would then appear elliptical in the experiments. Aligning these ellipses (e.g. Fig 2c) will of course produce an average elliptical shape, but this does not indicate that the individual spirals intrinsically have elliptical cross-sections.
- I could not find the criteria for classifying the different segments of filaments as straight or oscillatory (Fig 1). What were these criteria? Was this classification automated or done manually?

Minor :

- I find the terminology "pointed end" and "barbed end" (Fig 1c and elsewhere) confusing because it refers to the directionality of the motors, while in my opinion it would make more sense to refer to the direction of the applied force, which is in the opposite direction.
- In Fig 3a, indicating the subunit indices would be useful. It would help to see the correspondance with the other panels.
- The color choices for rises and twists in Fig 3b and Fig 4c are a bit confusing, because they are the same colors as for the two filament strands (for example, light blue correspond to the rise from light to dark blue, while dark blue corresponds to the rise from dark to light blue). Maybe a different color code would help?

Referee #2

(Remarks to the Author)

This study investigates how forces generated by myosin motors influence the structure of actin filaments and their ability to bind tension-sensing proteins such as zyxin and α -catenin. These findings could reveal more broadly how mechanical forces affect filament interactions with actin-binding proteins (ABPs) and contribute to signalling events.

The reader is expected to draw two main conclusions from this paper. First, actin filaments under tension exhibit a regular wavy (oscillatory) structure. Second, this wavy conformation enhances the binding of specific ABPs that respond to tension. If these findings are broadly applicable (generally true), they would represent an important and novel contribution to the field, meriting publication in a high-impact journal like Nature. Conversely, if these observations are isolated or incongruent with the authors' conclusions, they would still be of interest but might not have the broader impact required for such a prestigious venue.

My current evaluation is that, although the work is technically sound and creative, there are significant concerns about the validity of the interpretations and the overall significance of the observations.

Major Criticisms:

1. Statistical Significance of the Cellular Data: Figure 1 suggests that actin stress fibres under tension, marked by increased zyxin decoration, display wavy filaments. Is this observation statistically significant? The statement "Many filaments in both of these types of networks exhibited substantial curvature" on Page 4 is not quantitative. A thorough statistical analysis is needed to determine whether this wavy behaviour is observed in 0.01% or 50% of zyxin-decorated filaments, using the already available cryo-ET data. Additionally, Peter Gunning's work shows that most actin filaments in stress fibres are decorated with tropomyosin. How does this reconcile with uniform zyxin decoration and the presence of waves? Are there mixtures of tropomyosin-decorated straight filaments and zyxin-decorated wavy filaments in these stress fibres? A deeper cryo-ET characterisation of these stress fibres in cells might provide the most compelling and convincing narrative for publication in Nature.
2. Statistical Significance of the Cryo-EM Data: Similar concerns apply to the cryo-EM data. The technical challenge of freezing filaments under tension in cryo-grids is acknowledged, and the authors should be commended for this achievement. However, critical questions remain, such as the percentage of filaments showing wavy conformations as a function of myosin (V and VI) concentration and/or \pm ATP. The \pm ATP superhelix quantification shown in Fig. S3c suggests these appear in both \pm ATP and are only moderately more prevalent +ATP. Perhaps this could be moved to the main text, along with a different quantification, i.e., the absolute number of filaments showing vs. not showing superhelices per micrograph. However, it is concerning that superhelices are observed in the -ATP condition, because: a) are superhelices truly due to myosin-generated tension? b) The majority of the superhelices characterised in the +ATP condition are independent of myosin activity, as they are also observed in -ATP conditions.
3. Additionally, if myosins are attached to the carbon area, why do all the examples of wavy filaments shown in the paper not

extend into this region? Coincidentally, the colour tracing of filaments should extend into the carbon area to further clarify this point. Furthermore, why are only sections of some filaments wavy? Tension is expected to be propagated along the entire filament length, is it not? Do the waves dissipate with the distance from the carbon support? Are freezing conditions, ice thickness, or other experimental variables influencing this behaviour? Are there myosin motors in the holes also exerting forces on the filaments? How are the waves shown in Fig. S2c generated? If filaments are attached to the carbon support at only one end, how is tension maintained and not released? The authors' claim of filament "memory" for deformation, persisting even after tension release ("domains have the capacity to persist after filament breakages," Page 5), seems questionable. It is more plausible that such breakages occur during freezing. Furthermore, it is unclear why both myosins produce similar waves when analysed individually. Myosin VI, moving towards the pointed end, should generate pulling tension, while myosin V would likely buckle the filament, but regular wavelength buckles are unexpected. Could the authors provide a cartoon illustration to clarify all of these unexpected observations? The simulations attempt to address some of these questions, but what is really lacking is experimental evidence in support of these unusual observations and conclusions.

4. The reconstructions have resolutions ranging from 9 to 12 Å. Given this limited resolution, caution should be exercised in placing too much reliance on detailed analyses of subdomain motions within actin subunits (\pm α -catenin), as depicted in several figures (e.g., 3c, 4f, Fig. S6). While the filament-level waves are clearly observable and analysable, attempts to assess subunit-level causes of these waves may be unjustified given the resolution constraints. It is prudent to be cautious, particularly considering the significant variability in wavelength and the apparent infrequent observation of these waves. In other words, there is a risk of drawing elaborate conclusions based on a limited foundation.

5. I am also quite sceptical for another reason: why do filaments in muscle sarcomeres, which experience significant forces from numerous myosin heads, not display waves (DOI: 10.1126/science.abn1934; DOI: 10.1016/j.cell.2021.02.047)? Although sarcomeres can contract, substantial force is developed on individual filaments from both sides of the Z-disk, yet waves are not observed during contraction or relaxation conditions. It may be suggested that filaments in sarcomeres are decorated with proteins like tropomyosin. Yet, a similar situation occurs in non-muscle cells.

6. It is suggested that α -catenin preferentially binds to superhelices at sites with a higher subunit rise. The interpretation is that α -catenin 'recognises' these regions to alleviate strain by repositioning subdomain 2 of the actin subunit it binds to. However, as previously noted, drawing conclusions about subdomain movements from such low-resolution maps is precarious. Furthermore, it seems exaggerated to imply that α -catenin selectively binds to these sites, given that it can also bind to canonical (straight) actin filaments that are not under tension, as acknowledged by the authors. An alternative explanation might be that the binding site for α -catenin is more accessible in regions with a higher rise and masked in regions with a lower rise. Consequently, this observation does not provide a definitive explanation for the force-activated binding of α -catenin.

7. The more I consider the data presented, the more I am concerned that we might merely be observing the reported filament supercoiling occurring as a result of motor-induced twisting (DOI: 10.1093/plphys/kiad095; DOI: 10.1038/s41467-022-28961-x), which occurs as myosin motors move in a spiral path along the actin filament. How can it be demonstrated that the slight increase in waves observed in the +ATP condition is not due to myosin supercoiling?

Minor Points:

1. Page 59: The numbering of Methods References should probably follow the order of references in the Main Text.
2. Coarse-Grained Molecular Dynamics Simulations: These simulations may be valid and align with the experimental observations. However, they could be moved to the supplementary materials to maintain a concise and focused narrative. The primary emphasis should be on the strength and reliability of the experimental data. The simulations, while potentially informative, are supporting information rather than hard evidence.
3. Orientation distribution plots are not provided for any of the maps.

Referee #3

(Remarks to the Author)

This is an excellent paper that provides new insights into both actin and actomyosin force generation. Very creative approaches have been used to reveal a supercoiling of actin filaments caused by interactions with myosin. Many questions will be raised by this work, opening up new areas of investigation. My concerns and suggestions are relatively minor, but I think that addressing them will improve the paper.

First, I think that the term "spiral" should be dropped completely. The definition of a spiral is "winding in a continuous and gradually widening (or tightening) curve, either around a central point on a flat plane or about an axis so as to form a cone." But there is no change in radius in these supercoils, and they are not conical, so the terms "supercoil" and "supercoiling" should be used. Consider bacterial and archaeal flagellar filaments. These, in general, are homopolymers, and non-motile mutants can be found that are straight. But the wild-type filament supercoils, and these helical supercoils have been studied for > 50 years. Similarly, a coiled-coil contains two alpha-helices that are supercoiled, and these are never described as spirals. Further, "oscillatory" should be replaced by "sinusoidal", which is much clearer. The projection of a helix onto two-dimensions is a sinusoid, and the curves that are shown are sinusoidal, while oscillatory could be much more general. I would also suggest that "domains" (as in "individual filaments featuring domains of sharply oscillating curvature spanning ~300-400 nanometers") might simply be replaced by "regions", as the term subdomains has been used to describe the structure of the actin molecule. It may be confusing for the reader if domains and subdomains are used, with one having nothing to do with the other.

I think that the paper could be strengthened with background information about how actin can be modified so that actomyosin force generation is inhibited, but the trivial explanations that these modifications prevent myosin binding, or fail to activate the myosin ATPase activity, can be dismissed. Some old examples are (Prochniewicz and Yanagida, 1990),

(Prochniewicz et al., 1993), (Schwyter et al., 1990), (Drummond et al., 1990).

The simulations shown are very interesting, but perhaps they can be extended and more information deduced from them. The most interesting question for me is what determines the pitch (wavelength) of these supercoils? The supercoils can be characterized by two parameters, the radius and the pitch. How are these related to the helical parameters of the actin filament? The helical parameters of the model can be easily changed, and it would be interesting to see the effects. How does the ellipticity of the supercoil arise? This is never discussed. Is it purely the flexural rigidity of F-actin that determines the pitch, such that a more rigid actin filament would lead to a longer pitch? While the extensibility of the model is discussed, is the flexural rigidity of the model reasonable, and does it yield a persistence length of ~ 10 microns, as determined experimentally?

The modulation of the helical rise in the filament is very interesting, but some additional references could strengthen this aspect of the paper. It was traditionally assumed in the earliest models for muscle that F-actin was inextensible. But x-ray diffraction observations from the 1990s showed otherwise. There is a very interesting discussion of this (Goldman and Huxley, 1994), particularly the last paragraph, where it is asked if actin filaments could play a more active role in muscle force generation than simply acting as a rigid lattice. With regard to the modulation of the helical twist, it should be noted that the dramatic change in actin's twist by ADF/cofilin is only ~ 5 degrees (from -167 to -162 degrees), significantly less than what is observed here.

On p. 11 the 12 Å map is referred to as "moderate" resolution. I think that this description is rather dated, and we might refer now to a 5 Å map as having moderate resolution. This map should simply be described as having a poor resolution.

Drummond, D.R., Peckham, M., Sparrow, J.C., and White, D.C. (1990). Alteration in crossbridge kinetics caused by mutations in actin. *Nature* 348, 440-442.

Goldman, Y.E., and Huxley, A.F. (1994). Actin compliance: are you pulling my chain? *Biophysical Journal* 67, 2131-2133.

Prochniewicz, E., Katayama, E., Yanagida, T., and Thomas, D.D. (1993). Cooperativity in F-actin: chemical modifications of actin monomers affect the functional interactions of myosin with unmodified monomers in the same actin filament. *Biophysical Journal* 65, 113-123.

Prochniewicz, E., and Yanagida, T. (1990). Inhibition of sliding movement of F-actin by crosslinking emphasizes the role of actin structure in the mechanism of motility. *J. Mol. Biol.* 216, 761-772.

Schwyter, D.H., Kron, S.J., Toyoshima, Y.Y., Spudich, J.A., and Reisler, E. (1990). Subtilisin cleavage of actin inhibits *In vitro* sliding movement of actin filaments over myosin. *J. Cell Biol.* 111, 465-470.

Referee #4

(Remarks to the Author)

In this work, Carl, Reynolds, and coworkers report on structural studies of actin filaments under force generated by myosin motors, and implications for actin binding protein recruitment. I have been aware of this work for some time due to presentations from Prof. Alushin and his trainees and in general feel it to be novel and significant. I'm not aware of other studies where high resolution structures of proteins under force can be solved, so that would constitute a valuable methodological advance for the rapidly growing field of mechanobiology.

I was asked to comment specifically on the computational model in this work. To the best of my ability to judge, the model presented is appropriate and has been treated correctly. The code to parameterize and run the model using the EspressoMD software is available in the GitHub for this paper, and it seems to match what is reported in the paper. (Unfortunately, I didn't have time to run it myself just getting EspressoMD configured on short notice, but I would be happy to verify that before publication for the authors/editors after it is updated as suggested below).

I have a few minor comments on the modeling section -

- 1) The authors refer to the stiffness of filaments computed by Chu and Voth. In that paper, they compare to the stiffness of 44 pN/nm measured in this paper (<https://doi.org/10.1073/pnas.91.26.12962>) so it may be worth also citing these experiments if they are still considered correct (I do not think this 33% difference would alter any conclusions).
- 2) The authors might consider mentioning a CG model for actin we developed (<https://doi.org/10.1016/j.bpj.2017.05.016>) that has similar resolution, and discuss the differences in how it was parameterized.
- 3) In the model software, there are two stages, parameterization at zero temperature and then assessing the properties with different mechanical perturbations at room temperature. As far as I can tell, right now in the GitHub, the two are mixed together so that the simulation script will perform the mechanical perturbations at zero temperature. Perhaps these can be separated into two steps with a little more instruction for performing the first calculation and taking the output to set the parameters for the second one. Also, the four simulations performed in `sim.py` are all either compression or extension, depending on the sign convention, and I guess while separating out these instructions, the authors could give examples of both.
- 4) Given the way that EspressoMD treats units (as in, it does not use units), perhaps the authors could be a bit more explicit with how they ascribe the force units to be piconewtons, either in the manuscript, readme for the code, or both. It may follow directly from the extensional stiffness calibration but I'm not positive. On a perhaps related note, table S2 reports `k_l` axial

and lateral in pN/nm but perhaps those should be kJ/mol/nm² if all the other angular springs are in kJ (per mol) (because I observe that 265 and 53 appear in the code without unit conversion).

5) In the methods it says "The dynamics of an actin filament under force were simulated by fixing the initial five subunits and applying pulling or pushing force on the terminal five subunits (across both strands)." In the code, they have set up pulling on both strands like this, and also on a single strand. Given this sentence in the methods, it seems the manuscript only reports data where both strands are pulled and released, but I want to confirm that.

6) I also think that the frequency of transient pulling/pushing on the filament is not listed in the manuscript, even though this might just be in simulation units. Moreover, is there a dependence on the wavelength of coiling versus the frequency of transient pulling?

7) It is not clear from the manuscript how many times the authors repeated the simulations in Fig 2f. While I don't think there is any need for error bars there since these are just representative images, if they did replicates then it would be great if they could specifically mention that the behavior happens every time, most of the time etc for however many repeats. Given Fig 2g, it seems they did indeed repeat the apply/release simulations, but I don't see the number of replicates for that histogram.

-
Glen Hocky
Assistant Professor
New York University

(Remarks on code availability)
See above

Version 1:

Reviewer comments:

Referee #1

(Remarks to the Author)

The revised version is very significantly improved. Important new data have been added, the text has been rewritten and some conclusions have been revised. In particular, I find that the description of, and the proposed explanation for the oscillatory filament shapes that occur when motors of both polarities exert forces, now make more sense and are more convincing. The simulations have also been improved, and are now done in more realistic conditions.

The authors have addressed nearly all my points, and I recommend that their work be published provided that they address my remaining concern:

I am confused by the strategy followed to ensure that all the biotinylated actin is near the pointed end, i.e. what the authors call "filament annealing refractory conditions." In the methods, on page 45, they indicate that biotinylated actin is added in the second polymerization step, with rhodamine actin (line 1135). Maybe I missed something, but what would be the point of having biotinylated actin in both polymerization steps? Assuming this is a mistake, and that no biotinylated actin was used in the second step, I still find the strategy questionable because there is certainly plenty of annealing taking place during the one hour incubation of the biotinylated seeds with unbiotinylated G-actin (the second polymerization step, before capping). The preceding, shearing step should have created very short biotinylated segments that would then be incorporated all along the filament. Also, at the end of the first polymerization step, there should still be significant amounts of monomeric actin (even if steady-state is reached), including biotinylated monomers that would then copolymerize with the actin added in the second polymerization step. It is thus quite likely that the filaments contain several biotinylated actin segments, that would be too small to be detected in images like the ones shown in Extended Data Fig 2e-f.

I hope the authors will be able to address this remaining point, which is a bit technical but quite important, because their work should be an important contribution to the field.

Related, minor points: With the indicated actin concentrations, the biotin fraction should be 20% not 25% (line 1130) ; In Extended Data Fig 2e-f the colors used are counter-intuitive because, according to the methods, the biotin-seeds are made with ATTO-488 actin and should be represented green ; The legend does not explain what the arrowheads in Extended Data Fig 2f represent.

Referee #2

(Remarks to the Author)

The authors have performed a substantial amount of work, particularly in relation to the simulations, to address earlier criticisms, and they should be commended for this effort. Technically, the study is a tour de force, employing a wide range of

cutting-edge methodologies. Conceptually, the paper advances the claim that forces exerted by myosin motors on actin filaments generate regions of supercoiling that are subsequently recognised by mechanosensitive proteins, exemplified here by α -catenin. Similar supercoils (or sinusoidal regions) are reported in cells at actin filament sites under load, identified by zyxin enrichment.

Molecular dynamics simulations are presented in support of the findings; however, these appear to be flexibly adapted to the proposed model rather than providing independent predictive insight. Setting this aside, the central question is whether myosin-dependent supercoiling is indeed the mechanism by which mechanosensitive proteins recognise filaments in cells. If the answer were yes, this would represent a major advance, fully warranting publication in Nature. The difficulty is that I am not convinced the evidence, based on limited data that barely cross the threshold of significance, is sufficiently strong to sustain such a conclusion.

That actin filaments can undergo supercoiling as a result of myosin compressive forces is plausible, and merits publication in a more specialised journal, yet I find the biological significance of this observation overstated. In crowded cellular environments such as adhesions, actin filaments are frequently bundled and bound by multiple proteins, including some forty tropomyosin isoforms. Under these conditions, it is difficult to see how myosin-dependent supercoiling would be required for proteins such as α -catenin, zyxin, or other LIM-domain family members to bind. Structural changes in inter-subunit bonds of the actin filament under tension alone, without invoking supercoiling, could adequately explain the recruitment of tension-sensitive proteins.

Finally, I must note that this manuscript is difficult to read, not due to the complexity of the data, but to the convoluted style of writing. Once the text is disentangled from the results, it becomes clear that the substance presented does not, in my view, justify publication in Nature. Papers published in Nature rapidly come to be regarded as definitive, and I am not persuaded this study merits that level of endorsement.

Referee #3

(Remarks to the Author)

This is a monumental body of work being reported in a single paper, so it is by necessity quite dense. While I had enthusiasm for the initial manuscript, this revised paper is greatly improved compared to the initial submission. All of my concerns have been adequately addressed. In cases where conclusions are not firmly established or speculative, the authors clearly indicate that this is the case. I found only one mistake (p.3-4):

“Consistently, chemical modifications of F-actin have been reported which are permissive for myosin binding and ATPase activation but refractory to motor force production, implying conformational reciprocity between the filament and myosin’s mechanochemical cycle⁵⁵⁻⁵⁸.”

But ref. 57 involves mutations in actin, not chemical modifications!

Referee #4

(Remarks to the Author)

I feel the authors have done their best to respond to the criticisms/comments with substantial extra effort, and I still feel that the new methodology presented in this work is a major advance worthy of publication in Nature.

(Remarks on code availability)

I did not have the opportunity to run the code myself, but it looks appropriate to the problem and to do what is claimed in the manuscript.

Version 2:

Reviewer comments:

Referee #1

(Remarks to the Author)

I am satisfied with the latest revisions. The authors have addressed my concerns and I recommend publication of the article.

Referee #3

(Remarks to the Author)

The authors have done an excellent job of responding to the reviewers, and the paper has been revised accordingly. Reviewer #2 suggested that tension alone, without supercoiling, can induce structural changes in actin which would modulate the affinity of certain actin-binding proteins. It is certainly true that within highly ordered bundles of actin, such as the I-band in striated muscle, supercoiling of actin would not be possible, and the same may be true in parts of the cytoskeleton. But the forces and conformational changes in actin that induce supercoiling would induce structural changes, propagated cooperatively along filaments, that would modulate the affinity of actin-binding proteins even if supercoiling was itself suppressed. The authors might need to only add one or two sentence to the paper to clarify this and potentially satisfy Reviewer #2 (as well as generalize the conclusions of the paper).

We thank the reviewers for their thoughtful, thorough, and incisive comments. While all referees highlighted the novelty and significance of our studies, a number of substantive concerns were identified. We have performed extensive additional experiments, quantitative analysis, and simulations which we believe address essentially all of these concerns. Major new additions / revisions include:

- 1) Quantification of cellular tomography data, which supports enrichment of oscillatory F-actin domains (now termed “sinusoidal regions” at the suggestion of Reviewer #3) in cellular regions marked by zyxin.
- 2) In response to Reviewer #1, implementation of a new biochemical procedure to prepare F-actin from biotinylated seeds which is refractory to annealing between filament ends. In cryo-EM data, we continue to observe superhelical F-actin (now referred to as “supercoil” F-actin in response to a comment by Reviewer #3) using this procedure in the presence of myosin-6 forces. Nevertheless, we acknowledge that this likely does not represent solely a “tension” condition, as discussed in detail in the rebuttal below. Broadly, we have clarified that we believe compressive forces (which are likely to present in all experimental conditions we tested) are responsible for supercoil formation.
- 3) In response to Reviewers #1 and #2, implementation of an improved procedure for quantifying F-actin curvature oscillations in cryo-EM images, which reveals a much clearer difference between -ATP and +ATP conditions. This is consistent with visual inspection, which supports increased supercoil F-actin in the presence of active force generation.
- 4) In response to Reviewers #1, #3, and #4, extensive additional and improved simulations, which include viscous dissipation, conditions which mimic asynchronously firing motors, filament buckling and severing events, and application of torque, as well as modulation of F-actin’s helical parameters. We continue to observe supercoils in the presence of viscous dissipation, although in our improved simulation conditions their formation is no longer observed upon the release of tension. Supercoils are additionally formed in the presence of asynchronous motors and filament buckling / severing events. Applied torque alone does not produce supercoils in our simulations, suggesting that this type of force is not sufficient to explain our observations. This supports our mechanistic interpretation that F-actin’s helical architecture specifically transduces lengthwise compressive forces imposed by motors into torsional rearrangements.

We have also adjusted the language in the paper to avoid overinterpretation and clarify what confident statements can be made from our data, e.g. the types of forces produced by myosins which we believe lead to supercoil F-actin, as well as the importance of resolution limits when interpreting our cryo-EM reconstructions as noted by Reviewer #2. We have also corrected minor typos / errors we identified while revising the text. We believe this has substantially strengthened the paper, and we hope the referees and editors will now find our manuscript suitable in principle for acceptance and publication in *Nature*. Our detailed responses follow below.

Referees' comments:

Referee #1 (Remarks to the Author):

This manuscript by Carl et al. addresses a longstanding and very important question: how can mechanical forces applied to actin filaments, primarily by myosin motors in cells, change the conformation of the filaments and their ability to bind regulatory proteins? This question is of

paramount importance since it appears that mechanical forces induce biochemical changes, thereby regulating the architecture and the dynamics of the actin cytoskeleton. However, until now, structural evidence of the putative mechanically-induced conformational changes is lacking, hindering our ability to understand this key mechanism. The main reason for this lack of data is technical, as there is no straightforward way to apply controlled mechanical forces to a large population of filaments on an electron microscope grid.

We thank the reviewer for appreciating the biological significance of our studies and the technical difficulty in performing structural studies in the presence of active motor forces.

Here, the authors propose a way to do so, building on a previous assay developed in their lab, using two types of myosins motors. These in vitro experiments are complemented by observations in cells and by numerical simulations. They propose that compression and tension can both lead the filaments to adopt a new, supercoiled conformation.

This paper could be a landmark, and thus it would be terrible if flawed conclusions were published and were then to mislead scientists for years to come. Currently, I am afraid there is such a risk. I find the work exciting, and I hope that the authors will be able to alleviate my concerns.

My main problem is that I am not convinced by one of the main claims of the paper, that filaments under tension adopt a superhelical conformation, for the following reasons.

We acknowledge that the presentation in our initial submission (particularly the simulation section) may have given the impression that we were making this claim. In fact, we do not believe that “tension” in the strict mechanical sense causes F-actin to adopt the supercoil conformation. Our results are most consistent with compression directly causing this transition. In our initial simulations, tension only elicited supercoils upon elastic recoil of the filaments when tension was released, effectively resulting in compression. In revised simulations including viscous dissipation (see below), this is no longer observed, and only compressive conditions result in supercoils. We acknowledge based on the reviewer’s concerns and our new simulations that the “pull and release” mechanism included in our initial submission likely does not explain our observation of supercoils in the pointed end-directed force condition (biotin seeds and myosin-6). In addition to performing additional experiments to address the reviewer’s concerns about annealing (detailed below), we have also adjusted the text to highlight a more likely explanation for how supercoils can form in this condition, wherein the asynchronous activity of multiple myosin-6 motors can compress the filament (new Extended Data Fig. 4g, lines 241-250).

Fundamentally, the central claim of our paper is that “myosin forces” produce supercoil F-actin. It remains true (and we believe, quite striking), that all conditions we employed to reconstitute myosin forces produced supercoil F-actin, which we also observe in cells. While the field frequently uses the word “tension” as a catch-all proxy for cytoskeletal forces, our data suggest that, due to the stochastic nature of motors, physiologically relevant compressive forces are essentially always present, in a manner that is not necessarily immediately intuitive. We believe this is a key take-home message of the paper which we now explicitly emphasize in the text (including in the Summary, lines 36-38).

- First, it is not clear to me that filaments are only under tension in the “pointed end” experiments. Uncontrolled filament sticking to the surface could cause the pointed end motors to generate

compressive forces. Control experiments should be performed to show that this is not the case. Also, based on the way the filaments are prepared (page 42) it seems likely that the pointed ends of some biotinylated segments anneal with barbed ends of non-biotinylated segments. This would also result in portions of filaments being compressed in the experiment with myosin VI. I think the experiments could be done differently (for example, capping the pointed ends of the biotinylated seeds) to ensure that this problematic annealing does not happen. Also, it would be informative to quantify the occurrence of the different types of filament segments (straight vs. oscillatory) in each experiment.

Based on the reviewer's suggestion, we implemented an improved procedure using profilin-G actin to heavily bias the system towards barbed end elongation, as well as capping the filaments to prevent annealing (Methods lines 1127-1204; Extended Data Fig. 2e,f). We continue to observe sinusoidal filament regions in this condition (Extended Data Fig. 2g). We also implemented a per full-filament quantification procedure (Methods lines 1238-1261; Extended Data Fig. 2b,h), akin to that we use to analyze our *in vitro* tomography data (Fig. 2c,d). Due to the limitations of our machine-learning based picker, which like all automated pickers has difficulties with filament crossovers and junk in the micrographs, we manually traced filaments for analysis, as we were concerned about potential artifacts and believe this is the maximally rigorous approach available at this time. As this is a laborious procedure, we were not able to apply it to all the datasets in the paper. Nevertheless, we see a clear increase in sinusoidal filament regions (indicated by significant increases in both filament curvature and the peak-to-peak amplitude of curvature oscillations) in the presence of ATP for both the dual motor condition (Extended Data Fig. 2b), and the "pointed end force" condition, performed with the new biochemical approach described above (Extended Data Fig. 2h). This specifically manifests as substantial increases in the upper quartile high curvature / amplitude "tails" of the distributions, likely representing the appearance of sinusoidal filament region subpopulations against the backdrop of the majority straight filament population. While we did observe apparent differences between the dual motor and pointed end conditions (specifically a higher peak-to-peak amplitude in the dual motor condition in the presence of ATP), a caveat of our analysis is that it is sensitive to systematic error such as inaccuracies in pixel size calibration. We thus did not comment on this in the manuscript, instead focusing on internal comparisons within each condition (lines 139-141; 159-164).

- Second, I do not find the simulations convincing. Based on the movies, it appears that the filaments are simply able to fluctuate thermally once they are released. A control filament, with no mechanical force applied (neither tension nor compression, neither constant nor released) is required to clarify this.

The simulations included in our initial submission were performed in vacuum (i.e. lacking viscosity from solvent) at zero temperature, and thus thermal fluctuations were not present. The continued oscillations the reviewer notes were harmonic in nature. We acknowledge this aspect of the simulations was not physically realistic.

In our revision, we have incorporated implicit solvent featuring the friction coefficient of water, and performed simulations with thermal fluctuations at room temperature (298 K), conditions which mirror our experiments (Methods lines 1351-1539). As this is more realistic, we decided to replace our initial simulations, as well as perform additional simulations to address reviewer concerns (detailed below).

We performed control simulations of filaments experiencing thermal fluctuations in the absence of externally applied force for three different conditions: 1) with both ends free, resembling an unconstrained filament tumbling in solution; 2) with one end fixed, which resembles a filament with one end sticking to the carbon film and the other extending to the hole, providing a reference for release after tension/compression; 3) with both ends fixed, which resembles a filament spanning across a hole engaged by motors and / or the carbon film on both sides, providing a reference for constant tension/compression. In all three conditions, we observe minor thermal fluctuations of the filaments, as anticipated, but we do not observe the formation of clear sinusoidal filament regions. Quantification of the 3D peak-to-peak amplitude reveals a median of less than 12 nm for all three conditions, which intuitively decreases as the level of constraint on the filament ends increases (Fig. 2f). Using a 3D peak-to-peak amplitude cutoff of 16 nm (roughly twice the diameter of the filament) to designate sinusoidal filament regions, less than 10% of the simulation frames contain such regions in any of the three conditions, with the filament with both ends fixed producing virtually none (Extended Data Fig. 3e).

When applying force, our results with the improved simulation framework are essentially consistent with those in our initial submission. The sole exception is that the prevalence of supercoils in the “pull and release” condition is greatly reduced, such that it is statistically indistinguishable from the thermal fluctuation controls (Fig. 2h and Extended Data Fig. 3e). Upon consideration, this observation is commensurate with the low Reynolds number conditions present in both our experiments and simulations, where viscosity will dominate over the inertia of any elastic recoil in the filament. Given this result, the concerns raised about the likely presence of compressive forces in the pointed end-directed force condition due to asynchronous myosin power cycles (with which we agree, discussed below), as well the discussion above about the tendency to use the term “tension” as a catchall placeholder for force, we have adjusted the language/ included new text in the paper to make more explicitly clear that we believe compression is likely the dominant mechanism of supercoil formation across all conditions we tested (lines 241-250).

For the filament under constant compression, I would expect buckling to readily occur since the filament is approximately 1 μm long and a 25 pN compressive force is applied. Is that what we are observing? If that simulation were to run a bit longer, it seems that the filament would continue to collapse, wouldn't it?

In the simulations included in our initial submission, it was not possible to break bonds, which we believe is what the reviewer is referring to by “buckling and collapse”. We have now implemented this by incorporating critical bond lengths (Methods lines 1456-1474) estimated using the helical rise and twist measured from the cryo-EM density map of supercoil F-actin (Fig. 3b). We do indeed observe filament fragmentation (Extended Data Fig. 4b). However, we find that supercoils form prior to this collapse (Supplementary Movie 6, Extended Data Fig. 4b, compression with bond breaking), and can persist on broken segments, consistent with our experimental interpretation that they can persist for some time after mechanical breakage (Extended Data Fig. 2b, lines 141-147; 217-219).

Based on these results, we used the simulation frame where filament rupture occurs (which differed between conditions) to define the duration of force application when performing detailed quantification of supercoil properties (Fig. 2f-h; Extended Data Fig. 3e-h). We speculate this is why we now observe a relatively uniform distribution of pitches (Extended Data Fig. 3f, equivalent to wavelength in our 2D analyses), across conditions, as the duration of force application was chosen arbitrarily in our initial submission. The pitch values observed in our simulations do deviate from

the experimentally measured wavelengths (Fig. 1g; Extended Data Fig. 2a). This suggests that the critical bond lengths we selected may not be physically precise, but we believe there is not currently a better / more rigorous way of defining this parameter. This discrepancy is noted in the text (lines 223-227).

- Third, regardless of the issues mentioned in the previous paragraphs, I find it hard to compare the simulations with what is happening in the in vitro experiment and in cells. In the simulations, in order to observe an oscillatory shape when applied tension, a 25 pN force is applied and released. This would correspond to several motors (5 motors, according to the authors' estimation, page 49) pulling on the motor and simultaneously letting go. I have a hard time imagining this taking place in the experiments. Also, this would not correspond to the filaments that are observed with both ends attached to the surface. Rather, I expect that, in the experiments, filaments are pulled on by several motors, with non-synchronized cycles. The applied tension would certainly fluctuate over time around an average value, but not drop rapidly to zero as it is currently done in the simulations.

The reviewer is correct that multiple motors are likely operating asynchronously on the filaments in our experiments. We have now implemented a random motor-firing scheme using our simulation framework, in which five subunits were randomly selected from the lower half of the filament, with each subjected to a 6 pN force for a randomly assigned duration. We still observe the formation of supercoils in the compressive condition, but not in the tensile condition (Supplementary Movie 6, Extended Data Fig. 4d), consistent with our other simulations (Fig. 2e; Extended Data Fig. 4c).

Other points:

- The motors probably apply a significant mechanical torque to the filaments. This should be an important mechanical contribution that could lead to supercoiling. Why is this aspect ignored?

As measured in "twirling" assays by Yale Goldman and colleagues, where filaments sparsely labeled with fluorescent phalloidin were observed to rotate around their filament axes while translocating in gliding assays using polarization TIRF microscopy, both myosin-5 and myosin-6 do apply slight torques (PMIDs: 18931255, 18158894; refs. 70,71). This is, however, quite modest, as the filaments need to translocate ~1.5 microns in the presence of either motor to undergo a full rotation, consistent with only 8.5 degrees of rotation per crossover length (as an approximate proxy for a powerstroke).

Nevertheless, we explicitly examined the effects of torque in simulations (Supplementary Movie 6, Extended Data Fig. 4e). We find that small torques are quickly balanced and resisted by counter torque built up internally through filament twisting without evoking substantial changes to filament curvature, whereas large torques result in random looping of the filament that does not resemble the supercoils in our cryo-EM experiments or compression simulations. Applying torques with bond breaking enabled quickly results in filament fragmentation, and supercoils are not observed in any of the torque conditions we examined (Supplementary Movie 6, Extended Data Fig. 4e). Upon consideration, this is consistent with prior work on cofilin (which binds in cooperative patches and modulates the helical twist of F-actin), which to our knowledge has not been reported to result in supercoil formation.

We believe this result supports the overall interpretation that effective compression imposed by myosin activity is transduced into supercoiling by the helical architecture of F-actin, without the

requirement for external torque. This is now discussed in the manuscript (lines 251-262)

- The filaments in EM micrographs exhibit straight segments coexisting with oscillatory segments, with seemingly sharp transitions between the two. How can this be understood? The simulations (movie and Fig 2f) do not appear to show anything like this.

We believe this transition is due to viscous dissipation of force propagation along filaments, as in our improved simulations where this is implemented we now see decay of the oscillations with increasing distance from the site of force application (Fig. 2e, Supplementary Movie 6). This is now stated in the manuscript (lines 209-213).

- I am not sure to understand the authors' point about the elliptical cross-section of the superhelical filaments. The way I see it, superhelical filaments with a circular cross-section would be deformed (because of thermal fluctuations, at least) and their cross-sections would then appear elliptical in the experiments. Aligning these ellipses (e.g. Fig 2c) will of course produce an average elliptical shape, but this does not indicate that the individual spirals intrinsically have elliptical cross-sections.

We see elliptical cross sections in both tomograms of individual supercoils which are not averaged (Fig. 2c), regardless of their orientation in the ice (Extended Data Fig. 2i), as well as simulations (with thermal fluctuations, Fig. 2e; Extended Data Fig. 4c, and without [our original submission]). We believe this is an additional datapoint highlighting the congruence between our experimental observations and simulations.

Conversely, when many filaments were averaged in our single particle studies, their presumably individual elliptical cross sections averaged out to give a circular cross-section (Fig. 3a).

In response to the comments of Reviewer #3, we systematically varied the helical parameters of F-actin in simulations. While these variations did produce changes in the eccentricity, consistent with the importance of F-actin's helical properties for this phenomenon, there was not a clear trend, and all filament architectures maintained substantial ellipticity (Extended Data Fig. 3i).

We thus turned to an alternative hypothesis, that ellipticity may be linked to the dynamics of filament remodeling during compressive force application. Consistently, in our compression simulations we observe that ellipticity is always maximal at the beginning of force application, either remaining very high in 2 out of 5 simulations or decreasing in the other 3 (Extended Data Fig. 4h). This is consistent with filament segments that will supercoil initially undergoing uniplanar buckling upon compression, transitioning to supercoiling as the end-to-end distance of the filament continues to decrease. This trajectory is visible when a supercoiling segment is viewed end-on in simulations (Supplementary Video 6). We have updated the manuscript with this explanation (lines 278-289)

- I could not find the criteria for classifying the different segments of filaments as straight or oscillatory (Fig 1). What were these criteria? Was this classification automated or done manually?

All of the quantification of *in vitro* data in Fig. 1 in our original submission was performed manually, which we now describe more explicitly in the Methods (lines 1123-1125). We acknowledge this as a limitation. As noted above, we have now performed a more thorough procedural quantification (still

requiring manual tracing of all filaments, regardless of curvature, present in a micrograph). This clearly shows that myosin activity in the presence of ATP produces more sinusoidal filament regions (Extended Data Fig. 2b,h), indicated by significant increases in both filament curvature and the peak-to-peak amplitude of curvature oscillations.

We have now also quantified cellular tomography data as requested by Reviewer #2 (Fig. 1c; Extended Data Fig. 1d-f), detailed in our response to Reviewer #2.

Minor :

- I find the terminology "pointed end" and "barbed end" (Fig 1c and elsewhere) confusing because it refers to the directionality of the motors, while in my opinion it would make more sense to refer to the direction of the applied force, which is in the opposite direction.

We chose this nomenclature to emphasize that the forces in our experiments are complex forces produced by motor proteins. Other terminology such as "tension" and "compression" would, in our opinion, be misleading. Reversing the terminology would, also, in our opinion, increase confusion, and we have elected to maintain the current language in this instance.

- In Fig 3a, indicating the subunit indices would be useful. It would help to see the correspondance with the other panels.

We thank the reviewer for this excellent suggestion. We have indicated the subunit indices in Fig. 3a, and we have also done so in Fig. 4a, which has a parallel presentation.

- The color choices for rises and twists in Fig 3b and Fig 4c are a bit confusing, because they are the same colors as for the two filament strands (for example, light blue correspond to the rise from light to dark blue, while dark blue corresponds to the rise from dark to light blue). Maybe a different color code would help?

Respectfully, we believe the definitions of the colors are clear, as indicated by the color of the arrows on the cartoons in Fig. 3b. Alternatives (such as inverting the colors, etc.), would, we believe, be even more confusing.

Referee #2 (Remarks to the Author):

This study investigates how forces generated by myosin motors influence the structure of actin filaments and their ability to bind tension-sensing proteins such as zyxin and α -catenin. These findings could reveal more broadly how mechanical forces affect filament interactions with actin-binding proteins (ABPs) and contribute to signalling events.

The reader is expected to draw two main conclusions from this paper. First, actin filaments under tension exhibit a regular wavy (oscillatory) structure.

As noted above in our response to Reviewer #1, we do not believe that "tension" per se produces supercoil F-actin, but rather compression, which can occur through multiple mechanisms in the presence of myosin motors. We have clarified this point throughout the text, including in the Summary (lines 36-38).

Second, this wavy conformation enhances the binding of specific ABPs that respond to tension. If these findings are broadly applicable (generally true), they would represent an important and novel contribution to the field, meriting publication in a high-impact journal like Nature.

We thank the reviewer for noting the potential significance of our studies and their suitability, if sound, for publication in *Nature*. We hope that the reviewer will find our substantially revised manuscript, as well as our responses below, compelling.

Conversely, if these observations are isolated or incongruent with the authors' conclusions, they would still be of interest but might not have the broader impact required for such a prestigious venue.

My current evaluation is that, although the work is technically sound and creative, there are significant concerns about the validity of the interpretations and the overall significance of the observations.

Major Criticisms:

1. Statistical Significance of the Cellular Data: Figure 1 suggests that actin stress fibres under tension, marked by increased zyxin decoration, display wavy filaments. Is this observation statistically significant? The statement "Many filaments in both of these types of networks exhibited substantial curvature" on Page 4 is not quantitative. A thorough statistical analysis is needed to determine whether this wavy behaviour is observed in 0.01% or 50% of zyxin-decorated filaments, using the already available cryo-ET data.

Despite our best efforts, we were not able to implement an automated procedure which could reliably trace all of the individual actin filaments in our cellular tomograms, having tried to both adapt our own machine learning-based segmentation approaches which we have previously applied to *in vitro* tomograms (as described in Gong, Reynolds, et al., NSMB 2025, PMID: 39833469) and the cylinder-reference based approaches in Amira previously described by the Baumeister lab (e.g. in Jasnin et al., Nat. Comm. 2022, PMID: 35789161). We believe the primary limitation is the high filament density of the networks in our tomograms, which make it particularly challenging to consistently track them over the necessary length scale.

Nevertheless, we appreciate the reviewer's point about quantification. Since the major claims made from these data are 1) That sinusoidal filament regions exist in cells and 2) That they are associated with subcellular regions of high traction force marked by zyxin, we have focused on providing quantitative support for claim 2. We thus performed a blinded analysis of zyxin high vs. zyxin low tomograms, where zyxin level was assessed and sinusoidal filament regions were manually picked by different members of our team. The number of sinusoidal filament regions were then quantified (Methods lines 974-986).

As described in the text (lines 115-118) and Fig. 1c / Extended Data Fig. 1d-f, we observe significantly more sinusoidal filament regions in zyxin-high subcellular areas, consistent with our claim that they are associated with high local traction forces. We also performed Dragonfly segmentation to estimate the total amount of F-actin per tomogram (Extended Data Fig. 1e). We did not observe a significant difference in F-actin abundance between zyxin high and zyxin low subcellular areas, arguing that the increased prevalence of sinusoidal filament regions in the zyxin high subcellular areas is not due to systematic differences in the amount of F-actin between zyxin low and zyxin high subcellular areas.

To the Reviewer's point about prevalence, we observed on average ~12 oscillatory domains per zyxin-high tomogram, with a maximum of 22. As these tomograms visually feature hundreds of actin filaments, our quantification suggests that sinusoidal filament regions indeed appear on a subpopulation of actin filaments in these tomograms, on the order of a few percent. We speculate these segments could be suitable for initiating signaling processes by recruiting force-sensitive ABPs, but they likely do not represent a global transition that would e.g. compromise the local mechanical integrity of the cytoskeletal network. However, as future studies will be required to assign precise functions to sinusoidal filament regions in specific subcellular processes, we have not discussed this aspect extensively in the manuscript text.

Additionally, Peter Gunning's work shows that most actin filaments in stress fibres are decorated with tropomyosin. How does this reconcile with uniform zyxin decoration and the presence of waves? Are there mixtures of tropomyosin-decorated straight filaments and zyxin-decorated wavy filaments in these stress fibres? A deeper cryo-ET characterisation of these stress fibres in cells might provide the most compelling and convincing narrative for publication in Nature.

First, we respectfully believe the Reviewer is overinterpreting our cryo-fluorescence data. These data do show that zyxin is enriched in the subcellular areas we target at the diffraction-limited resolution of fluorescence microscopy relative to neighboring subcellular areas. However, this does not by any means demonstrate that particular actin filaments in that subcellular area, whether or not they feature sinusoidal filament regions, are specifically bound by zyxin. We were thus careful not to state that our data provides evidence that zyxin binds sinusoidal filament regions. This is a reasonable hypothesis based on our findings, but it would require an extensive (*in vitro*) experimental campaign to visualize the force-mediated zyxin-F-actin complex, which is beyond the scope of the current study.

Furthermore, it is important to note that F-actin is not zyxin's only binding partner in adhesions. Consistently, our previous work (Sun et al. Dev Cell 2020, PMID: 33058779, ref. 39) showed that mutations which disrupt zyxin's binding to actin under force do not completely eliminate the protein's adhesion localization. Zyxin is thus most appropriately viewed as a marker of traction forces in these experiments, agnostic to its precise mechanism of localization. We have expanded upon these points in a new "Limitations" section in the text (lines 459-474).

Second, we appreciate the Reviewer's point about the potential role of tropomyosin. Our cellular cryo-ET data were collected at relatively low magnification to facilitate capturing the maximum field-of-view per tomogram, an important consideration due to the uncertainty in targeting via cryo-fluorescence (a major reason for the modest number of cellular tomograms presented in the paper). Our current data thus are not suitable for subtomogram averaging studies to assess tropomyosin decoration. Moreover, the total number of sinusoidal filament regions we observed in cells (on the order of ~100 in 14 tomograms) is insufficient to perform reliable averaging / classification. We thus unfortunately were not able to address this particular request from the Reviewer in the current work. We nevertheless believe the influence of tropomyosin on supercoil formation, and its intersection with other actin-binding proteins, is a very important topic for future studies, including *in vitro*. We have included a brief discussion of this in "Limitations" section (lines 462-465).

2. Statistical Significance of the Cryo-EM Data: Similar concerns apply to the cryo-EM data. The technical challenge of freezing filaments under tension in cryo-grids is acknowledged, and the authors should be commended for this achievement.

We thank the reviewer for appreciating the technical difficulty of capturing F-actin for cryo-EM studies in the presence of active myosin force generation. As noted above, we caution that this likely does not represent “tension” in the strict mechanics sense, which is why we have been careful to avoid this term when discussing the cryo-EM data in the manuscript.

However, critical questions remain, such as the percentage of filaments showing wavy conformations as a function of myosin (V and VI) concentration and/or \pm ATP. The \pm ATP superhelix quantification shown in Fig. S3c suggests these appear in both \pm ATP and are only moderately more prevalent +ATP. Perhaps this could be moved to the main text, along with a different quantification, i.e., the absolute number of filaments showing vs. not showing superhelices per micrograph. However, it is concerning that superhelices are observed in the -ATP condition, because: a) are superhelices truly due to myosin-generated tension? b) The majority of the superhelices characterised in the +ATP condition are independent of myosin activity, as they are also observed in -ATP conditions.

Due to the technically demanding nature of our experiments as noted by the Reviewer above and the potential number of conditions to be tested, it was not feasible for us to perform myosin concentration series. Nevertheless, as also noted by Reviewer #1, we do appreciate the point about quantitatively showing that formation of sinusoidal filament regions is dependent on motor activity (e.g. the presence of ATP).

As noted in our response to Reviewer #1, we implemented a quantification procedure similar to that which we used for our *in vitro* tomograms (Fig. 2c). This required manual tracing to capture full filaments due to limitations of our machine-learning based picker, followed by automated analysis of the traces (Methods lines 1238-1261).

We first applied this procedure to our initial dual motor -/+ ATP datasets presented in Fig. 1e, which were collected at low magnification on a TF20 microscope / CCD camera during the earliest phase of this long-running project. We see a clear increase in sinusoidal filament regions (indicated by significant increases in both filament curvature and the peak-to-peak amplitude of curvature oscillations, Extended Data Fig. 2b) in the presence of ATP. This specifically manifests as substantial increases in the upper quartile high curvature / amplitude “tails” of the distributions, likely representing the appearance of sinusoidal filament region subpopulations against the backdrop of the majority straight filament population.

Nevertheless, some sinusoidal filament regions are present in the -ATP condition, whose abundance we found varied from preparation to preparation. We reasoned this could be due to residual ATP present in the sample, as myosin motors must be stored in the presence of ATP for stability. We thus performed an additional control experiment where we included apyrase in the -ATP condition (which hydrolyzes ATP / ADP to AMP), using improved sample preparation conditions for capturing the “pointed end force” condition (using profilin-actin to strongly bias barbed end elongation from biotin-F-actin seeds, as well as including capping protein to prevent filament annealing, Methods lines 1127-1204). Analyzing cryo-EM collected at high magnification using a

modern direct electron detector, we once again saw a clear and statistically significant increase in filament curvature and the amplitude of curvature oscillations in the +ATP condition vs. apyrase (Extended Data Fig. 2h). Moreover, while sinusoidal filament regions were not completely eliminated by apyrase treatment, they were, by visual inspection, quite rare.

As noted above in our response to Reviewer #1, since our datasets were collected on different microscopes with different magnifications and detectors, systematic error in pixel size calibration makes rigorously quantitatively comparing them problematic. Thus, in the manuscript, we restricted ourselves to internal comparisons within each experimental condition +/- ATP. Nevertheless, as these experiments were performed using completely independent sample preparation conditions, data collection schemes, and by different members of our team, we believe this analysis provides strong support for our claim that myosin motor activity promotes sinusoidal filament region formation.

3. Additionally, if myosins are attached to the carbon area, why do all the examples of wavy filaments shown in the paper not extend into this region? Coincidentally, the colour tracing of filaments should extend into the carbon area to further clarify this point.

Respectfully, due to the strong electron scattering by the thick carbon support film relative to F-actin, we cannot reliably trace filaments on the film. We nevertheless have observed examples where oscillatory domains extend onto the film, shown here:

We speculate that the extent a sinusoidal filament region extends on the carbon likely depends on where it is engaged by myosin motors. However, individual motors cannot be resolved on the carbon.

Since we believe the requested analyses cannot be performed rigorously, we have not included them in the revised manuscript.

Furthermore, why are only sections of some filaments wavy? Tension is expected to be propagated along the entire filament length, is it not? Do the waves dissipate with the distance from the carbon support?

We believe that viscous dissipation of force propagation is the most likely explanation for the boundaries between sinusoidal filament regions and straight filament regions. As noted in our response to Reviewer #1, we have performed improved simulations where we now include viscosity of the medium and thermal fluctuations (Fig. 2e-h; Extended Data Figs. 3,4). We now observe decay in oscillation amplitude / modulation of wavelength with distance from the site of force application (Fig. 2e, Supplementary Movie 6). We indeed observe experimentally that sinusoidal filament regions are generally found adjacent to the carbon support, where motors can apply force, and dissipate away from the support, consistent with this idea. This is now discussed in the text (lines 209-213)

Are freezing conditions, ice thickness, or other experimental variables influencing this behaviour?

We have anecdotally observed that sinusoidal filament regions are not present in specimens with very thin ice, potentially due to surface tension rupturing already mechanically strained filaments. This is one aspect (the requirement for thick ice) that limited the resolution of our single-particle studies. However, under the experimental conditions presented in the paper, we do not observe a preferential orientation of supercoils relative to the ice film in our *in vitro* tomograms (Extended Data Fig. 2i), suggesting that the air-water interface is not playing a major role in their formation or properties.

Are there myosin motors in the holes also exerting forces on the filaments?

We do occasionally observe objects in the holes which could represent “escaped” myosins, but they are rare. Consistently, we did not obtain any myosin-decorated classes in our single particle analysis. While this does not mean no myosin bound filament segments are present, it does suggest that they are rare enough that they do not present a meaningful signal that is picked up by averaging.

How are the waves shown in Fig. S2c generated? If filaments are attached to the carbon support at only one end, how is tension maintained and not released? The authors' claim of filament “memory” for deformation, persisting even after tension release (“domains have the capacity to persist after filament breakages,” Page 5), seems questionable. It is more plausible that such breakages occur during freezing.

As noted in our response to Reviewer #1, we have now performed simulations which incorporate filament breakages (Extended Data Fig. 4b). We observe that supercoils can form prior to breakages, and persist after breakages, aligned with our previously speculative explanation for waves on filaments with free ends.

The reviewer is correct that blotting forces could also cause breakages in filaments pre-stressed by motor forces, in addition to the motor forces themselves causing breakages. However, we don't believe this fundamentally alters the explanation we provide. If blotting forces alone robustly produced supercoils, we believe it is likely they would have been reported in previous cryo-EM studies of actin filaments alone by ourselves or others. To our knowledge no such report is present in the literature at this time.

Furthermore, it is unclear why both myosins produce similar waves when analysed individually. Myosin VI, moving towards the pointed end, should generate pulling tension, while myosin V would likely buckle the filament, but regular wavelength buckles are unexpected. Could the authors provide a cartoon illustration to clarify all of these unexpected observations? The simulations attempt to address some of these questions, but what is really lacking is experimental evidence in support of these unusual observations and conclusions.

We realize that the results with individual motors are counterintuitive, and we have done our best to clarify this aspect of the study. The summary is that any condition featuring stochastically operating populations of motors which undergo cycles of binding, powerstroke, and release can cause effective compression of the filament, even under conditions where the directionality of the motor should intuitively impose tension.

While the mechanism producing regular wavelengths is difficult to probe experimentally, in response to Reviewer #3 we performed simulations where we varied the helical parameters of F-actin, which correspondingly varied the wavelength / amplitudes of supercoils (Extended Data Fig. 3g). This argues that these properties of myosin-force produced supercoils are specified by the inherent helicity of F-actin, with the helical rise being the dominant parameter determining the architecture of supercoils.

The most difficult case to understand, as noted by the reviewer, is myosin-6 alone (the "pointed end force" condition). The pull and release mechanism we proposed in our initial submission is one potential explanation, but new simulations performed in the presence of viscous dissipation do not support this as a predominant mechanism (Extended Data Fig. 4c). In response to Reviewer #1's concerns, we performed additional experiments focusing on this condition, optimizing our protocol to avoid filament annealing events which could result in positioning biotinylated anchoring segments in the middle of filaments, where myosin-6 would be anticipated to produce either tension or compression depending on the position of the motor relative to the anchor (Extended Data Fig. 2e-h). We nevertheless observe sinusoidal filament regions in the optimized condition, suggesting stable plus-end tethering is not required for their formation with myosin-6. We therefore believe the most likely explanation is a mechanism which occurs at the level of multiple motors. In this model, as asynchronously firing myosin-6 molecules engage along a filament, some effectively serve as anchors while others apply force in a geometry that locally produces compression. This phenomenon has been suggested to occur in a preprint by Ron Rock's lab (Krenc et al. bioRxiv 2016, DOI: 10.1101/068163, ref. 68). We now discuss this in the text (lines 241-250) and a new supplementary figure (Extended Data Fig. 4f.g), which includes a summary of all force geometries. We thank the reviewer for the helpful suggestion of including this figure, which we believe improves the clarity of the paper.

4. The reconstructions have resolutions ranging from 9 to 12 Å. Given this limited resolution, caution should be exercised in placing too much reliance on detailed analyses of subdomain

motions within actin subunits (\pm α -catenin), as depicted in several figures (e.g., 3c, 4f, Fig. S6). While the filament-level waves are clearly observable and analysable, attempts to assess subunit-level causes of these waves may be unjustified given the resolution constraints.

We appreciate the reviewer's point about resolution limitations, and we attempted to give a balanced interpretation of the MDFF results. We do note that the lack of appreciable distortions to actin subunits in our control fittings (Fig. 3c, Extended Data Fig. 7) suggest that the resolution range of our maps does not inherently drive spurious rearrangements using the restrained MDFF approach we employed. Moreover, the consistent pattern of subdomain rearrangements along protomers in both the supercoil reconstruction (Extended Data Fig. 7) and the α -catenin bound reconstruction (Extended Data Fig. 10e) argues against the recovered rearrangements being driven by noisy random fluctuations in the maps, as no symmetry has been applied. We have nevertheless included cautionary statements in the text about how higher resolution reconstructions would be required for detailed interpretations with atomic models (lines 385; 459-474). As the reviewer notes, we believe the most important conclusions of our paper can be made at the level of subunit-level repositioning (which we term "architectural remodeling"), yet we do believe based on the points discussed above that some level of subunit deformation-level interpretation is appropriate.

It is prudent to be cautious, particularly considering the significant variability in wavelength and the apparent infrequent observation of these waves. In other words, there is a risk of drawing elaborate conclusions based on a limited foundation.

Respectfully, we note that just because a biomolecular species is rare does not mean it is unimportant. For instance, DNA breaks (which can also be produced by biomechanical forces, e.g. during mitosis) are rare, yet they clearly have important physiological consequences.

Furthermore, it is a general (often undiscussed) practice in the cryo-EM field to include only a small subset of initially picked particles in the final reconstruction presented in a paper, yet this is usually presented as "the structure" of the macromolecular species under scrutiny. To use two arbitrary examples from a recent issue of *Nature* (May 1, 2025) Shahid et al. (PMID: 40108462) used 8.2 % of their initial picks in one of their reconstructions, while He et al. (PMID: 40044865) used 1.7% of their initial picks in one of their maps. The fraction of initially picked supercoil segments (276K) that we incorporated into our final reconstruction (13,146) is comparable (4.8%).

We believe that our ability to achieve interpretable reconstructions of rare (which is expected due to the non-equilibrium and energetically unfavorable nature of mechanically-excited structural landscapes) species, as well as our quantitative analysis of variability trajectories (Fig. 4b,c; Extended Data Fig. 10a) rather than simply using variability methods as a means of identifying homogenous particle subsets, are strengths, not weaknesses, of our study.

5. I am also quite sceptical for another reason: why do filaments in muscle sarcomeres, which experience significant forces from numerous myosin heads, not display waves (DOI: 10.1126/science.abn1934; DOI: 10.1016/j.cell.2021.02.047)? Although sarcomeres can contract, substantial force is developed on individual filaments from both sides of the Z-disk, yet waves are not observed during contraction or relaxation conditions. It may be suggested that filaments in sarcomeres are decorated with proteins like tropomyosin. Yet, a similar situation occurs in non-muscle cells.

We believe that the quantification of cellular tomograms we performed in response to the reviewer's point 1 convincingly show that sinusoidal filament regions are enriched in subcellular areas marked by zyxin, a proxy for high traction forces, in the non-muscle cells (Ptk2) we examined (Fig. 1c; Extended Data Fig. 1d-f).

Regarding muscle sarcomeres, it is difficult for us to reliably comment on why something "does not" happen, i.e. has not previously been reported in the scientific literature, in a system we have not studied. Thus, the following response is highly speculative.

First, per their respective Methods sections, the two studies the reviewer cites were performed solely on extracted muscle fibers prepared in ATP-free rigor buffer, where the fibers had been incubated for extended periods ("overnight" in both cases). We speculate that F-actin rearrangements dependent on active force generation would be unlikely to be preserved after this procedure.

Second, it is possible that the highly organized packing of thin and thick filaments in muscle prevents the formation of supercoils. Consistent with this speculation, in our cellular tomography data we anecdotally observed sinusoidal filament regions that were misaligned relative to the colinear F-actin arrays in which they were embedded, suggesting they may have "popped out" of the network, which may not be possible in the dense arrays of muscle:

However, as this phenomenon is challenging to quantify rigorously, we have elected not to discuss it in the manuscript.

Third, it is possible that the suite of highly specialized actin-binding proteins in muscle suppress the formation of supercoils. For instance, nebulin (which, unlike tropomyosin, is not found in non-

muscle cells), fills a groove which runs between the two F-actin strands as reported in [10.1126/science.abn1934](https://doi.org/10.1126/science.abn1934). This groove must remodel for supercoils to form, suggesting nebulin could block their formation. This is an intriguing hypothesis which could be explored through reconstitution studies in the future.

Fourth, and related to the second and third possibilities, it may be the case that supercoil formation is more likely to occur in damaged muscle. Interestingly, titin cleavage has been observed to result in disorganized A bands during muscle contraction using conventional electron microscopy (Li et al., *Elife* 2020. PMID: 33357376), which could be more permissive for the formation of supercoils. Future cryo-ET studies guided by markers of muscle damage could shed light on this idea.

Finally, fifth, it is possible that researchers have observed supercoils in muscle tomograms, but did not know how to interpret them and thus chose not to focus on them in their papers. We hope that our report that myosin forces promote the formation of supercoils will stimulate future work by the field on their distribution and function in diverse biological contexts.

As we have not directly examined muscle in our work, we have elected not to include a discussion of why supercoils have not (yet) been reported in the muscle literature in our paper.

6. It is suggested that α -catenin preferentially binds to superhelices at sites with a higher subunit rise. The interpretation is that α -catenin ‘recognises’ these regions to alleviate strain by repositioning subdomain 2 of the actin subunit it binds to. However, as previously noted, drawing conclusions about subdomain movements from such low-resolution maps is precarious. Furthermore, it seems exaggerated to imply that α -catenin selectively binds to these sites, given that it can also bind to canonical (straight) actin filaments that are not under tension, as acknowledged by the authors. An alternative explanation might be that the binding site for α -catenin is more accessible in regions with a higher rise and masked in regions with a lower rise. Consequently, this observation does not provide a definitive explanation for the force-activated binding of α -catenin.

We believe we have adequately responded to the reviewer’s legitimate concerns about resolution limitations in point 4. Similar to our response there, we note that the regular pattern of subdomain repositioning we observe when performing MDFF into the asymmetrically α -catenin decorated map (Extended Data Fig. 10e,f), distinct from those we observed for the undecorated supercoil (Fig. 3c, Extended Data Fig. 7), argues against our observations representing fitting map noise fluctuations.

We believe the reviewer’s statement of the potential mechanism is largely aligned with our interpretation: α -catenin is likely engaging positions with increased rise, allowing it to in turn stabilize actin subunits in a more compatible conformation through binding contacts. We of course acknowledge that our static structure cannot explain the precise order of dynamic molecular events which produce the final outcome we observe, which is an important topic of future study that would require substantial technical innovations beyond those we have already introduced in our paper. To clarify this point, we have expanded our discussion of the α -catenin mechanism and included a cautionary statement about resolution limitations in the “Limitations” section (lines 459-474).

7. The more I consider the data presented, the more I am concerned that we might merely be observing the reported filament supercoiling occurring as a result of motor-induced twisting (DOI:

10.1093/plphys/kiad095; DOI: 10.1038/s41467-022-28961-x), which occurs as myosin motors move in a spiral path along the actin filament. How can it be demonstrated that the slight increase in waves observed in the +ATP condition is not due to myosin supercoiling?

Reviewer #1 also inquired about the potential role of motor-generated torques. As we noted in our response there, myosin-5 and myosin-6 do generate modest torques, albeit at micron length-scales which are mismatched with the formation of the nanoscale supercoils we report here. This is aligned with the torsional waves in filopodia reported in 10.1038/s41467-022-28961-x, which have wavelengths and amplitudes on the micron scale. We believe the reference 10.1093/plphys/kiad095 may be in error, as this paper concerns the role of VILLIN2 in remodeling the plant (cotton) cytoskeleton, and we could not find any mention of supercoiling.

Nevertheless, we performed simulations where we applied a range of torques, including very high suprphysiological torques (Extended Data Fig. 4e; Supplementary Video 6). While this force regime caused clear twist remodeling of the filament, it did not produce supercoils. As we noted in our response to Reviewer 1, this is consistent with the lack of reports of supercoil formation in the presence of substoichiometric cofilin binding, which also remodels F-actin twist.

Along with the rest of the data presented in our paper, this highlights how supercoils are specifically produced by active myosin forces. Our data and simulations are consistent with this occurring through axial compression of F-actin, which can occur through multiple mechanisms in the presence of asynchronously operating motors. This axial compression is transduced into a torsional rearrangement and supercoil formation through the helical architecture of the F-actin lattice.

Minor Points:

1. Page 59: The numbering of Methods References should probably follow the order of references in the Main Text.

We thank the reviewer for catching this oversight: we have corrected the reference numbering.

2. Coarse-Grained Molecular Dynamics Simulations: These simulations may be valid and align with the experimental observations. However, they could be moved to the supplementary materials to maintain a concise and focused narrative. The primary emphasis should be on the strength and reliability of the experimental data. The simulations, while potentially informative, are supporting information rather than hard evidence.

We appreciate the reviewer's highlighting the importance of experimental structural studies for determining mechanisms of mechanical regulation by active motor forces. To our knowledge, our study is unique thus far in achieving this.

Nevertheless, we respectfully disagree that the simulations should be downplayed, as they also play a central role in our narrative. We have found them to be instrumental for interpreting the experimental data and testing the viability of different mechanistic hypotheses. Notably, the journal recruited a referee (Reviewer #4) specifically to assess the simulations. We believe that expanding this portion of our study in this revision has substantially strengthened the paper, and we believe that the current presentation (with some simulations featured in a main Figure) is balanced and appropriate.

3. Orientation distribution plots are not provided for any of the maps.

We have now provided orientation distribution plots for all cryo-EM reconstructions (Extended Data Fig. 6e; Extended Data Fig. 9d). As anticipated from the directional FSC plots (Extended Data Fig. 6c,d; Extended Data Fig. 9c), there are minor orientation biases in some reconstructions. However, again consistent with the directional FSC plots, the corresponding anisotropy in the maps is modest and does not substantially impact their interpretability.

Referee #3 (Remarks to the Author):

This is an excellent paper that provides new insights into both actin and actomyosin force generation. Very creative approaches have been used to reveal a supercoiling of actin filaments caused by interactions with myosin. Many questions will be raised by this work, opening up new areas of investigation. My concerns and suggestions are relatively minor, but I think that addressing them will improve the paper.

We thank the reviewer for noting the significance and creativity of our work, and its potential impact on opening new research directions for the field. We also appreciate the helpful suggestions, which we have addressed below.

First, I think that the term “spiral” should be dropped completely. The definition of a spiral is “winding in a continuous and gradually widening (or tightening) curve, either around a central point on a flat plane or about an axis so as to form a cone.” But there is no change in radius in these supercoils, and they are not conical, so the terms “supercoil” and “supercoiling” should be used.

We agree that it is best to be precise in the use of these terms, and we have replaced “spiral” and “spiraling” with “supercoil” and “supercoiling”, respectively. We have also replaced the term “superhelix” with “supercoil”, as these terms have identical meaning, to maintain a simpler and consistent terminology.

Consider bacterial and archaeal flagellar filaments. These, in general, are homopolymers, and non-motile mutants can be found that are straight. But the wild-type filament supercoils, and these helical supercoils have been studied for > 50 years. Similarly, a coiled-coil contains two alpha-helices that are supercoiled, and these are never described as spirals. Further, “oscillatory” should be replaced by “sinusoidal”, which is much clearer. The projection of a helix onto two-dimensions is a sinusoid, and the curves that are shown are sinusoidal, while oscillatory could be much more general.

I would also suggest that “domains” (as in “individual filaments featuring domains of sharply oscillating curvature spanning ~300-400 nanometers”) might simply be replaced by “regions”, as the term subdomains has been used to describe the structure of the actin molecule. It may be confusing for the reader if domains and subdomains are used, with one having nothing to do with the other.

We have replaced the term “oscillatory domain” with “sinusoidal filament region”.

I think that the paper could be strengthened with background information about how actin can be modified so that actomyosin force generation is inhibited, but the trivial explanations that these modifications prevent myosin binding, or fail to activate the myosin ATPase activity, can be dismissed. Some old examples are (Prochniewicz and Yanagida, 1990), (Prochniewicz et al., 1993), (Schwyter et al., 1990), (Drummond et al., 1990).

We note that *Nature* has strict word limits, but we have included a sentence of introduction about this indeed interesting point (lines 81-84) and included the suggested references (refs. 55-58).

The simulations shown are very interesting, but perhaps they can be extended and more information deduced from them. The most interesting question for me is what determines the pitch (wavelength) of these supercoils? The supercoils can be characterized by two parameters, the radius and the pitch. How are these related to the helical parameters of the actin filament? The helical parameters of the model can be easily changed, and it would be interesting to see the effects.

We thank the reviewer for this excellent suggestion. We have performed simulations modulating F-actin's helical parameters around their canonical values. We find that both supercoil peak-to-peak amplitude (analogous to 2x radius) and pitch increase significantly with F-actin helical rise. Helical twist has a more modest effect: the amplitude shows no clear monotonic trend, while the pitch displays a weak but statistically significant increasing trend (Extended Data Fig. 3g). This is aligned with the rise being asymmetrically altered in the supercoils (Fig. 3b), which was not observed in uniplanar bent actin (Extended Data Fig. 6h; ref. 63). We also discuss this in the text (lines 264-277).

How does the ellipticity of the supercoil arise? This is never discussed.

We attempted to address this question while modulating F-actin's helical parameters in simulations as described above, as well as flexural rigidity. We do see that altering these parameters changes the cross-sectional ellipticity of the supercoil, but there is no obvious trend (Extended Data Fig. 3i). Moreover, all conditions we tested maintained a high degree of ellipticity.

We thus turned to an alternative hypothesis, that ellipticity may be linked to the dynamics of filament remodeling during compressive force application. Consistently, in our compression simulations we observe that, in a segment which will supercoil, ellipticity is always maximal at the beginning of force application, remaining very high in 2 out of 5 simulations and decreasing in the other 3. This is consistent with the filament segment initially undergoing uniplanar buckling upon compression, transitioning to supercoiling as the end-to-end distance of the filament continues to decrease. We have updated the manuscript with this explanation (Extended Data Fig. 4h; Supplementary Movie 6; lines 278-289)

Is it purely the flexural rigidity of F-actin that determines the pitch, such that a more rigid actin filament would lead to a longer pitch? While the extensibility of the model is discussed, is the flexural rigidity of the model reasonable, and does it yield a persistence length of ~ 10 microns, as determined experimentally?

A target persistence length of 9 microns (based on published data on the persistence length of ADP F-actin measured experimentally, PMID: 7744781, ref. 72) was used to optimize the force field

during our parameter scan, which ensures a reasonable flexural rigidity (Extended Data Fig. 3c). Because flexural rigidity scales linearly with persistence length per Equation 7 ($P = \frac{B_s}{k_B T}$; P , persistence length; B_s , flexural rigidity; k_B , Boltzmann constant; T , absolute temperature), we used persistence length as a proxy for flexural rigidity. We modulated it by adjusting the harmonic angle potential constants in our simulations. We find that the supercoil pitch increases with persistence length, aligned with the Reviewer's intuition, while the amplitude shows no clear trend (Extended Data Fig. 3h).

The modulation of the helical rise in the filament is very interesting, but some additional references could strengthen this aspect of the paper. It was traditionally assumed in the earliest models for muscle that F-actin was inextensible. But x-ray diffraction observations from the 1990s showed otherwise. There is a very interesting discussion of this (Goldman and Huxley, 1994), particularly the last paragraph, where it is asked if actin filaments could play a more active role in muscle force generation than simply acting as a rigid lattice.

We have included this commentary reference (ref. 54) when discussing the pioneering work of Huxley and Wakabayashi in showing F-actin rise modulation by muscle contraction (discussed in lines 79-81, where we already cite the two key primary references which are the subject of the Goldman / Huxley commentary [refs 52 and 53]).

With regard to the modulation of the helical twist, it should be noted that the dramatic change in actin's twist by ADF/cofilin is only ~ 5 degrees (from -167 to -162 degrees), significantly less than what is observed here.

We have added a sentence noting that the distortions we observe here are more extreme than those imposed by cofilin binding (lines 324-325).

On p. 11 the 12 \AA map is referred to as "moderate" resolution. I think that this description is rather dated, and we might refer now to a 5 \AA map as having moderate resolution. This map should simply be described as having a poor resolution.

We have adjusted the statement to "low resolution" (lines 376; 385).

Drummond, D.R., Peckham, M., Sparrow, J.C., and White, D.C. (1990). Alteration in crossbridge kinetics caused by mutations in actin. *Nature* 348, 440-442.

Goldman, Y.E., and Huxley, A.F. (1994). Actin compliance: are you pulling my chain? *Biophysical Journal* 67, 2131-2133.

Prochniewicz, E., Katayama, E., Yanagida, T., and Thomas, D.D. (1993). Cooperativity in F-actin: chemical modifications of actin monomers affect the functional interactions of myosin with unmodified monomers in the same actin filament. *Biophysical Journal* 65, 113-123.

Prochniewicz, E., and Yanagida, T. (1990). Inhibition of sliding movement of F-actin by crosslinking emphasizes the role of actin structure in the mechanism of motility. *J. Mol. Biol.* 216, 761-772.

Schwyster, D.H., Kron, S.J., Toyoshima, Y.Y., Spudich, J.A., and Reisler, E. (1990). Subtilisin cleavage of actin inhibits *In vitro* sliding movement of actin filaments over myosin. *J. Cell Biol.* 111, 465-470.

Referee #4 (Remarks to the Author):

In this work, Carl, Reynolds, and coworkers report on structural studies of actin filaments under force generated by myosin motors, and implications for actin binding protein recruitment. I have been aware of this work for some time due to presentations from Prof. Alushin and his trainees and in general feel it to be novel and significant. I'm not aware of other studies where high resolution structures of proteins under force can be solved, so that would constitute a valuable methodological advance for the rapidly growing field of mechanobiology.

We thank Prof. Hocky for his appreciation of the novelty and significance of our work, and its potential to advance the mechanobiology field. We also appreciate the helpful suggestions and critiques of the simulation work, which we address below.

I was asked to comment specifically on the computational model in this work. To the best of my ability to judge, the model presented is appropriate and has been treated correctly. The code to parameterize and run the model using the EspressoMD software is available in the GitHub for this paper, and it seems to match what is reported in the paper. (Unfortunately, I didn't have time to run it myself just getting EspressoMD configured on short notice, but I would be happy to verify that before publication for the authors/editors after it is updated as suggested below).

I have a few minor comments on the modeling section -

1) The authors refer to the stiffness of filaments computed by Chu and Voth. In that paper, they compare to the stiffness of 44 pN/nm measured in this paper (<https://doi.org/10.1073/pnas.91.26.12962>) so it may be worth also citing these experiments if they are still considered correct (I do not think this 33% difference would alter any conclusions).

We chose not to use this experimentally measured stiffness of 44 pN/nm in our force field optimization because that value was obtained from phalloidin-bound F-actin, which is known to exhibit greater stiffness than the ADP F-actin used in our cryo-EM/ET studies. Moreover, the experimental setup in that work did not guarantee that the measured actin filament is oriented perpendicular to the deflecting needle, and any angular deviation could compromise the measurement's accuracy.

2) The authors might consider mentioning a CG model for actin we developed (<https://doi.org/10.1016/j.bpj.2017.05.016>) that has similar resolution, and discuss the differences in how it was parameterized.

We thank Prof. Hocky for this suggestion. We have added a discussion comparing / contrasting the approach in this paper versus the present study to the Methods section (lines 1386-1395; ref. 117).

3) In the model software, there are two stages, parameterization at zero temperature and then assessing the properties with different mechanical perturbations at room temperature. As far as I can tell, right now in the GitHub, the two are mixed together so that the simulation script will perform the mechanical perturbations at zero temperature. Perhaps these can be separated into two steps with a little more instruction for performing the first calculation and taking the output to set the parameters for the second one. Also, the four simulations performed in sim.py are all either

compression or extension, depending on the sign convention, and I guess while separating out these instructions, the authors could give examples of both.

We have now separated the parameterization and the application of mechanical perturbations into two distinct scripts, named `param_scan.py` and `run_sim.py` (uploaded to Github), respectively. In this updated version of the code, tensile parameters (harmonic bond constants) were optimized at extremely low temperature (0.298 K) to minimize deviations caused by thermal fluctuations. Bending (harmonic angle potential constants) and twisting parameters (dihedral potential constants) were optimized at room temperature (298 K) by fitting to the persistence length of ADP-bound F-actin and the published cumulative twist variance as a function of actin subunit index, respectively. All mechanical perturbations were conducted at 298 K. We have also added extensive documentation and comments throughout the code to improve clarity.

4) Given the way that EspressoMD treats units (as in, it does not use units), perhaps the authors could be a bit more explicit with how they ascribe the force units to be piconewtons, either in the manuscript, readme for the code, or both. It may follow directly from the extensional stiffness calibration but I'm not positive. On a perhaps related note, table S2 reports k_l axial and lateral in pN/nm but perhaps those should be kJ/mol/nm² if all the other angular springs are in kJ (per mol) (because I observe that 265 and 53 appear in the code without unit conversion).

Due to updated assumptions in our revision (e.g., the relative stiffness between bonds associated with the same type of interaction) and an updated protocol that separately parameterizes the harmonic angle and dihedral potentials, the optimized force field parameters have been adjusted accordingly. These changes, along with the derivation of parameter units, are detailed in the Methods section (lines 1361-1449). The updated parameters are provided in Extended Data Table 2.

5) In the methods it says "The dynamics of an actin filament under force were simulated by fixing the initial five subunits and applying pulling or pushing force on the terminal five subunits (across both strands)." In the code, they have set up pulling on both strands like this, and also on a single strand. Given this sentence in the methods, it seems the manuscript only reports data where both strands are pulled and released, but I want to confirm that.

In early exploratory work, we simulated mechanical perturbations by applying tensile or compressive forces to a single F-actin strand. However, the results were highly similar to those obtained by simultaneously applying force to both strands. As these findings did not offer additional insights, we chose not to include them in the manuscript. We have removed the associated functions from the updated code.

6) I also think that the frequency of transient pulling/pushing on the filament is not listed in the manuscript, even though this might just be in simulation units. Moreover, is there a dependence on the wavelength of coiling versus the frequency of transient pulling?

In the transient pulling/pushing experiments, the filament was subjected to a single pulling or pushing event for a defined duration before being released. Due to space constraints and the already extensive number of conditions we examined, we have not examined the effect of pulling/pushing frequency on the resultant wavelength. The most physiologically relevant scenario, that of stochastically firing motors raised by Reviewer #1, produces supercoils which are morphologically very similar to those generated by either continuous compression or a single

application of compression followed by release (Extended Data Fig. 4d). We have more extensively discussed these points, as well as several other findings from the simulations, in the revised manuscript (lines 191-289, Methods lines 1351-1539)

7) It is not clear from the manuscript how many times the authors repeated the simulations in Fig 2f. While I don't think there is any need for error bars there since these are just representative images, if they did replicates then it would be great if they could specifically mention that the behavior happens every time, most of the time etc for however many repeats. Given Fig 2g, it seems they did indeed repeat the apply/release simulations, but I don't see the number of replicates for that histogram.

In our initial submission, the simulations were essentially deterministic since they were performed at zero temperature and without viscosity. However, in our revised manuscript we have now incorporated both thermal fluctuations and viscosity, as noted in our responses to the other Reviewers, which introduces variability. Therefore, for each condition, five independent simulations were now performed. We have now included this information (N = 5) in the figure legends and Methods.

-

Glen Hocky
Assistant Professor
New York University

Referee #4 (Remarks on code availability):

See above

We thank the referees for their thoughtful comments on our revised manuscript. Our responses are below.

Referees' comments:

Referee #1 (Remarks to the Author):

The revised version is very significantly improved. Important new data have been added, the text has been rewritten and some conclusions have been revised. In particular, I find that the description of, and the proposed explanation for the oscillatory filament shapes that occur when motors of both polarities exert forces, now make more sense and are more convincing. The simulations have also been improved, and are now done in more realistic conditions.

The authors have addressed nearly all my points, and I recommend that their work be published provided that they address my remaining concern:

I am confused by the strategy followed to ensure that all the biotinylated actin is near the pointed end, i.e. what the authors call “filament annealing refractory conditions.” In the methods, on page 45, they indicate that biotinylated actin is added in the second polymerization step, with rhodamine actin (line 1135). Maybe I missed something, but what would be the point of having biotinylated actin in both polymerization steps?

We thank the Reviewer for catching this, this was indeed an error. Biotinylated actin was only included in the second step for the fluorescence microscopy experiments we used to calibrate the assay, where streptavidin was used to anchor the filaments to coverslips for TIRF imaging. We have corrected the Methods section to indicate that no biotinylated actin was included in this step for cryo-EM grid preparation.

Assuming this is a mistake, and that no biotinylated actin was used in the second step, I still find the strategy questionable because there is certainly plenty of annealing taking place during the one hour incubation of the biotinylated seeds with unbiotinylated G-actin (the second polymerization step, before capping). The preceding, shearing step should have created very short biotinylated segments that would then be incorporated all along the filament. Also, at the end of the first polymerization step, there should still be significant amounts of monomeric actin (even if steady-state is reached), including biotinylated monomers that would then copolymerize with the actin added in the second polymerization step. It is thus quite likely that the filaments contain several biotinylated actin segments, that would be too small to be detected in images like the ones shown in Extended Data Fig 2e-f.

I hope the authors will be able to address this remaining point, which is a bit technical but quite important, because their work should be an important contribution to the field.

We thank the Reviewer for recognizing the significance of our work. As the Reviewer notes, we cannot fully exclude the possibility that there is a small amount of biotinylated actin present in the extensions from seeds, although we believe we have minimized it to the extent feasible. However, we note that even if this were the case, it would not impact our conclusions. As noted in the cartoon in Extended Data Fig. 4g, we believe that the stochastic operation of motors produces compressive forces in the pointed-end directed (myosin-VI alone) force condition. This model invokes a strongly bound myosin-VI molecule localized towards the barbed end relative to a force-generating molecule, such that the filament segment between the two motors is compressed. Operationally, small patches of biotin actin could serve an analogous anchoring role to the strongly-bound myosin-VI molecule in this scheme. The key take-home message is that stochastic ensembles of motors of either directionality can produce compressive forces that elicit F-actin supercoiling, a conclusion we believe is now adequately supported by the data presented (thanks in part to excellent feedback from all the Reviewers).

Related, minor points: With the indicated actin concentrations, the biotin fraction should be 20% not 25% (line 1130) ;

Thanks for catching this: corrected.

In Extended Data Fig 2e-f the colors used are counter-intuitive because, according to the methods, the biotin-seeds are made with ATTO-488 actin and should be represented green ;

Respectfully, we believe the data are clearer as presented. It is much easier to see magenta seeds on the background of green filaments rather than vice-versa. As we have clearly annotated the figure, we believe it is understandable to the reader.

The legend does not explain what the arrowheads in Extended Data Fig 2f represent.

We have removed these arrowheads, which were not necessary to understand the figure.

Referee #2 (Remarks to the Author):

The authors have performed a substantial amount of work, particularly in relation to the simulations, to address earlier criticisms, and they should be commended for this effort. Technically, the study is a tour de force, employing a wide range of cutting-edge methodologies. Conceptually, the paper advances the claim that forces exerted by myosin motors on actin filaments generate regions of supercoiling that are subsequently recognised by mechanosensitive proteins, exemplified here by α -catenin. Similar supercoils (or sinusoidal regions) are reported in cells at actin filament sites under load, identified by zyxin enrichment.

We thank the Reviewer for appreciating the technical strength of our studies and the effort put forth in our revision.

Molecular dynamics simulations are presented in support of the findings; however, these appear to be flexibly adapted to the proposed model rather than providing independent predictive insight.

We respectfully disagree with this assertion. Notably, our original submission proposed an alternative model for supercoil formation, where tension and release could produce supercoiling. This was effectively falsified by our improved simulations, and we updated our conclusions accordingly.

Setting this aside, the central question is whether myosin-dependent supercoiling is indeed the mechanism by which mechanosensitive proteins recognise filaments in cells. If the answer were yes, this would represent a major advance, fully warranting publication in Nature. The difficulty is that I am not convinced the evidence, based on limited data that barely cross the threshold of significance, is sufficiently strong to sustain such a conclusion.

That actin filaments can undergo supercoiling as a result of myosin compressive forces is plausible, and merits publication in a more specialised journal, yet I find the biological significance of this observation overstated. In crowded cellular environments such as adhesions, actin filaments are frequently bundled and bound by multiple proteins, including some forty tropomyosin isoforms. Under these conditions, it is difficult to see how myosin-dependent supercoiling would be required for proteins such as α -catenin, zyxin, or other LIM-domain family members to bind. Structural changes in inter-subunit bonds of the actin filament under tension alone, without invoking supercoiling, could adequately explain the recruitment of tension-sensitive proteins.

We agree with the Reviewer that there are likely additional mechanisms that control the localization of other mechanosensitive actin-binding proteins in the cell. We did our best to be careful in articulating this, particularly for LIM-domain proteins like zyxin. Our data show that supercoil recognition can explain the mechanosensitive behavior of α -catenin. We believe much of the significance in our paper lies in demonstrating, to our knowledge for the first time, that it is possible to directly visualize force-evoked structural changes generated by active motor forces which underlie such mechanosensitive recognition. The methodology we introduce sets the stage for analyzing many other categories of mechanosensitive proteins in the future.

Finally, I must note that this manuscript is difficult to read, not due to the complexity of the data, but to the convoluted style of writing. Once the text is disentangled from the results, it becomes clear that the substance presented does not, in my view, justify publication in Nature. Papers published in Nature rapidly come to be regarded as definitive, and I am not persuaded this study merits that level of endorsement.

We are sorry that the Reviewer did not find our paper straightforward to read. We have revised the text to shorten it and clarify the presentation, which we hope has made it more approachable for the general readership of *Nature*.

Referee #3 (Remarks to the Author):

This is a monumental body of work being reported in a single paper, so it is by necessity quite dense. While I had enthusiasm for the initial manuscript, this revised paper is greatly improved compared to the initial submission. All of my concerns have been adequately addressed. In cases where conclusions are not firmly established or speculative, the authors clearly indicate that this is the case. I found only one mistake (p.3-4): “Consistently, chemical modifications of F-actin have been reported which are permissive for myosin binding and ATPase activation but refractory to motor force production, implying conformational reciprocity between the filament and myosin’s mechanochemical cycle⁵⁵⁻⁵⁸.”

But ref. 57 involves mutations in actin, not chemical modifications!

We thank the Reviewer for their strong endorsement of our revised manuscript. We have adjusted this sentence to differentiate between papers reporting actin mutations versus those reporting chemical modifications.

Referee #4 (Remarks to the Author):

I feel the authors have done their best to respond to the criticisms/comments with substantial extra effort, and I still feel that the new methodology presented in this work is a major advance worthy of publication in *Nature*.

Referee #4 (Remarks on code availability):

I did not have the opportunity to run the code myself, but it looks appropriate to the problem and to do what is claimed in the manuscript.

We thank the Reviewer for their strong endorsement of our revised manuscript and for their careful review of the simulations, which are a major pillar of this work.

We thank the reviewer for this particularly thoughtful and constructive comment. We have added two sentences to the end of the first Discussion paragraph (lines 359-363) which we believe distill this point. We agree this addition addresses a component of Reviewer 2's significance concern, and it correspondingly strengthens the paper.